# Factor-Wise Homogeneity of Slot-Attention for Continual Object-Centric Learning

## Abstract

Can current AI models continually learn object-centric representations? Object-Centric Learning and Continual Learning are both critical areas of AI research, yet their intersection remains underexplored. In this work, we observe that Slot Attention, a popular OCL method, exhibits a distinctive behavior: It organizes latent representations into small and separated regions, each of which preserves the same factor states, referred to as *factor-wise homogeneity*. This phenomenon emerges not only in previously trained data but also in upcoming data with unseen factor states, offering significant advantages for continual learning that incrementally expands factor states, such as novel shapes. To harness this property, we propose a simple and effective method, *Decoder only Post Replay*, that freezes the encoder and the Slot Attention as a generator of factor-wise homogeneous representations and employs a decoder-only fine-tuning strategy after the novel task training is done. Although Slot Attention has been widely studied, its representational behavior has been largely overlooked. This paper highlights its unique strengths in continual object-centric learning. We also introduce a novel validation and analysis environment for Continual-Object Centric Learning, establishing a strong baseline for future research.

## 1 Introduction

The ability to comprehend the compositional structure of complex scenes lies at the core of human intelligence. Humans naturally interpret visual input in terms of discrete, object-centric units, which facilitates continual generalization to novel and dynamic environments. Object-Centric Learning (OCL) Greff et al. (2020) aims to replicate this capability by learning a set of disentangled representations, each corresponding to an individual object without the need for explicit supervision. OCL has recently gained increasing attention for its strong compositional generalization, with applications spanning object detection and segmentation Lee et al. (2024); Zhang et al. (2021); Wang et al. (2023b); Zoran et al. (2021), image generation Wang et al. (2023c); Akan & Yemez (2025); Wu et al. (2023); Jiang et al. (2023), embodied AI Hamdan & Güney (2024); Baek et al.; Wu et al. (b); Zadaianchuk et al.; Didolkar et al., complex reasoning Mondal et al. (b; 2024); Stammer et al., and even text-based tasks Park et al. (2024); Kim et al. (2023a); Ma et al. (2023); Behjati & Henderson.

Despite these advances, existing OCL methods, such as Slot Attention Locatello et al. (2020), remain confined to single-task settings, where generalization is typically evaluated within fixed training environments. These assumptions impose fundamental limits on OCL's capacity for generalization, particularly in scenarios that require the sequential acquisition of new object categories while avoiding catastrophic forgetting McCloskey & Cohen (1989), a long-standing challenge in neural network research. When information is introduced continuously, previously acquired representations are prone to drift or collapse, inducing overlap that compromises the stability of the slot space and diminishes compositional interpretability. These scenarios raise a critical challenge: the effectiveness of OCL depends on achieving and enhancing generalization, which requires stable representations under incremental integration. While prior work has examined OCL generalization in out-of-distribution and zero-shot settings Dittadi et al. (2021); Didolkar et al. (2025); Chen et al. (2024) and explored the reusability of object-centric features Pan et al., the behavior of Slot Attention under continuous introduction of information has not yet been systematically investigated.

In this paper, we take the first step toward enabling class-incremental scenarios within the unsupervised object discovery task in OCL. Inspired by previous pioneering works in class-incremental

learning Li & Hoiem (2017); Rebuffi et al. (2017); Shmelkov et al. (2017); Castro et al. (2018); Michieli & Zanuttigh (2019); Cermelli et al. (2020), we propose Continual Object Centric Learning (C-OCL) benchmarks to evaluate the ability of OCL models to discover novel objects under continual task settings. We build upon two widely used benchmarks in OCL research Burgess et al. (2019); Locatello et al. (2020); Greff et al. (2019); Dittadi et al. (2021) and introduce their class-incremental variants: Continual-Tetrominoes and Continual-CLEVR.

While Object-Centric Learning methods have seen growing adoption, their robustness under class-incremental settings remains largely unexplored. To address this gap, we examine how Slot Attention Locatello et al. (2020) organizes its latent space in continual environments, aiming to analyze its representational dynamics under sequential task through our proposed benchmarks. Interestingly, we found that Slot Attention structures its latent representations into small, well-separated regions, each consistently preserving the same factor states. We refer this property as *factor-wise homogeneity*, in which each region preserve identical semantics and semantically different slots are separately distributed. Importantly, this behavior generalizes beyond single tasks and persists across even continual training tasks, offering significant advantages for continual learning. To harness this property, we propose a simple, effective, and efficient method, *Decoder-only Post Replay*, which freezes the encoder and Slot Attention as a generator of factor-wise homogeneous representations and employs post replay strategy only to the decoder. Our results show that DPR leverages the inherent factor-wise homogeneity property of Slot Attention to recover performance efficiently, without modifying the core architecture. We validate its effectiveness on our C-OCL benchmarks and further extend the evaluation to more general environments, providing an in-depth comparative analysis against baseline methods.

We summarize our main contributions as follows: (1) We analyze the internal dynamics of the slot representation space and identify a property of Slot Attention, termed *factor-wise homogeneity*, in which latent representations organize into compact, well-separated regions that consistently preserve factor states, consistent under continual learning. (2) We propose Decoder-only Post Replay, a simple and efficient method that fine-tunes only the decoder to exploit this property without modifying the core architecture. (3) We introduce the first continual object-centric learning benchmarks, conduct in-depth analyses validating DPR, and extend the evaluation through an extension to more general environments, thereby confirming its generalizability.

## 2 RELATED WORK

**Continual Object-Centric Learning**   Recently, OCL Burgess et al. (2019); Greff et al. (2019); Engelcke et al.; Locatello et al. (2020) has attracted growing interest in a variety of applications. Several studies Gokul et al. (2022); Wang et al. (2023b) have demonstrated that the structural representations learned by OCL are effective for object localization tasks, where object-centric representations have been used to guide unlabeled object segmentation Zoran et al. (2021); Lee et al. (2024) and to mitigate dataset biases Zhang et al. (2021). OCL has also shown benefits in image generation Wang et al. (2023c), enabling systematic generalization in generative models Singh et al. (a); Wu et al. (2023); Jiang et al. (2023); Akan & Yemez (2025). Furthermore, OCL has been applied to learn compositional scene structure for understanding dynamics Hamdan & Güney (2024); Baek et al.; Wu et al. (b); Zadaianchuk et al.; Didolkar et al., and to support more complex forms of reasoning Mondal et al. (b; 2024; a); Stammer et al., as well as to model relational structures among discrete language tokens Park et al. (2024); Kim et al. (2023a); Ma et al. (2023); Behjati & Henderson. Despite OCL methods have been adopted in various domains, the core challenge of adapting OCL to continual object discovery remains largely underexplored. Generalization to OOD Dittadi et al. (2021), zero-shot Didolkar et al. (2025); Chen et al. (2024), re-usability Pan et al. has been discussed in previous works. Although not specific to OCL, shape-texture consistency (STCR Shi et al. (2025)) improves representation robustness in continual learning, aiding both generalization to new data and reduction of forgetting. However, the ability to continuously handle novel objects without forgetting remains an open challenge.

**Representation Analysis of Slot Attention**   Slot Attention Locatello et al. (2020) is one of the most widely used frameworks in object-centric learning. It produces a set of vectors, called *slots*, which bind to individual objects in the input image. Previous studies have investigated the *identifiability* of slots Jia et al.; Kim et al. (2023b); Brady et al. (2023); Didolkar et al.; Kori et al., where each slot consistently corresponds to the same object or semantic concept across varying inputs.

Liu et al. (2025) utilized additional prototypes to remove duplication between slots and consistent object assignments. Other works have examined Slot Attention in terms of (factor-wise) compositionality Chang et al. (2022); Singh et al. (b); Jung et al.; Montero et al.; Wu et al. (a); Mansouri et al. (2023); Wiedemer et al.; Baek et al.. Specially, Wiedemer et al. provides a theoretical guarantee of compositional generalization to unseen combinations of latent slots, Biza et al. (2023) analyzes the equivariance and invariance of slot representations to object pose, and Mansouri et al. (2023) leverages weak supervision from perturbations for disentanglement learning. Advances in object-centric learning address the weaknesses of Slot Attention on complex datasets through pretrained models Seitzer et al.; Kakogeorgiou et al. (2024) and structural consistency methods Wang et al. (2023d); Singh et al. (2022); Wen et al. (2022); Manasyan et al. (2025); DJukic et al. Wen et al. (2022); Manasyan et al. (2025); Zhao et al. (2025). Among these, DINOSAUR Seitzer et al. combines self-supervised pretrained models and achieves strong performance on complex images, making it a solid basis for future works. However, little attention has been given to how slots are distributed in the representation space, motivating our empirical analysis of their internal dynamics.

**Replay-based Methods in Continual Learning** Experience Replay Lin (1992) mitigates catastrophic forgetting by jointly training on a mixture of current task data and stored samples from previous tasks within each mini-batch. Subsequent variants Riemer et al. (2018); Lopez-Paz & Ranzato (2017b); Chaudhry et al. (2018); Buzzega et al. (2020b); Rolnick et al. (2019); Kumari et al. (2022) have aimed to enhance replay effectiveness by improving sample quality or sampling strategies. However, the question of *when* to incorporate replay samples has received comparatively less attention. Some prior works have explored two-stage strategies Li et al. (2024); Liu et al. (2023); Wang et al. (2023a); Gupta et al. (2020); Unal et al. (2023); Wang et al. (2024); ji et al. (2024), either by introducing auxiliary modules Wang et al. (2023a; 2024) or by jointly training with current and replay samples in separate phases Li et al. (2024); Liu et al. (2023); ji et al. (2024). Notably, Gupta et al. (2020); Unal et al. (2023) proposed distinct stages of current-task training and jointly training with current and replay samples, while performing balancing to alignment between tasks. We emphasize the importance of replay scheduling, we propose *Post Replay*, which trains on replay samples and a subset of current-task data *after* completing the main training on the current task.

## 3 CONTINUAL OBJECT-CENTRIC LEARNING

**Goal of Novel C-OCL Benchmark** The goal of our benchmark is to analyze and evaluate the ability of object-centric methods in the task of *unsupervised object discovery*, where novel objects are continuously introduced. The task of *object discovery* involves decomposing an input image into a set of latent representations, often referred to as *slots*, where each slot is expected to encode information about a distinct object in the scene. These slots are individually decoded into partial reconstructions, which are then composed to recover the original image, enabling object-level understanding without explicit supervision. However, as novel object classes are incrementally introduced across tasks, models are required not only to generalize to the new classes presented in the current task, but also to maintain their performance on the classes encountered in previous tasks.

**Continual Learning Scenarios** We begin by formalizing the task of C-OCL. The model is trained on a stream of tasks $\mathcal{T} = \{\mathcal{T}_t\}_{t=1}^{M}$, where $M$ is the total number of tasks and each $t$-th task $\mathcal{T}_t$ comprises unlabeled multi-object images $x \in \mathcal{X}$ drawn from a disjoint set of novel object classes $\mathcal{C}_t$. In this work, we focus on introducing novel *shape* classes, as *shape* provides a broader range of variation compared to *position* or *color*, which are limited to bounded continuous ranges. We assume that the object classes $\mathcal{C}_t$ are mutually exclusive across tasks: $\mathcal{C}_t \cap \mathcal{C}_k = \varnothing$ for all $t \neq k$, ensuring that objects presented in task $\mathcal{T}_t$ never appear in any previous or future task.

**Datasets and Evaluation Metrics** To evaluate the performance of OCL methods within the C-OCL, we introduce two benchmarks: (1) Continual-Tetrominoes, and (2) Continual-CLEVR. These datasets build upon the original Tetrominoes Kabra et al. (2019) and CLEVR Johnson et al. (2017) used in Locatello et al. (2020); Kabra et al. (2019). To better support C-OCL, we augment the original datasets with additional object (*shape*) classes. We adopt three training and evaluation scenarios inspired by prior work Shmelkov et al. (2017); Michieli & Zanuttigh (2019); Cermelli et al. (2020), with modifications tailored for object-centric learning. The scenarios are defined as follows: (1) *Single Step addition of Two classes* (SST), (2) *Single Step addition of Multiple classes* (SSM), and (3) *Multi Step addition of Two classes* per step (MST). We follow quantitative evaluation in Dittadi et al. (2021). For evaluation, follow Dittadi et al. (2021) and use Foreground Adjusted Rand

Index (FG-ARI) Rand (1971), mean squared error (MSE) score, Segmentation Covering (SC) Arbelaez et al. (2010) and mean Segmentation Covering (mSC) Engelcke et al.. Details of our datasets (including training scenarios) and evaluation metrics are discussed in Appendix C and Appendix B.5.

# 4 FACTOR-WISE HOMOGENEITY OF SLOT ATTENTION

We begin by examining the internal dynamics of Slot Attention Locatello et al. (2020), one of the most widely used OCL methods. Prior works have primarily assessed its effectiveness through downstream tasks, leaving its representational structure in the latent space underexplored. Moreover, how core object-centric frameworks adapt to continual object discovery remains a fundamental open challenge. In this section, we investigate the internal dynamics of the slot representation space.

## 4.1 PRELIMINARY

**Slot Attention for Object Discovery**    Slot Attention Locatello et al. (2020) is consisted with three components: (1) an encoder, (2) a slot attention module, and (3) a decoder. Given an input image $x_i \in \mathbb{R}^{C \times H \times W}$ of $t$-th task $\mathcal{T}_t$, the encoder $\mathcal{F}_t : \mathcal{X}_t \to \mathcal{S}_t$ extracts $D$-dimensional features $\boldsymbol{Z}_i \in \mathbb{R}^{N \times D}$. Then, slot attention module $\mathcal{SA}_t : \mathcal{Z}_t \to \mathcal{S}_t$ produces a set of $K$ vectors called *slots* $\boldsymbol{S}_i \in \mathbb{R}^{K \times D_{slots}}$, where each slot corresponds to an individual object present in the input image $x_i$. Slots are initially sampled from the Gaussian distribution with learnable $\mu$ and $\sigma$. Each slot is assigned to each distinct feature from the given inputs $\boldsymbol{Z}_i$ through iterative steps. Each step performs dot-product attention Luong et al. (2015) between $\boldsymbol{Z}_i$ and normalized slots from previous steps. Aggregated slots are updated using a Gated Recurrent Unit (GRU) Chung et al. (2014) and a residual multi-layer perceptron (MLP) with normalization. Finally, each slot is independently decoded using a spatial broadcast decoder Watters et al. (2019); Locatello et al. (2019) $\mathcal{G}_t : \mathcal{S}_t \to \hat{\mathcal{X}}_t$, and the model is trained to minimize mean squared error.

**Empirical Analysis Setting**    We investigate the distribution of slots $\boldsymbol{S}_i \in \mathcal{S}$ trained on the C-Tetrominos dataset under the SST setting. Since the model is trained without label supervision, we assign pseudo-labels to each slot to examine its representational semantics. Following *mask matching* in Dittadi et al. (2021), we assign *semantic* pseudo-labels (*shape, position, color*). Representations are collected across different tasks $\mathcal{T}_t$. We label them as *task pairs* (E$i$/T$j$), representing evaluation on task $\mathcal{T}_i$ using a model continuously trained from the initial task up to task $\mathcal{T}_j$. We visualize the resulting slot representations using t-SNE Van der Maaten & Hinton (2008), where each point corresponds to a slot and is color-coded according to its assigned label. Note that we only consider slots of foreground objects in the following sections. For comparison, we also evaluate SlotMLP Locatello et al. (2020), a variant that replaces the slot attention module with a MLP. A detailed illustration of this analysis setup is provided in Appendix A.1.

## 4.2 FACTOR-WISE HOMOGENEITY FOR REPRESENTATION SEPARATION ACROSS TASKS

**Internal structure of slot representations**    Figure 1a, reveals several notable behaviors. **(1) Local Separation within Single Task.** For slots originating from the same task (i.e., *intra-task* pairs such as (E0/T0), where no continual learning effects are present, we find that the representation space is organized into small, localized regions. This pattern holds consistently across other intra-task pairs, indicating that Slot Attention dynamics naturally induce such localized distinctions in the representation space. **(2) Separation from Upcoming Tasks with Unseen Factors.** We also observe clear *inter-task* separation. In particular, the separation between (E0/T0) and (E1/T0) shows no overlap, even when the latter involves completely unseen factor states. Since the model has not yet encountered images from (E1) during training on (T0), this separation confirms that the behavior is primarily driven by the encoder and slot attention modules, independent of the decoder. Crucially, following experiments revealed the decoder to be the generalization bottleneck, constraining the application of the learned features to novel compositions. **(3) Separation from Previous Tasks.** Importantly, the separation between (E0/T1) and (E1/T1) highlights a separation between representations of previous task (E0) and novel task (E1) after training on novel task (T1). Even under continual updates, the representation space consistently separates into small, localized regions across different task pairs.

**Evaluation of Inter-Task Separation**    For quantitative evaluation of the task-wise region separation, we introduce two metrics: (1) Top-$k$ Inter-Task Class Similarity, and (2) Top-$k$ Nearest

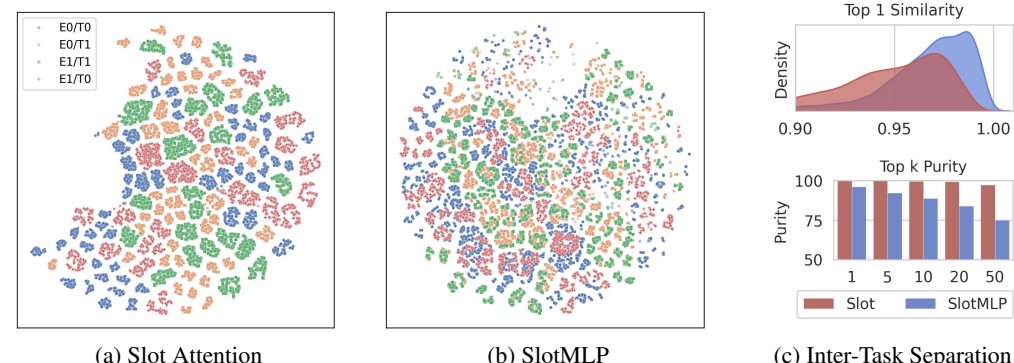

|             |           |                        |
|:-----------:|:---------:|:----------------------:|
| (a) Slot Attention | (b) SlotMLP | (c) Inter-Task Separation |

Figure 1: (a, b) Slot inter-task separation visualization via t-SNE. (c) Quantitative evaluation of inter-task separation using (top) Top-$k$ Inter-Task Class Similarity and (bottom) Top-$k$ Nearest Neighbor Class Purity. "Slot" denotes Slot Attention, which exhibits well-separated features across tasks.

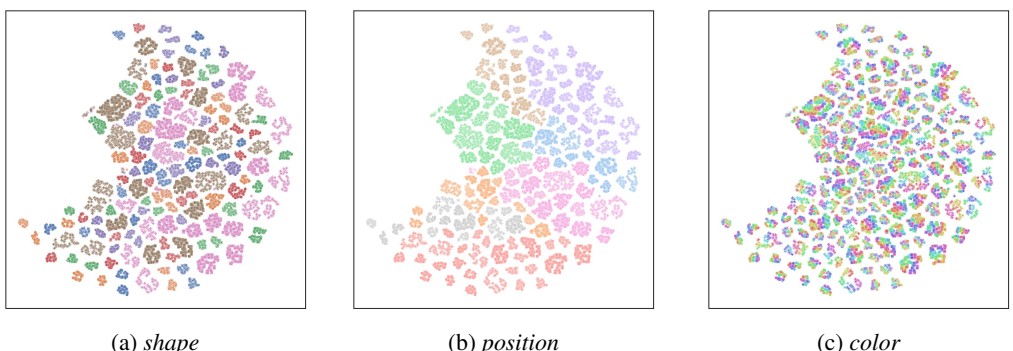

|             |           |           |
|:-----------:|:---------:|:---------:|
| (a) *shape* | (b) *position* | (c) *color* |

Figure 2: t-SNE visualization of slot representation to show the homogeneity of the states for each factor as *shape, position*, and *color*. Each color indicates a state of each factor.

Neighbor Class Purity. The Top-$k$ Inter-Task Class Similarity (Equation 1) measures the similarity between a given slot and its top-$k$ nearest neighbors belonging to different *task pair* classes. This metric reflects the degree of local similarities between inter-task representations. For each slot, the Top-$k$ Nearest Neighbor Class Purity (Equation 2) evaluates the proportion of top-$k$ nearest neighbors that share the same label. It measures the semantic purity of local neighborhoods in the representation space. Both metrics rely on Cosine Similarity, providing two views: one highlights inter-task similarity, the other emphasizes intra-task consistency.

$$\text{Similarity}^k(s_i) = \frac{1}{k} \sum_{s_j \in \mathcal{N}_k^*(s_i)} \cos(s_i, s_j), \text{where } \mathcal{N}_k^*(s_i) = \max_{s_j \in \mathcal{S}\ ,\text{task}(s_j)\neq\text{task}(s_i)}^{k} \cos(s_i, s_j) \quad (1)$$

$$\text{Purity}^k(s_i) = \frac{1}{k} \sum_{s_j \in \mathcal{N}_k(s_i)} \mathbb{K}(\text{task}(s_j) = \text{task}(s_i)), \text{where } \mathcal{N}_k(s_i) = \max_{s_j \in \mathcal{S}}^{k} \cos(s_i, s_j) \quad (2)$$

Compared to the original Slot Attention (Figure 1a), the representations produced by SlotMLP exhibit greater overlap across *tasks* (Figure 1b). The inter-task similarity metric (Figure 1c, top) shows that Slot Attention yields significantly lower density of high-similarity neighbors, indicating lower cross-task similarity and hence clearer task-level distinctions. Similarly, the nearest neighbor class purity (Figure 1c, bottom) demonstrates that Slot Attention maintains higher purity and suffers less degradation as $k$ increases. Together, these results indicate that Slot Attention preserves clearer task boundaries and more localized representations, whereas SlotMLP tends to blur them and remain overlapped. This confirms that Slot Attention effectively forms well-separated distributions across tasks, yielding consistent results under continual learning.

**Factor-wise Homogeneity of Slot Representation** To further analyze the separated regions, we visualize slots using their semantic labels. As shown in Figure 2, slots within the same region

consistently preserve identical semantic factor values (*intra-region consistency*), while slots with different factor values are pushed into distinct regions (*inter-task separation*).

We refer to this property as ***Factor-wise Homogeneity***: across the slot representation space, slots exhibit semantic consistency with respect to individual factors while remaining well separated from those of different factors. Importantly, this phenomenon extends beyond a single task and persists across both unseen and previously trained tasks (e.g., (E0/T0) and (E1/T0)), indicating that factor-wise homogeneity is preserved even under continual learning scenarios. This constitutes a robust inductive bias that organizes slots into semantically coherent and task-generalizable structures.

### 4.3 LEVERAGING FACTOR-WISE HOMOGENEITY FOR C-OCL

From the previous section, we identified *factor-wise homogeneity* property, where slots form well-separated regions preserving identical factors across continual tasks. Leveraging this property, we propose a simple yet effective method that mitigates performance degradation in C-OCL.

**Benefits of Factor-Wise Homogeneity to C-OCL** We focus on the factor-wise homogeneity property observed in slot representations. Due to the factor-wise homogeneity property, these representations remain well-separated in the slot representation space $\mathcal{S}$, even across tasks (i.e. (E0/T1) VS (E1/T1) in Figure 1a). Such separation provides a strong inductive bias for continual learning scenarios, mitigating performance degradation caused by representation overlap Buzzega et al. (2020a); Dittadi et al. (2021); Zhu et al. (2021); Kim et al. (2024).

**A Limitation of Factor-Wise Homogeneity as a Standalone Solution for C-OCL** Although the factor-wise homogeneity property offers strong benefits for continual object-centric learning, it is primarily observed from the encoder and slot attention module. A limitation arises in the decoder trained on task $\mathcal{T}_t$. For instance, while the encoder and slot attention successfully preserve factor-wise homogeneity of slot representations our subsequent experiments demonstrate that the decoder consistently acts as a generalization bottleneck. To address this, we employ a replay buffer that stores selected samples from previous tasks and utilize them to fine-tune the decoder. This allows the model to recover past performance without disrupting the upstream components.

**A Minimal Implementation: Decoder-Only Post Replay** We aim to fully leverage factor-wise homogeneity property of Slot Attention in the C-OCL, while avoiding degradation due to unintended training biases. To address this, we propose a simple yet effective method *Decoder-only Post Replay* (DPR). DPR is based on two core components: (1) we freeze the encoder and slot attention module and only fine-tune the decoder, in order to maintain factor wise separated representations observed in Slot Attention space $\mathcal{S}$; and (2) we introduce a *Post Replay* (PR) strategy, wherein the model is fine-tuned after the *task training phase*, i.e., training on the current task $\mathcal{T}_t$ without any continual learning methods, thus completely excluding sources of interference interference from replay biases during training on novel task Wu et al. (2019); Rolnick et al. (2019).

## 5 EXPERIMENTS

We validate our proposed Decoder-only Post Replay method on the Continual-Object Centric Learning benchmarks. We conduct additional in-depth analysis under various comparison baselines. Specifically, we compare DPR against: (1) non-object centric baseline, and (2) various replay-based approaches, and (3) regularization-based methods. We further evaluate the (4) *factor-wise homogeneity* property in more challenging environments such as real-world and complex datasets.

### 5.1 SETTINGS

**Training DPR in C-OCL** We conduct experiments on two C-OCL benchmark with three training scenarios introduced in Section 3. Dataset details are provided in Appendix C. DPR employs a replay buffer to store representative samples from previous tasks. Following Wu et al. (2018), we store the reconstructed samples $\hat{x}$ using the model trained on each task and randomly sample among them per each task. During Post Replay for task $\mathcal{T}_t$, we construct a replay set by combining these samples with an equally sized subset of current task inputs $x^t$, and only fine-tune the decoder. Models are trained using the Adam optimizer Kingma & Ba (2014) with a learning rate of $4 \times 10^{-4}$. We apply a warm-up phase for the initial 2% of training steps. The model is trained for 200 epochs on the initial task $\mathcal{T}_0$, 100 epochs on subsequent tasks $\mathcal{T}_{t>0}$, and 50 epochs for post replay. Training configurations are reported in Appendix B.2, and the algorithms are provided in Algorithm 1.

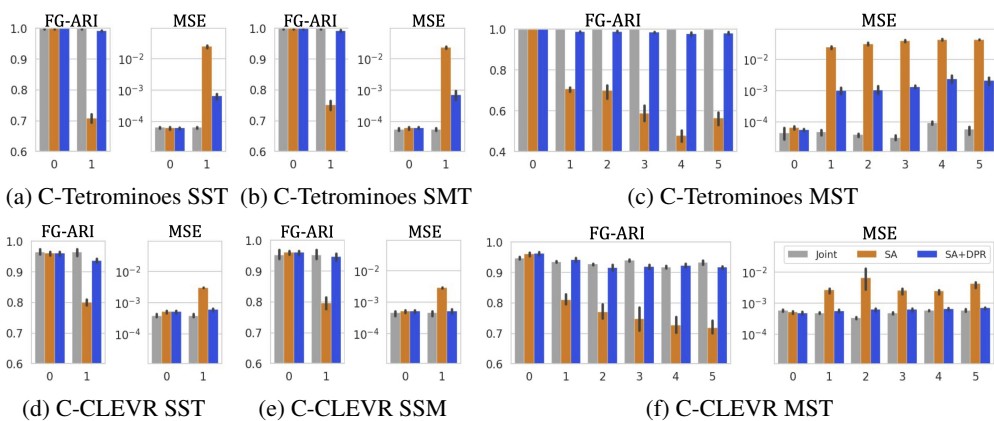

(a) C-Tetrominoes SST    (b) C-Tetrominoes SMT         (c) C-Tetrominoes MST

(d) C-CLEVR SST      (e) C-CLEVR SSM          (f) C-CLEVR MST

Figure 3: Performance on C-OCL benchmarks measured by FG-ARI(↑) and MSE(↓), averaged over the current and all previously seen tasks after each task training. (Joint: Slot Attention trained with joints of tasks, SA: Slot Attention, SA+DPR: Slot Attention with DPR)

**Settings for Comparative Analysis**    To evaluate the *factor-wise homogeneity* property of Slot Attention, we construct diverse baselines. For **(1) comparison against a non-object-centric baseline**, we use a Variational Autoencoder (VAE) Kingma et al. (2013) equipped with a spatial broadcast decoder Watters et al. (2019), consistent with the decoder architecture in Slot Attention. We apply proposed PR and DPR methods to the VAE, along with standard Experience Replay, as described below. We then **(2) compare with DPR and replay-based baselines**. We apply standard Experience Replay (ER) to Slot Attention (SA+ER). Following prior work Lopez-Paz & Ranzato (2017a); Rebuffi et al. (2017); Chaudhry et al. (2019); Buzzega et al. (2020a), we perform *joint training* by concatenating replay samples with the current task's training data in the mini-batch. For fair comparison with PR and DPR, we align the replay buffer size in SA+ER by randomly sampling 2,000 examples per task. We also evaluate a generative variant (SA+GR) motivated by Generative Replay (GR) methods Shin et al. (2017); Wu et al. (2018); Ayub & Wagner; Zhai et al. (2019); Gao & Liu (2023), which utilize an auxiliary generative model to produce replay samples. and performs joint training with generated samples. We adopt a VAE as the generative model, following the setup in Shin et al. (2017). In addition, we **(3) compare DPR with regularization-based methods**, commonly used in class-incremental learning. We evaluate DPR in comparison with Slot Attention combined with Learning without Forgetting (LwF) Li & Hoiem (2017) (SA+LWF) and Elastic Weight Consolidation (EWC) Kirkpatrick et al. (2017) (SA+EWC). To assess the compatibility and robustness of DPR with respect to these strategies, we also report results for hybrid models that combine DPR with each regularization method. Finally, to **(4) evaluate in more challenging environments**, we use real world datasets, COCO Lin et al. (2014) and PASCAL VOC Everingham et al. (2010). We split each dataset into multiple tasks ($|\mathcal{T}| = 2, 4$) with disjoint objects. However, since Slot Attention does not scale well to complex data, we use DINOSAUR Seitzer et al., a slot-based model integrated with pre-trained DINO Caron et al. (2021), which outperforms Slot Attention. We report FG-ARI and mean Best Overlap (Equation 13) of instance ($mBO^i$) and class levels ($mBO^c$) as evaluation metrics. In addition, we extend our evaluation of vanilla Slot Attention to complex synthetic data using CLEVRTex Karazija et al., a more challenging variation of CLEVR with rich textures and diverse backgrounds. Full details of each evaluation are demonstrated in Appendix A.9 A.10 A.11 A.12.

For comparison, we introduce a variant of our method, *Decoder-only Replay without Task training* (DRwT), which leverages the separated representations of Slot Attention for unseen tasks. Unlike DPR, DRwT does not perform task training on new tasks; instead, it relies solely on replay samples and a subset of the novel task dataset to extend the reconstruction capability of the decoder , while keeping both the encoder and the slot attention module frozen (Algorithm 3).

## 5.2 PERFORMANCE RESULTS

**Quantitative Results**    Figure 3 reports performance averaged over the current and previous tasks after continuously training task $\mathcal{T}_t$. Comparing Slot Attention with and without DPR, we observe clear degradation on previously learned tasks when training continues on novel ones, whereas applying DPR significantly improves performance across datasets and evaluation scenarios. Results with

additional metrics show consistent improvements (Appendix A.6), indicating that the *factor-wise homogeneity* property of Slot Attention helps preserve a well-separated slot representation space, providing a strong inductive bias for continual learning scenarios where simple decoder refinement suffices to improve object discovery performance. We provide numerical results in Appendix A.7. Beyond overall performance, we verify that DPR benefits extend beyond *shape* to other semantic factors. Regarding Figure 2c, *color* features may appear less clearly separated in the global latent space; however, they still form localized regions in the slot representation space. Furthermore, we evaluate the robustness of DPR beyond *shape*, where novel *color* factor is introduced. We observe consistent improvements, as shown in Figure 29.

**Qualitative Results** Figure 4 presents reconstruction results for each slot. Without DPR, Slot Attention (middle row) tends to reconstruct objects biased toward the task $\mathcal{T}_1$, generating incorrect shapes. With DPR (bottom row), however, each slot yields reconstructions with semantically accurate object shapes for previous task (right, red arrow highlights)), while effectively preserving reconstruction quality in novel tasks (left, blue circle highlight). Results for other scenarios are in Appendix A.8.

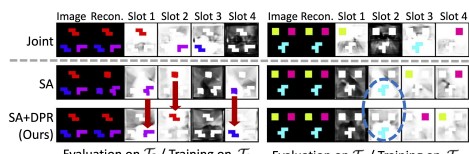

Figure 4: Qualitative reconstruction results of Slot Attention with and without DPR.

## 5.3 IN-DEPTH ANALYSIS

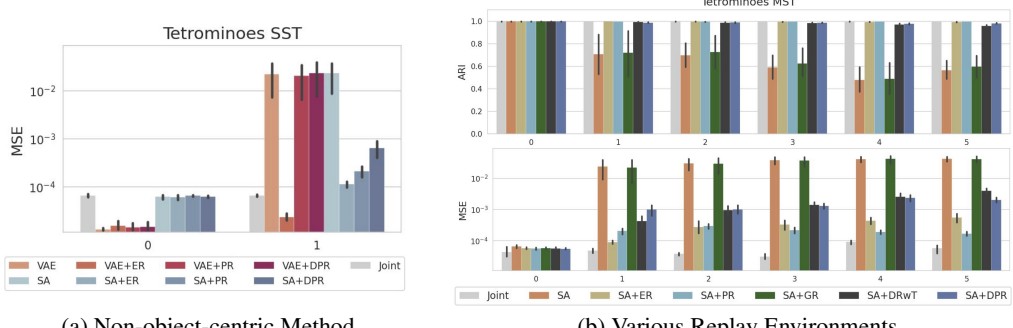

(a) Non-object-centric Method    (b) Various Replay Environments

Figure 5: In-depth analysis results under various comparison baselines. (a) Comparison with non-object-centric baseline (VAE) on C-Tetrominoes under the SST setting. (b) Comparison under various replay environments using C-Tetrominoes with the MST scenario.

**Necessity of Slot Attention for Enabling PR and DPR** We evaluate whether DPR and PR are effective for non-Slot Attention methods. Figure 5a presents a comparison with the non-object-centric baseline, VAE. We found that vanilla VAE suffers from performance degradation after learning new tasks, while applying ER leads to improvement. However, PR and DPR do not yield similar benefits as observed with Slot Attention, where VAE still exhibits substantial degradation. In contrast, Slot Attention consistently benefits from both PR and DPR, showing robust performance improvements. These results suggest that the effectiveness of DPR and PR depends on the presence of well-separated and factor-wise consistent latent representations-i.e., the factor-wise homogeneity property observed in Slot Attention. Results for other scenarios are provided in Figure 21.

**Comparable Performance without Novel Task Learning** DRwT leverages the zero-shot generalization induced by the factor-wise homogeneity property, and does not require any *task training phase* for novel tasks. In DRwT, both the encoder and slot attention modules are frozen after training on the initial task ($\mathcal{T}_0$), and only the decoder is updated using replay samples and a subset of data from the current task. As shown in Figure 5b, SA+DRwT achieves performance comparable to SA+DPR, even outperforming it in earlier tasks. This result indicates that the separated representations induced by factor-wise homogeneity reduce the reliance of the task training phase.

**Further Improvement through Relaxing the Decoder-Only Constraint** PR relaxes the *decoder only* constraint imposed by DPR by allowing fine-tuning of both the encoder and Slot Attention during the post-replay phase. While replays are still performed after the task training phase on $t$-th

task $\mathcal{T}_t$, PR offers flexibility in leveraging the factor-wise homogeneity. As shown in Figure 5b, both SA+PR and SA+DPR improve upon vanilla Slot Attention. Among the two, SA+PR achieves higher performance, particularly in longer task sequences. These results suggest that, while DPR provides a minimal implementation that preserves factor-wise homogeneity, further gains can be achieved by relaxing this constraint—so long as the homogeneity property is not disrupted via PR.

**Ablation of Slot Attention Components on Task Separation** We ablate two components of Slot Attention—slot Gaussian initialization and GRU updates—to assess their effect on *factor-wise homogeneity*. Removing GRU updates shows noticeable increased overlaps across tasks, and inter-task separation metrics likewise show reduced distinction among representations from different tasks (Figure 11). Together, these results suggest that recurrent updates are essential for sustaining factor-wise homogeneity. Details of ablation study are provided in Appendix A.2.

### 5.4 Extension to More General Environments

**Comparison with Various Replay Environments** Figure 5b presents the performance of Slot Attention under different replay methods across a multi-step task sequence. We observe that SA+ER consistently improves performance over training without replay. Notably, SA+PR performs comparably to SA+ER in earlier tasks but clearly surpasses it as the task sequence advances. We also find that SA+GR is ineffective for Slot Attention. Results for other scenarios are provided in Figure 22.

**Robustness to Continual Learning Regularizer** Figure 6 presents comparisons and combinations with regularization methods in continual learning. Compared to DPR, SA+LwF and SA+EWC are less effective for reconstruction when applied to Slot Attention, as they continue to suffer performance degradation after training on new tasks. Interestingly, combining SA+LwF or SA+EWC with DPR yields slightly improved performance on MSE metric over using DPR alone, suggesting that DPR is robust to integration with regularization in continual learning. Results for other scenarios are provided in Figure 23.

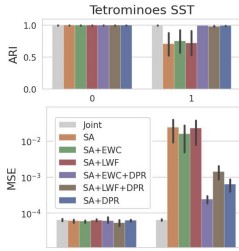

Figure 6: Robustness to regularizer combinations in continual learning.

**Performance on Real-World Datasets with Practical Pre-trained Backbone** We further evaluate the *factor-wise homogeneity* property in more challenging environments to assess its robustness and scalability beyond simplified synthetic benchmarks and basic models. In Table 1, where four sequential real-world tasks are given, our DPR demonstrates consistent performance improvements on real-world datasets. Also, visualization of slots indicate that *factor-wise homogeneity* is preserved and slots remain well separated across tasks (Figure 24) and DPR effectively improve mask prediction qualities (Figure 25) . These results clearly confirm that factor-wise homogeneity provides a robust inductive bias that extends beyond synthetic settings to real-world continual learning scenarios. The performance gap increases with longer task sequences, highlighting the limitations of DINOSAUR in retaining prior performances. In these cases, our method yields increasingly larger gains, showing that *factor-wise homogeneity* provide effective inductive biases even under long-term continual learning (Table 14). Additionally, evaluation on complex synthetic data such as CLEVR-Tex shows consistent results, further confirming the robustness of our findings in visually complex settings (Table 15). Details are provided in Appendix A.12.

| Dataset | Model | Evaluation on $\mathcal{T}_0$ / Trained on $\mathcal{T}_0$ | | Evaluation on $\mathcal{T}_0$ / Trained on $\mathcal{T}_3$ | |
|---------|-------|---------|---------|---------|---------|
| | | FG-ARI | mBO$^c$/mBO$^i$ | FG-ARI | mBO$^c$/mBO$^i$ |
| COCO | DINOSAUR | $23.01_{\pm 0.8}$ | $45.87_{\pm 0.4}$ / $35.43_{\pm 0.4}$ | $21.25_{\pm 0.6}$ | $42.14_{\pm 0.6}$ / $32.76_{\pm 0.6}$ |
| | DPR$^\dagger$ | | | $\mathbf{23.03}_{\pm 0.8}$ | $\mathbf{43.47}_{\pm 0.5}$ / $\mathbf{33.80}_{\pm 0.6}$ |
| PASCAL | DINOSAUR | $18.01_{\pm 0.6}$ | $55.06_{\pm 0.8}$ / $50.11_{\pm 0.4}$ | $14.63_{\pm 0.7}$ | $51.47_{\pm 0.8}$ / $47.26_{\pm 0.4}$ |
| | DPR$^\dagger$ | | | $\mathbf{16.69}_{\pm 0.6}$ | $\mathbf{52.92}_{\pm 0.7}$ / $\mathbf{48.56}_{\pm 0.5}$ |

Table 1: Evaluation on $|\mathcal{T}| = 4$ tasks of real-world images (5 runs, mean$_{\pm \text{std}}$). We evaluate on $\mathcal{T}_0$ after training on $\mathcal{T}_0$ and continuously trained to $\mathcal{T}_3$. DPR$^\dagger$ denotes DPR with DINOSAUR.

## 5.5 DISCUSSION & LIMITATION

Our core contribution lies in identifying factor-wise homogeneity in Slot Attention, which functions as a strong inductive bias for continual learning and, which is not tied to decoder architecture. Leveraging this structural robustness, we introduce DPR, which effectively mitigates the decoder generalization bottleneck in C-OCL.

That said, we acknowledge that decoder design remains an important axis of improvement. Stronger decoders may inherently enhance reconstruction and generation quality, even without explicit mechanisms like DPR. While our experiments cover Slot Attention with a spatial broadcast decoder and DINOSAUR with an autoregressive transformer decoder, we do not explore a broader spectrum decoders, such as diffusion-based object-centric decoders Jiang et al. (2023); Wu et al. (2023). This constitutes a limitation of the present work and an interesting direction for future research.

One intuitive approach to avoid decoder generalization bottlenecks is to reduce reliance on the decoder and instead depend solely on the robustness of Slot Attention's learned representations. However, object-centric frameworks yet require decoder outputs to satisfy the reconstruction-based objectives. Thus, this work focuses primarily on analyzing and revealing the intrinsic continual-learning behavior of Slot Attention, rather than proposing new decoding strategies. Systematic improvements to decoding or prediction modules are therefore left for future research.

## 6 CONCLUSION & FUTURE WORK

In this paper, we take the first step forward enabling continual object-centric Learning. We highlight the unique behavior of Slot Attention, referred as *factor-wise homogeneity*. This property organizes the slot representations into small, well-separated regions that preserve identical semantics, providing strong inductive bias for continual learning. To harness this property, we proposed *Decoder only Post Replay*, a simple yet effective method that fine-tunes only the decoder while preserving the property of Slot Attention. Finally, we introduce the first C-OCL benchmarks and conduct in-depth analyses that validate our findings, further extending evaluation to more general environments.

Representation collapse between distinctive features remains a fundamental challenge not only in continual learning but in neural network. Our findings suggest that factor-wise homogeneity may offer a promising direction for mitigating such collapse, both for seen and unseen features. Moreover, enabling task-wise separation without novel task learning points toward a potential paradigm shift in designing effective continual learning systems. We hope future work explores the broader implications of factor-wise homogeneous representations across diverse frameworks and applications.

## 7 REPRODUCIBILITY STATEMENT

We ensure the reproducibility of our work as follows:

- To verify the reproducibility of our findings on *factor-wise homogeneity* and the proposed method *Decoder-only Post Replay* (DPR), we provide detailed analyses in Section 4, experimental results in Appendix A, implementation and training configurations in Appendix B, and demonstrations of our proposed benchmarks (C-Tetrominoes, C-CLEVR) in Appendix C.

- To reproduce our implementation, we include the complete code for training DPR in the supplementary material, along with a README file that provides detailed instructions for setup and usage.

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

## A  APPENDIX: EXPERIMENT SETTINGS AND RESULTS

### A.1  SLOT ATTENTION REPRESENTATION SPACE

We trained the Slot Attention model as described in Appendix B.3. We provide t-SNE visualizations for Slot Attention and its variants, including SlotMLP, Slot w/o GRU, and Slot w/o Initialization.

**Implementation of t-SNE visualization**  We visualize the slot representations using t-SNE Van der Maaten & Hinton (2008) based on semantic pseudo-labels: *shape*, *position*, and *color*. These labels are assigned to each slot via *mask matching*, which uses the Hungarian algorithm Kuhn (1955), following the procedure in Dittadi et al. (2021) and the matching loss formulation in Locatello et al. (2020). Given $M$ ground-truth object masks and $K$ predicted masks for each slot, we treat the label assignment as a bipartite matching problem. We compute a cost matrix between all pairs of predicted and ground-truth masks, and apply the Hungarian algorithm to find an optimal one-to-one assignment that minimizes the cost. Once the matching is established, we transfer the semantic labels of the matched ground-truth objects to the corresponding slots. However, note that *task pair* labeling used in Section 4.2 is GT labels, which does not require mask matching.

**Implementation of SlotMLP**  We follow SlotMLP from Locatello et al. (2020). `SlotMLP` replaces Slot Attention module with Multi-Layer Perceptron (MLP). Table 2 demonstrated detailed implementations of SlotMLP.

| Module | Layer / Operation | Details |
|---|---|---|
| Encoder | Conv $5 \times 5$ | 32 channels, stride 1, padding 2, ReLU |
| | Conv $5 \times 5$ | 32 channels, stride 1, padding 2, ReLU |
| | Conv $5 \times 5$ | 32 channels, stride 1, padding 2, ReLU |
| | Conv $5 \times 5$ | 32 channels, stride 1, padding 2, ReLU |
| | Adaptive AvgPool2d | Resize to $(16 \times 16)$ |
| | Positional Embedding | Added to feature map before flatten |
| | GroupNorm + Conv $1 \times 1$ | Output shape: $(B, 32, 256)$ |
| SlotMLP | Input Flattening | Flatten to $(B, 32 \times 16 \times 16)$ |
| | 4-layer MLP | $512 \rightarrow 1024 \rightarrow 1024 \rightarrow K \times D$ |
| | Per-slot MLP | Applied independently to each slot |
| | Output | $(B, K, D)$ slot representations |
| Decoder | Spatial Broadcast | Repeat each slot $\rightarrow (D, H, W)$ |
| | Positional Embedding | Added to each slot map |
| | Conv $5 \times 5$ | 32 channels, stride 1, padding 2, ReLU |
| | Conv $5 \times 5$ | 32 channels, stride 1, padding 2, ReLU |
| | Conv $5 \times 5$ | 32 channels, stride 1, padding 2, ReLU |
| | Conv $3 \times 3$ | 4 ch. (RGB+mask), stride 1, padding 1 |

Table 2: Architecture specification of the SlotMLP baseline.

**Implementation of Slot Attention without GRU**  . The outputs of the attention module (referred as `updates` from  Locatello et al. (2020)) are updated using a Gated Recurrent Unit (GRU) Chung et al. (2014) followed by a residual multi-layer perceptron (MLP) with normalization. We ablate the GRU from the Slot Attention. Outputs of the attention module (*updates*) are directly passed to residual MLP.

**Implementation of Slot Attention without Random Initialization**  Slot Attention produces a set of $K$ *slots* $\boldsymbol{S} \in \mathbb{R}^{K \times D_{slots}}$, where each slot is initially sampled from the Gaussian distribution with learnable $\mu$ and $\sigma$. To evaluate the effects of random initialization of slots, we trained a modified version of Slot Attention which consists of learnable embeddings, similar to Jia et al.. Instead of initializing $\boldsymbol{S} \in \mathbb{R}^{K \times D}$ from $\mathcal{N}(\mu, \sigma)$, we use $\boldsymbol{S}_{embed} \in \mathbb{R}^{K \times D_{slot}}$ as slot initial representations and follow the Slot Attention process.

**t-SNE visualization results** Each dot in the visualization is colored according to its semantic pseudo-label. The marker shape of each dot indicates the source of the slot: a *circle* represents slots from (E0/T0), a *cross* represents (E0/T1), *square* represents (E1/T1), *plus* represents (E1/T0).

Figure 7 shows the t-SNE visualization of Slot Attention slot representations using the C-Tetrominos SST dataset. We used 2,000 validation images, each contributing $K{=}4$ slots. Although the background slots are included in the t-SNE computation, they are excluded from the visualization for clarity. Note that we discretized labels for *position* since have continuous values.

The semantic labels are defined as follows:

- *Shape*: 7 foreground object classes + 1 background class

- *Position*: 8 foreground spatial bins + 1 background class

- *Color*: 10 foreground color classes + 1 background class

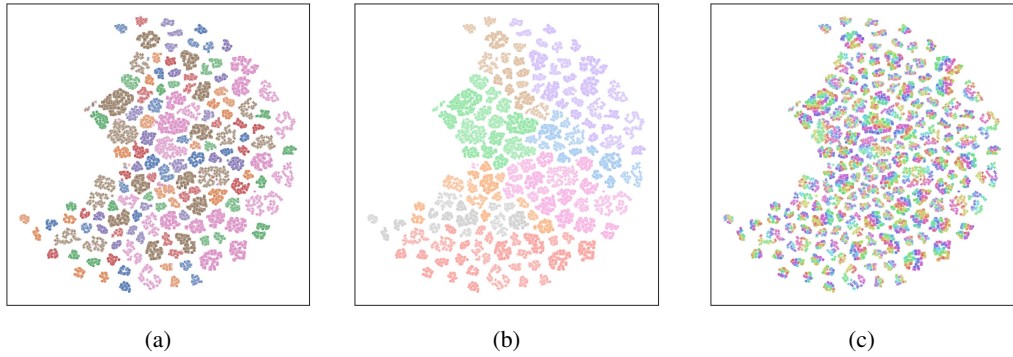

(a)           (b)           (c)

Figure 7: t-SNE visualization results of slots from Slot Attention. Each color of the dots represent (a) *shape*, (b) *position*, (c) *color*.

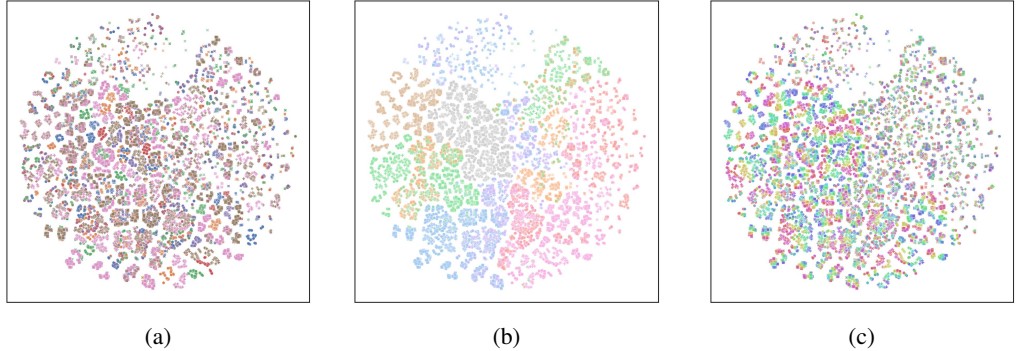

(a)           (b)           (c)

Figure 8: t-SNE visualization results of slots from SlotMLP. Each color of the dots represent (a) *shape*, (b) *position*, (c) *color*.

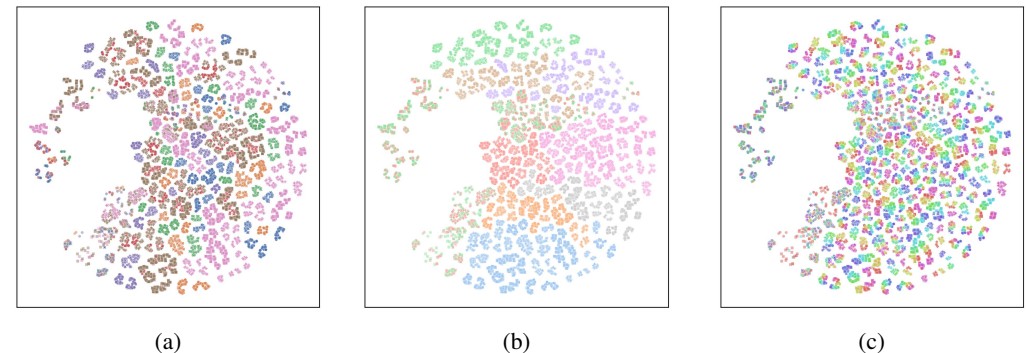

Figure 9: t-SNE visualization results of slots from Slot w/o Initialization. Each color of the dots represent (a) *shape*, (b) *position*, (c) *color*.

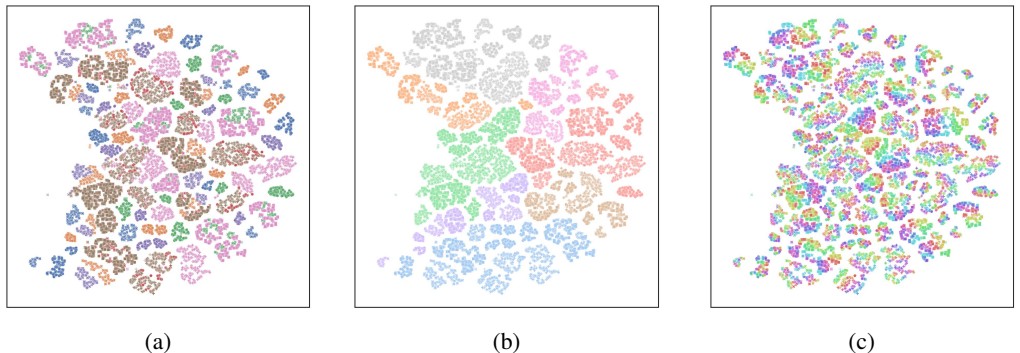

Figure 10: t-SNE visualization results of slots from Slot w/o GRU. Each color of the dots represent (a) *shape*, (b) *position*, (c) *color*.

## A.2    ABLATION ANALYSIS OF SLOT ATTENTION COMPONENTS

We investigate which architectural components of Slot Attention contribute most significantly to the emergence of factor-wise homogeneity. We focus on two key components: (1) *Random initialization* of slots, and (2) *GRU updates*. To evaluate the individual contribution of each, we conduct targeted ablation studies by selectively disabling or modifying them (Appendix A.1).

**Ablation Results**    From Figure 11, ablating the GRU component results in a substantial degradation in factor-wise homogeneity across both metrics. This degradation is also evident in the visualizations (Figure 11b), which show increased inter-task overlap and reduced separation among slot representations. While the ablation of random initialization has a less pronounced effect than GRU, it still leads to noticeable degradation.

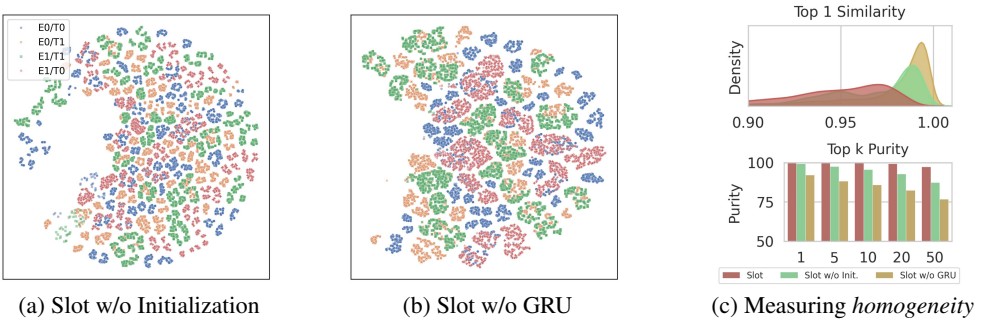

(a) Slot w/o Initialization            (b) Slot w/o GRU            (c) Measuring *homogeneity*

Figure 11: t-SNE visualization using ablation of Slot Attention. Each colors indicate a *task pairs*. *Homogeneity* is measured using Density of Top-$k$ Inter-Task Class Similarity per point (top), and Top-$k$ Nearest Neighbor Class Purity (down). Term 'Slot' denotes Slot Attention.

### A.3 ANALYSIS ON GRU DYNAMICS AND FACTOR-WISE HOMOGENEITY

In Section 4, we found factor-wise homogeneity property of Slot Attention, where slots consistently preserve identical semantic factor values while differentiating with different factors. This property extends beyond a single task and persists across continual tasks, preserving inter task separation which provides a strong inductive bias for continual learning scenarios, mitigating performance degradation caused by representation overlap.

In this section we further analysis the emergence of factor-wise homogeneity property of Slot Attention and inter task separation in continual settings. From our ablation study in Section 5.3 and Appendix A.2, ablating the GRU component in Slot Attention causes an identifiable reduction of factor-wise homogeneity and inter-task separation.

The main question we address here is whether the recurrent update dynamics of the GRU are responsible for the stability of factor-wise homogeneity under continual adaptation. To empirically analyze this phenomenon, we leverage recent theoretical studies on time-scale dynamics in gated RNNs, which demonstrate that gating mechanisms effectively create "slow modes"—hidden-state dimensions that evolve slowly and stably, serving as reliable channels for long-term information preservation Tallec & Ollivier (2018); Krishnamurthy et al. (2022); Can et al. (2020); Smith et al. (2021); Miller & Hardt. We hypothesize that these slow modes are utilized by Slot Attention to stabilize factor-specific information.

We employ the GRU update gate $z_t$ as an empirical measure of dimensional stability (a 'slow mode rank') and compute the Spearman's rank correlation across slots to confirm that slots sharing identical semantic factors rely on the same highly stable dimensions. Our analysis confirms that the GRU effectively performs an identical-mapping operation conditioned on each object's intrinsic variability, stabilizing factor-specific slow modes. Furthermore, these stability characteristics are robustly maintained under continual learning conditions, providing structural foundation for the robust task-level separation observed in Slot Attention. This slow-mode mechanism successfully explains both the factor-wise homogeneity and the task separation observed in Slot Attention under continual learning.

**Time-scale dynamics in gated RNNs**  A substantial body of work has shown that gated recurrent architectures inherently produce multi–time-scale dynamics in their hidden states. Theoretical analyses demonstrate that gating mechanisms effectively learn local time-steps, causing certain hidden-state dimensions to evolve slowly and thus act as stable carriers of long-range information Tallec & Ollivier (2018). Subsequent studies provide direct evidence that gated RNNs generate marginally stable directions that persist across extended horizons, forming slow modes that dominate the network's long-term behavior Krishnamurthy et al. (2022). Complementary dynamical-systems perspectives show that gates induce eigen-directions with eigenvalues clustered near one, enabling long-timescale stability and supporting continuous attractor structures within the hidden-state space Can et al. (2020). Smith et al. (2021) propose a reverse-engineering framework based on Jacobian switching linear dynamical systems, showing that trained RNNs can be locally approximated by low-dimensional linear dynamics controlled by a small set of dominant eigen-directions. Importantly, many of these directions evolve slowly, revealing that long-horizon behavior in RNNs is governed by a compact set of stable dynamical modes. In parallel, Miller & Hardt analyze stability properties of recurrent models and demonstrate that trained RNNs often operate near nearly-identity transformations. Such near-identity dynamics naturally give rise to slowly changing hidden-state dimensions, providing an inherent mechanism for maintaining semantically meaningful information over time. Taken together, these findings establish that gated RNNs reliably form persistent, slowly evolving subspaces—a phenomenon directly aligned with our use of GRU update-gate behavior to characterize slow modes.

**Empirical Analysis for GRU Dynamics of Slot Attention**  Because our ablation study suggested that removing the GRU disrupts this homogeneity, we further investigate whether the recurrent up-

dates are responsible for its stability. We adopt the theoretical perspective that the gating mechanisms in RNNs, such as the GRU, effectively generate "slow modes"—hidden-state dimensions whose update gate ($z_t$) remains consistently small and thus changes minimally across iterations. These slow modes act as stable structural channels for preserving key semantic information over long horizons. In this view, the theory establishes that a "slow mode" corresponds to a hidden-state dimension whose update gate ($z_t$) remains consistently small, as denoted by the final update (Equation 3):

$$h_t = (1 - z_t) \odot h_{t-1} + z_t \odot \tilde{h}_t. \tag{3}$$

We hypothesize that this stability is the theoretical basis for the emergence of stable, factor-specific representations. We leverage this stability property by using the GRU update gate $z_t$ as an empirical measure of dimensional stability (a "slow mode rank"). We then apply this stability measure to analyze factor-wise homogeneity by correlating these 'slow mode ranks' across slots to confirm that slots sharing identical semantic factors rely on the same highly stable (slow mode) dimensions.

To assess consistency between semantic factors, we analyze the slot representation of the $i$-th object, denoted as $o^i_{s,c,p}$ as in Figure 2 (where $s$ is shape, $c$ is color, and $p$ is position). We collect GRU update-gate activations $z^i_{t,d}$ (Equation 3) for each slot dimension $d \in \mathbb{R}^D$ and compute a slow mode rank vector $r^i \in \mathbb{R}^D$. We then compute the Spearman's rank correlation (Equation 4) to analyze whether the slow modes of each dimension show consistency of the ordering (rank) of these stable dimensions across slots with respect to factors:

$$\rho = \text{Spearman}\left(r^i, r^j\right) \tag{4}$$

**GRU Slow Mode Consistency Across Semantic Factors**   We first seek to validate if the stability of the GRU's slow mode dimensions directly corresponds to semantic factors and remains highly consistent across slots sharing identical factors. This test confirms the meaningfulness of our theoretical approach in explaining the observed structural robustness. For object in intra-tasks where evaluation is done on models that is trained on identical task (e.g. E0/T0, E1/T1), we compute correlations between (1) object pairs with identical factors and (2) object pairs with different factors.

| Evaluation Setting ($E_i/T_j$) | Objects with identical factors ($\rho$) | Objects with different factors ($\rho$) |
| --- | --- | --- |
| E0/T0 | 0.9921 | 0.7931 |
| E1/T1 | 0.9920 | 0.7958 |

Table 3: Spearman's rank correlation ($\rho$) of GRU slow mode ranks for objects within the same task. The high correlation for identical factors confirms the consistency of factor-specific stable dimensions.

As shown in Table 3, objects sharing identical factors exhibit a nearly perfect correlation ($\rho \approx 0.99$), showing that slow-mode GRU dimensions are highly consistent across slots with the same semantics. Conversely, although correlations for different-factor pairs are numerically substantial (e.g., $\rho \approx 0.79$), the consistent and significant gap indicates the presence of distinctive factor-specific slow-mode dimensions. These findings directly align with the geometric observation in Figure 2, confirming that the GRU effectively performs an identical-mapping operation conditioned on each object's intrinsic variability, stabilizing slots that share the same semantics along these consistent slow-mode dimensions.

**Preservation of Slow Mode Structure Under Continual Learning**   Finally, we examine whether the observed structural robustness persists under continual settings, which is essential for maintaining task separation. The primary purpose of this analysis is to test whether the GRU slow modes

maintain the representation quality (intra-task retention of previous task) and remain consistent during continual settings (inter-task separation between previous task and novel task). We compute correlations for (1) samples from the previous task (intra-task retention) and (2) inter-samples across previous and novel tasks (inter-task separation). We compare these results with the earlier correlations (Table 3), treating them as the baseline under standard (non-continual) conditions.

| Evaluation Setting (E$i$ / T$j$) | Objects with identical factors ($\rho$) | Objects with different factors ($\rho$) |
|:---:|:---:|:---:|
| E0/T1 | 0.9935 | 0.8054 |
| E0/T1 VS E1/T1 | - | 0.7929 |

Table 4: Spearman's rank correlation ($\rho$) of GRU slow mode ranks under the continual learning setting. Note that *shape* factors for all sequential tasks are disjoint, leading none of objects with identical factors for inter-tasks (bottom row).

Results in Table 4 strongly support the preservation of factor-wise homogeneity through the GRU dynamics: Intra-Task Retention: The intra-task correlation for the previous task (E0/T1) remains high and consistent with the baseline, indicating that the GRU performs an identical-mapping operation even for previously seen samples. Inter-Task Separation: Crucially, the inter-task correlation (E0/T1 VS E1/T1) closely matches the correlation observed for objects with different semantic factors under the standard condition. This demonstrates that both tasks retain their distinctive factor-specific slow-mode dimensions, confirming that structural factor-wise separation is robustly maintained under continual settings. These consistency demonstrates that the GRU dynamics provide a robust mechanism for task-level separation by stabilizing factor-specific slow modes, which is the strong inductive bias for continual object-centric learning.

**Summary: GRU Dynamics and Structural Robustness**   Motivated by the degradation observed in our ablation study, this section aimed to analyze the emergence of factor-wise homogeneity and task-level separation in continual learning through the lens of GRU slow modes. We applied the slow mode rank, derived from the GRU update gate, as an empirical measure of dimensional stability. Our analysis demonstrated two key findings: (1) Slots sharing identical semantic factors exhibit a nearly perfect correlation ($\rho \approx 0.99$) in their slow mode ranks and distinctive gap between correlation among slots with different factors (Table 3), confirming that the GRU stabilizes shared semantic factors along consistent dimensions. (2) Under the continual learning setting, the inter-task correlation between old and new tasks remains low (Table 4), robustly maintaining the task separation observed in the single-task baseline.

Our analysis shows that the GRU's marginally stable slow modes effectively perform an identical-mapping operation conditioned on each object's intrinsic variability, stabilizing slots that share the same semantic factors. This slow-mode mechanism successfully explains both the robustness of factor-wise homogeneity and the natural task separation observed in Slot Attention under continual learning. This consistency demonstrates that the GRU dynamics provide a robust mechanism for task-level separation by stabilizing factor-specific slow modes, which is the necessary structural foundation for effective continual object-centric learning.

### A.4 UNIQUENESS OF FACTOR-WISE HOMOGENEITY PROPERTY TO SLOT ATTENTION

In this section, we validate whether our findings of Slot Attention is a unique property of its own or common property on object-centric models. In this section, to validate the uniqueness of the Factor-Wise Homogeneity property of Slot Attention, we conduct a comparative analysis with a representative non-Slot Attention OCL model, MONet (Multi-Object Network) Burgess et al. (2019). The primary goal of this analysis is to determine if this desirable property is an inherent characteristic of general OCL architectures or if it is specific to the Slot Attention mechanism's unique iterative attention scheme. We reproduced the MONet architecture and evaluated its latent space using the same qualitative (t-SNE visualization) and quantitative (inter-task class similarity and nearest neighbor purity) metrics established in Section 4.2. We follow detailed hyper-parameters from Burgess et al. (2019); Dittadi et al. (2021) for MONet reproduce.

**T-SNE visualization of latent representations** We first evaluate the distribution of each models latent representations (slots) via t-SNE. Figure 12 shows the t-SNE visualization of slot representations using the C-Tetrominos from three different models: (1) Slot Attention, (2) SlotMLP, (3) MONet. As shown in Figure 12, MONet does not exhibit the latent separation observed in Slot Attention. When representations are labeled by *task pairs*, a large portion of representations from different tasks overlap, making task-wise separation difficult. A similar pattern appears in Figure 13, when labeled with *semantic pairs*: MONet shows little to no generalization for the *shape* factor and only mild alignment for *color* and *position*, which remains insufficient to consider the representations well separated compared to Slot Attention.

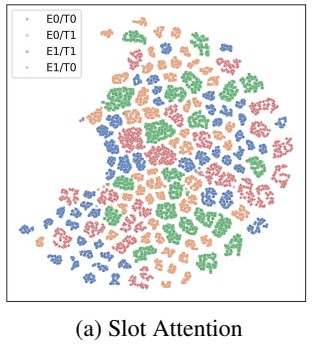 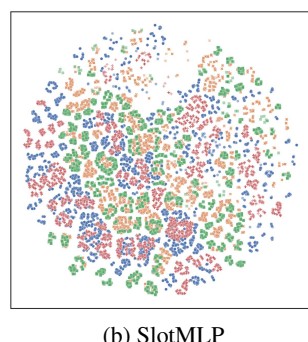 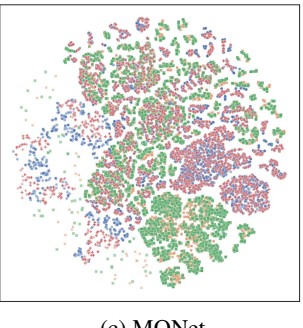

(a) Slot Attention           (b) SlotMLP           (c) MONet

Figure 12: Inter-task separation visualization of (a) Slot Attention, (b) SlotMLP, (c) MONet via t-SNE.

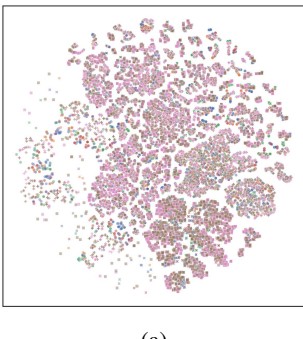 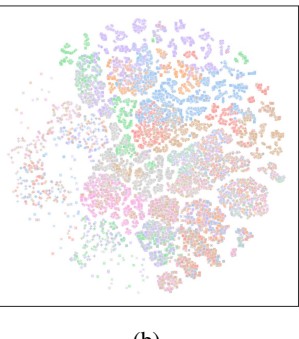 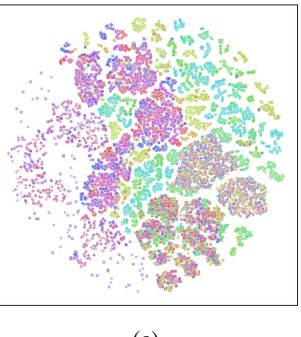

(a)                 (b)                 (c)

Figure 13: t-SNE visualization results of slots from MONet. Each color of the dots represent (a) *shape*, (b) *position*, (c) *color*.

**Quantitative analysis of Inter-Task Separation**   We also evaluated MONet's inter-task separation following the quantitative analysis in Section 4.2 (i.e., inter-task class similarity and nearest neighbor class purity). The results in Figure 14 further confirm that MONet quantitatively lacks factor-wise homogeneity in its latent space. Specifically, unlike Slot Attention which yields lower density of high-similarity neighbors (indicating clearer task-level distinctions) and maintains higher nearest neighbor class purity, MONet shows high cross-task similarity and fails to maintain localized representations.

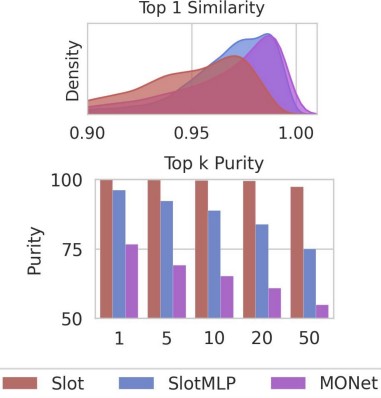

Figure 14: Quantitative evaluation of inter-task separation using (top) Top-$k$ Inter-Task Class Similarity and (bottom) Top-$k$ Nearest Neighbor Class Purity. "Slot" denotes Slot Attention, which exhibits well-separated features across tasks.

In summary, the results from both the qualitative t-SNE visualizations and the quantitative inter-task separation metrics demonstrates that factor-wise homogeneity is not a universal feature of Object-Centric Learning models. Instead, this desirable property—which is crucial for enabling effective continual learning—is uniquely attributable to the Slot Attention architecture.

### A.5 CAUSAL LINK BETWEEN FACTOR-WISE HOMOGENEITY AND DPR EFFECTIVENESS

While Section 4 and Appendix A.4 demonstrate the existence of the unique factor-wise homogeneity property in Slot Attention, a critical question remains: **is this property a direct prerequisite for the effectiveness of our DPR strategy?** To verify the causal link between structural alignment in the latent space and DPR's ability, we conducted comparative experiments: applying DPR (1) where latent representations (slots) do not exhibit factor-wise homogeneity property (e.g., SlotMLP, MONet), and (2) when the representations of slots are disrupted to diminish this property (ablating the critical component GRU found in ablation study).

**Evaluation on various models with and without DPR** We follow the same evaluation setting as in Section 5.2, utilizing 5 independent runs on the C-Tetrominoes SST dataset. Table 5 reports the FG-ARI measure on the previous task trained on the novel task (E0/T1) to identify whether DPR is effective where factor-wise homogeneity does not exist.

| Model | FG-ARI$_{\pm\text{std.}}$ (E0 / T1) | |
|---|---|---|
| | Without DPR | With DPR |
| SlotMLP | $32.85_{\pm 0.04}$ | $45.75_{\pm 0.10}$ |
| MONet | $43.51_{\pm 0.06}$ | $44.41_{\pm 0.04}$ |
| Slot w/o GRU | $35.83_{\pm 0.04}$ | $57.54_{\pm 0.02}$ |
| Slot Attention | $41.39_{\pm 0.03}$ | $\mathbf{96.83}_{\pm 0.01}$ |

Table 5: FG-ARI results of various models with and without DPR on C-Tetrominoes evaluated on the previous task (E0) after training on the novel task (T1). We report average of 5 runs.

First, methods lacking homogeneity—namely, SlotMLP and MONet—showed little to no performance gains on the previous task (E0/T1) when trained with DPR, exhibiting a significantly diminished effectiveness compared to the substantial improvement observed with Slot Attention. This confirms that the baseline OCL framework, without the specific structure, cannot utilize DPR effectively.

Second, when we intentionally disrupted Slot Attention's homogeneity by ablating the GRU (thereby removing the critical iterative gating dynamics), the benefits of DPR largely diminished. Specifically, the $FG - ARI$ for the Slot w/o GRU model only improved from $0.3583$ to $0.5754$, representing a modest gain, in sharp contrast to the Slot Attention model, which improved substantially from $0.4139$ to $0.9683$. Finally, the additional observation from Figure 5, where DPR is applied to a non-OCL model with a reconstruction objective (Variational Autoencoder (VAE)), also fails to benefit from DPR. This failure, further illustrated by the lack of separation in Figure 15, isolates the importance of the factorized, object-centric structure.

Collectively, these results confirm that DPR is effective precisely because Slot Attention provides a well-separated and consistently aligned latent representation space. Without this pre-existing structural alignment, the DPR strategy is unable to effectively improve degradation of previous tasks.

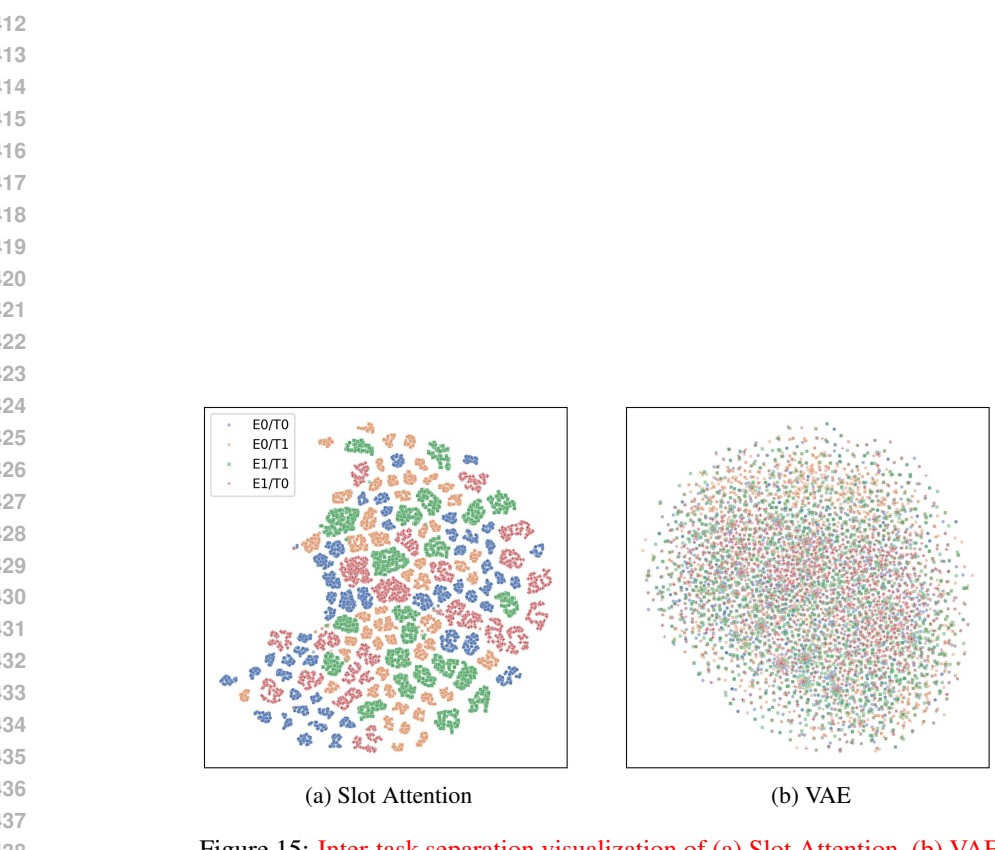

(a) Slot Attention            (b) VAE

Figure 15: Inter-task separation visualization of (a) Slot Attention, (b) VAE

A.6 DETAILED EVALUATION RESULTS OF DECODER ONLY POST REPLAY

Figure 16 and Figure 17 shows task individual performances on Continual Object Centric Learning benchmarks. We report individual evaluation and train task-wise performance using metrics of Adjusted Rand Index (ARI), Mean Squared Error(MSE), mean Segmentation Covering (mSC), and Segmentation Covering (SC) metrics. We report evaluations of three different scenarios, (1) Single Step addition of Two classes (SST), (2) Single Step addition of Multiple classes (SSM), and (3) Multi Step addition of Two classes (MST) per step. For MST, we only show evaluation of $\mathcal{T}_0$.

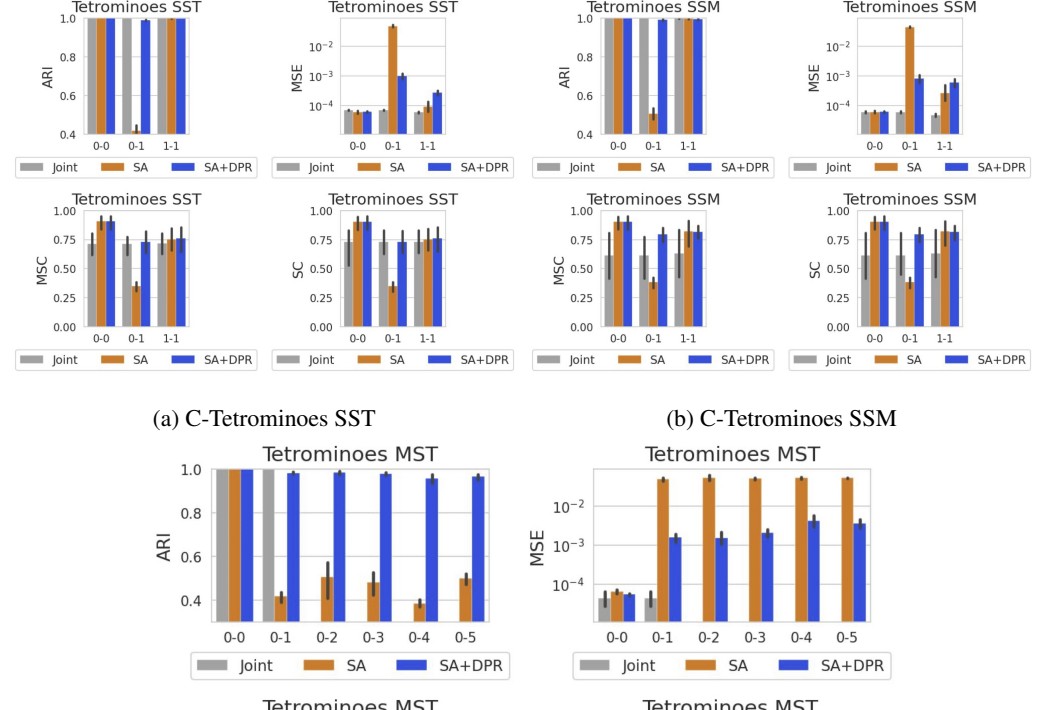

Figure 16: Individual performance on C-Tetrominos.

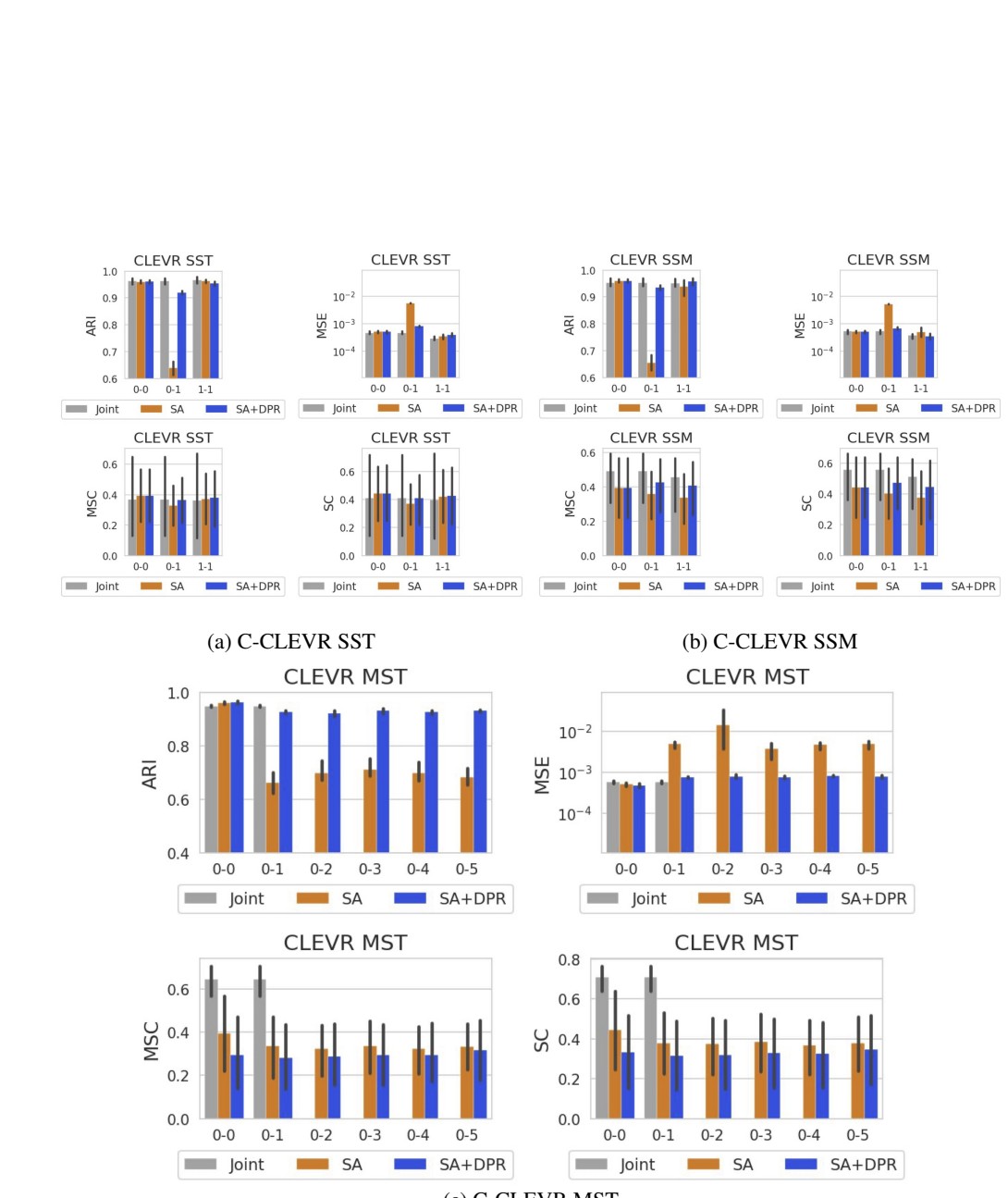

(a) C-CLEVR SST                                        (b) C-CLEVR SSM

(c) C-CLEVR MST

Figure 17: Individual performance on C-CLEVR.

## A.7 Numerical Evaluation Results of Decoder only Post Replay

To ensure that the performance improvements observed with DPR are not incidental but statistically reliable, we additionally report results averaged over five independent runs, including mean and standard deviation. The numerical results are summarized in Table 6–8 and Table 9–11. For MST evaluation, we report results after training the final task.

| Model | E0/T0 | E0/T1 | E1/T1 |
|---|---|---|---|
| Joint | 0.9989 $_{\pm 0.00}$ | | 0.9985 $_{\pm 0.00}$ |
| SA$^\dagger$ | 0.9986 $_{\pm 0.00}$ | 0.4171 $_{\pm 0.03}$ | 0.9979 $_{\pm 0.00}$ |
| SA$^\dagger$ + DPR | 0.9988 $_{\pm 0.00}$ | 0.9881 $_{\pm 0.00}$ | 0.9970 $_{\pm 0.00}$ |

Table 6: FG-ARI($_{\pm std.}$) performance on C-Tetrominoes SST evaluated across previous tasks $\mathcal{T}_i$ (E$i$) after training on novel task $\mathcal{T}_j$ (T$j$). SA$^\dagger$ denotes Slot Attention.

| Model | E0/T0 | E0/T1 | E1/T1 |
|---|---|---|---|
| Joint | 0.9986 $_{\pm 0.00}$ | | 0.9982 $_{\pm 0.00}$ |
| SA$^\dagger$ | 0.9986 $_{\pm 0.00}$ | 0.5044 $_{\pm 0.04}$ | 0.9966 $_{\pm 0.00}$ |
| SA$^\dagger$ + DPR | 0.9986 $_{\pm 0.00}$ | 0.9910 $_{\pm 0.01}$ | 0.9942 $_{\pm 0.00}$ |

Table 7: FG-ARI($_{\pm std.}$) performance on C-Tetrominoes SSM evaluated across previous tasks $\mathcal{T}_i$ (E$i$) after training on novel task $\mathcal{T}_j$ (T$j$). SA$^\dagger$ denotes Slot Attention.

| Model | E0/T5 | E1/T5 | E2/T5 | E3/T5 | E4/T5 | E5/T5 |
|---|---|---|---|---|---|---|
| Joint | 0.9985 $_{\pm 0.00}$ | 0.9982 $_{\pm 0.00}$ | 0.9985 $_{\pm 0.00}$ | 0.9980 $_{\pm 0.00}$ | 0.9984 $_{\pm 0.00}$ | 0.9983 $_{\pm 0.00}$ |
| SA$^\dagger$ | 0.4980 $_{\pm 0.03}$ | 0.6115 $_{\pm 0.06}$ | 0.4895 $_{\pm 0.09}$ | 0.3693 $_{\pm 0.05}$ | 0.4171 $_{\pm 0.06}$ | 0.9983 $_{\pm 0.00}$ |
| SA$^\dagger$ + DPR | 0.9641 $_{\pm 0.02}$ | 0.9887 $_{\pm 0.01}$ | 0.9848 $_{\pm 0.00}$ | 0.9804 $_{\pm 0.01}$ | 0.9793 $_{\pm 0.01}$ | 0.9910 $_{\pm 0.01}$ |

Table 8: FG-ARI($_{\pm std.}$) performance on C-Tetrominoes MST evaluated across previous tasks $\mathcal{T}_i$ (E$i$) after training on novel task $\mathcal{T}_j$ (T$j$). SA$^\dagger$ denotes Slot Attention.

| Model | E0/T0 | E0/T1 | E1/T1 |
|---|---|---|---|
| Joint | 0.9624 $_{\pm 0.01}$ | | 0.9666 $_{\pm 0.01}$ |
| SA$^\dagger$ | 0.9601 $_{\pm 0.01}$ | 0.6383 $_{\pm 0.03}$ | 0.9621 $_{\pm 0.01}$ |
| SA$^\dagger$ + DPR | 0.9603 $_{\pm 0.01}$ | 0.9200 $_{\pm 0.01}$ | 0.9536 $_{\pm 0.01}$ |

Table 9: FG-ARI($_{\pm std.}$) performance on C-CLEVR SST evaluated across previous tasks $\mathcal{T}_i$ (E$i$) after training on novel task $\mathcal{T}_j$ (T$j$). SA$^\dagger$ denotes Slot Attention.

| Model | E0/T0 | E0/T1 | E1/T1 |
|---|---|---|---|
| Joint | $0.9519_{\pm0.02}$ | | $0.9511_{\pm0.02}$ |
| SA$^{\dagger}$ | $0.9602_{\pm0.01}$ | $0.6554_{\pm0.04}$ | $0.9381_{\pm0.04}$ |
| SA$^{\dagger}$ + DPR | $0.9602_{\pm0.01}$ | $0.9355_{\pm0.01}$ | $0.9581_{\pm0.02}$ |

Table 10: FG-ARI$(_{\pm std.})$ performance on C-CLEVR MST evaluated across previous tasks $\mathcal{T}_i$ (E$i$) after training on novel task $\mathcal{T}_j$ (T$j$). SA$^{\dagger}$ denotes Slot Attention.

| Model | E0/T5 | E1/T5 | E2/T5 | E3/T5 | E4/T5 | E5/T5 |
|---|---|---|---|---|---|---|
| Joint | $0.9390_{\pm0.00}$ | | $0.9408_{\pm0.00}$ | $0.9185_{\pm0.00}$ | $0.9333_{\pm0.01}$ | $0.9333_{\pm0.01}$ |
| SA$^{\dagger}$ | $0.6812_{\pm0.04}$ | $0.6574_{\pm0.04}$ | $0.6662_{\pm0.05}$ | $0.6612_{\pm0.04}$ | $0.6911_{\pm0.04}$ | $0.9530_{\pm0.01}$ |
| SA$^{\dagger}$ + DPR | $0.9305_{\pm0.01}$ | $0.8686_{\pm0.01}$ | $0.9218_{\pm0.00}$ | $0.9454_{\pm0.01}$ | $0.8896_{\pm0.02}$ | $0.9507_{\pm0.01}$ |

Table 11: FG-ARI$(_{\pm std.})$ performance on C-CLEVR MST evaluated across previous tasks $\mathcal{T}_i$ (E$i$) after training on novel task $\mathcal{T}_j$ (T$j$). SA$^{\dagger}$ denotes Slot Attention.

## A.8 RECONSTRUCTION RESULTS OF DECODER ONLY POST REPLAY

In this section, we reconstruction results of DPR using three different training scenarios (SST, SSM, MST) compared with vanilla Slot Attention.

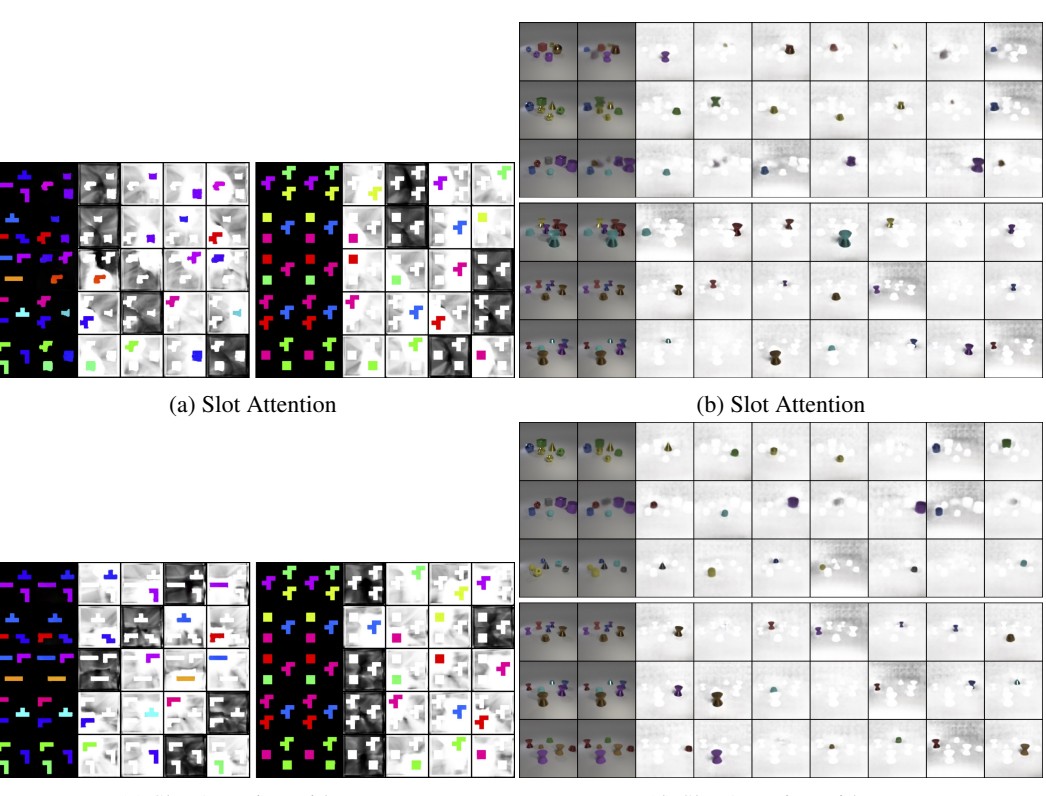

(a) Slot Attention

(b) Slot Attention

(c) Slot Attention with DPR

(d) Slot Attention with DPR

Figure 18: Reconstruction results of Slot Attention using with (a,b) and without DPR (c,d) on C-Tetrominoes SST, and C-CLEVR SST. We used the model after continuously training the last task ($\mathcal{T}_1$). Figures on left (top) are images of $\mathcal{T}_0$, and figures on right (bottom) are images of $\mathcal{T}_1$

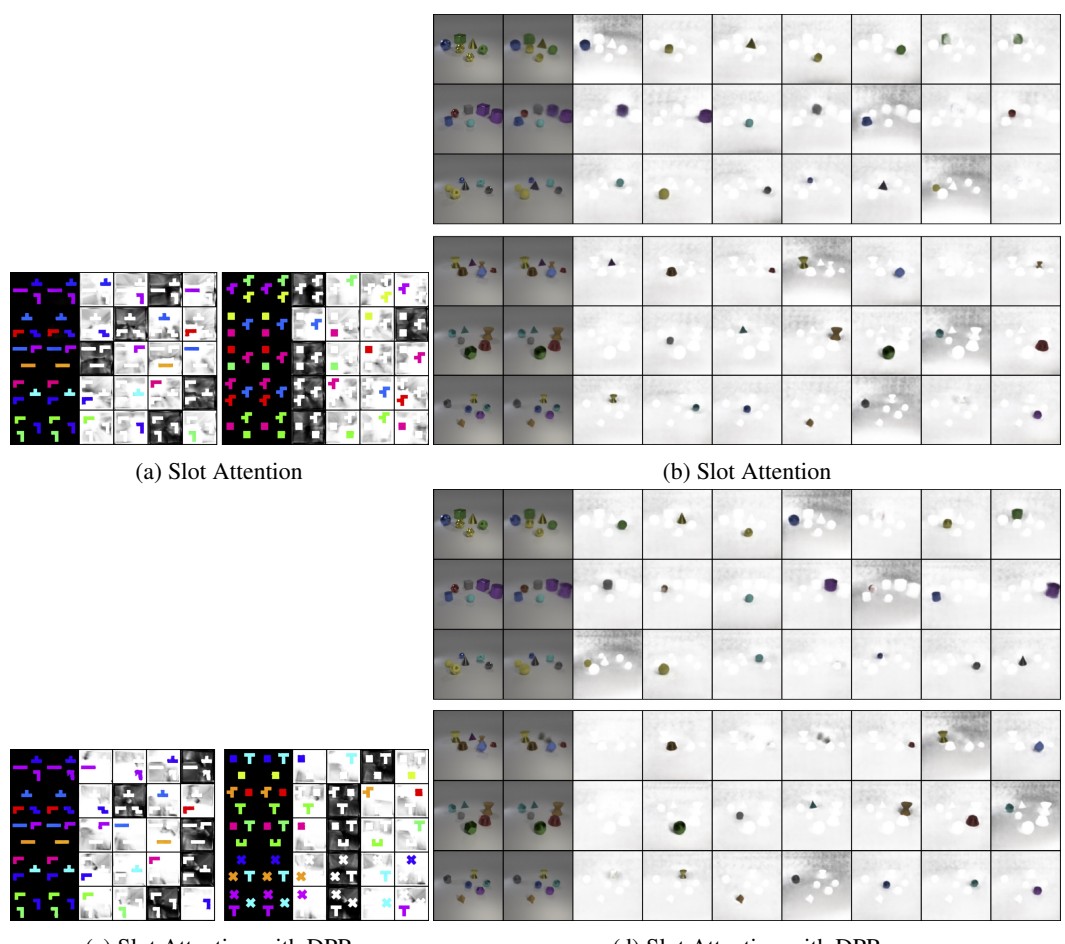

(a) Slot Attention

(b) Slot Attention

(c) Slot Attention with DPR

(d) Slot Attention with DPR

Figure 19: Reconstruction results of Slot Attention using with (a,b) and without DPR (c,d) on C-Tetrominoes SSM, and C-CLEVR SSM. We used the model after continuously training the last task ($\mathcal{T}_1$). Figures on left (top) are images of $\mathcal{T}_0$, and figures on right (bottom) are images of $\mathcal{T}_1$

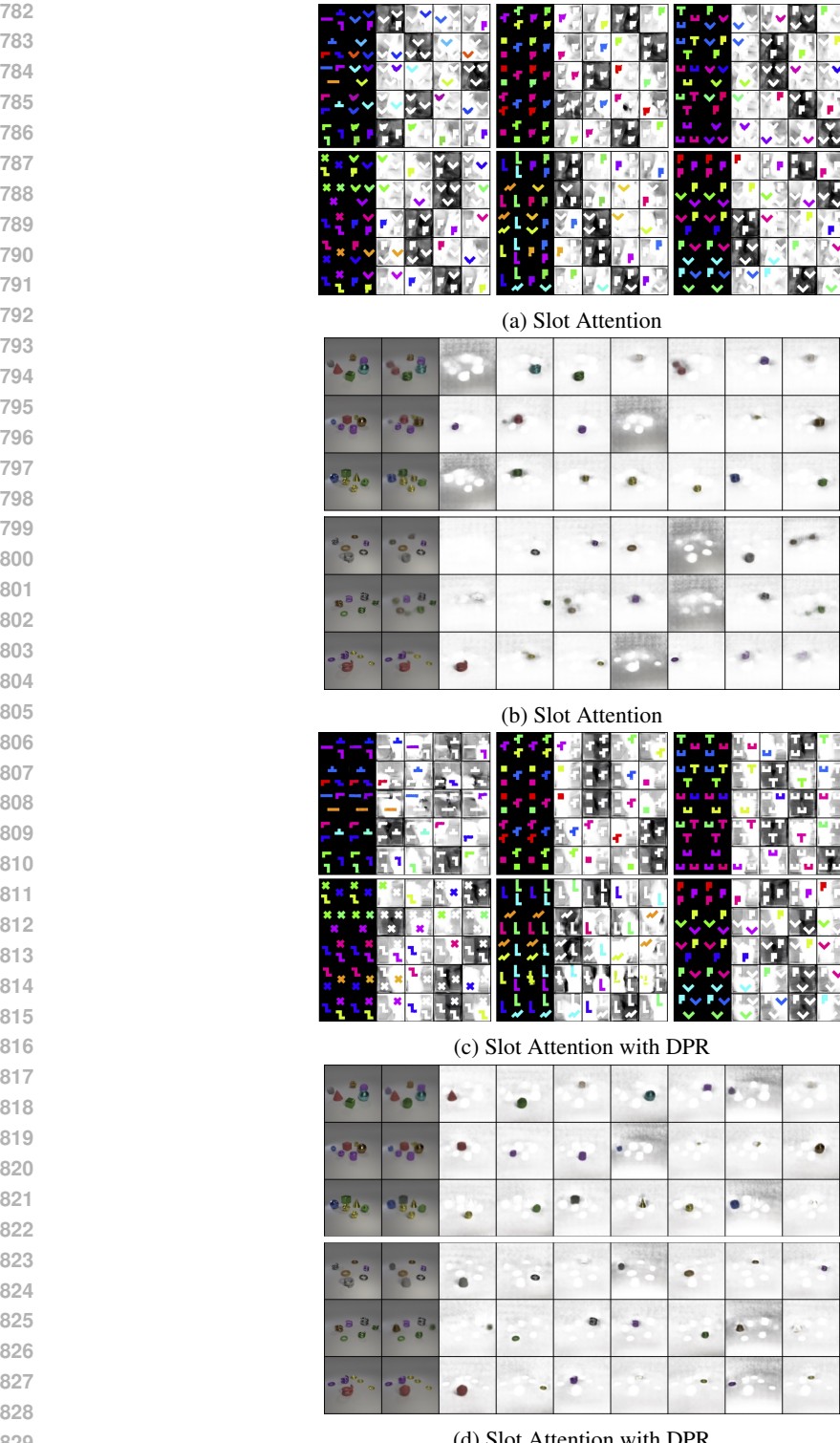

(a) Slot Attention

(b) Slot Attention

(c) Slot Attention with DPR

(d) Slot Attention with DPR

Figure 20: Reconstruction results of Slot Attention using with (a,b) and without DPR (c,d) on C-Tetrominoes MST, and CLEVR MST. We used the model after continuously training the last task ($\mathcal{T}_5$). Figures are in order of sequential tasks from $\mathcal{T}_0$ to $\mathcal{T}_5$ for Tetrominoes, and we only report results of initial $\mathcal{T}_0$ and last $\mathcal{T}_5$ task for CLEVR.

### A.9 COMPARISON WITH NON-OBJECT-CENTRIC METHOD

To evaluate the effectiveness of factor-wise homogeneity property of Slot Attention, we compare post-replay method with non-object-centric method: Variational Auto Encoder (VAE) Kingma et al. (2013) as baseline. Architectural implementation of VAE is discussed at Table 23. We apply Experience Replay, Post Replay, and Decoder only Post Replay to VAE using three different training scenarios (SST, SSM, MST). Figure 21 shows the comparisons VAE with various replay methods.

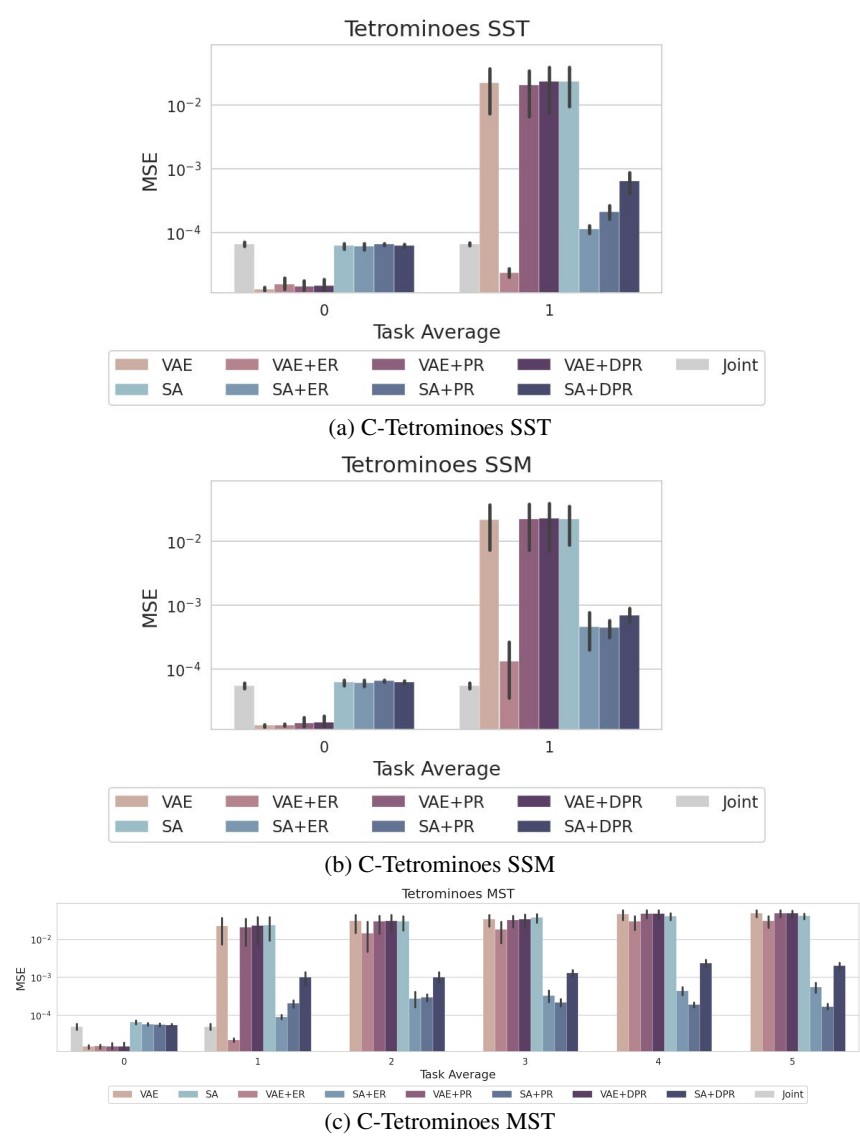

(a) C-Tetrominoes SST

(b) C-Tetrominoes SSM

(c) C-Tetrominoes MST

Figure 21: Comparison with non-object-centric method (VAE).

### A.10 COMPARISON WITH RELAY-BASED VARIATIONS

In this section, we provide comparisons of DPR with existing replay-based approaches using three different training scenarios (SST, SSM, MST) compared with baselines. As a baseline, we apply standard Experience Replay to Slot Attention (SA+ER). Following prior work Lopez-Paz & Ranzato (2017a); Rebuffi et al. (2017); Chaudhry et al. (2019); Buzzega et al. (2020a), we perform *joint training* by concatenating replay samples with the current task's training data (Appendix B.4).

In addition, we evaluate a generative variant (SA+GR) motivated by Generative Replay methods Shin et al. (2017); Wu et al. (2018); Ayub & Wagner; Zhai et al. (2019); Gao & Liu (2023), which utilize an auxiliary generative model to produce replay samples. For each new task, SA+ER performs joint training with generated samples. We adopt a Variational Autoencoder (VAE) Kingma et al. (2013) as the generative model, following the setup in Shin et al. (2017).

Finally, we report results of using Post Replay (SA+PR) and Decoder only Experience Replay (SA+DRwT). SA+PR fine-tunes the entire parameters of the model after the main training phase of task $\mathcal{T}_t$ using replay samples from the buffer. SA+DRwT freezes both the encoder $\mathcal{F}$ and Slot Attention module $\mathcal{SA}$ after training on $\mathcal{T}_0$, and train only the decoder $\mathcal{G}$ on replay samples from subsequent tasks. Figure 22 shows comparisons with replay-based baselines.

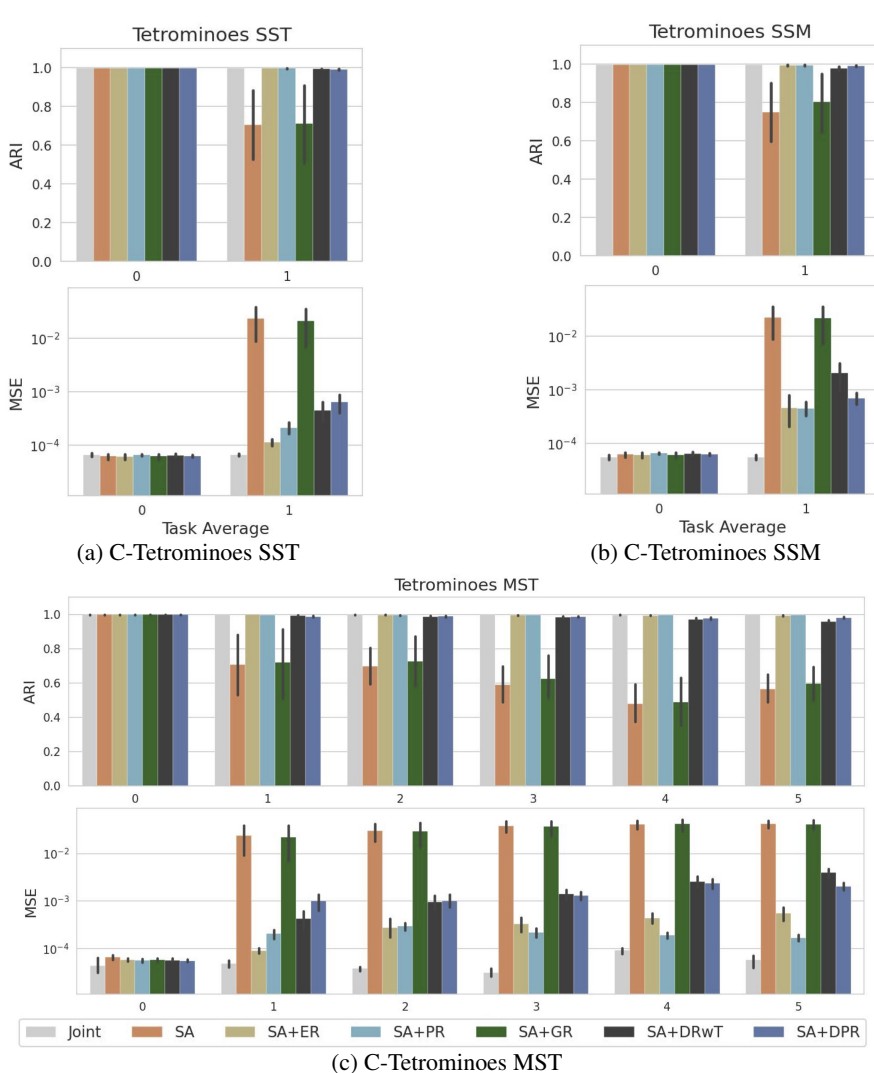

(a) C-Tetrominoes SST    (b) C-Tetrominoes SSM

(c) C-Tetrominoes MST

Figure 22: Comparison with previous replay methods.

### A.11 COMPARISON WITH REGULARIZER-BASED METHODS

In this section, we provide comparisons with prior regularization based method using three different training scenarios (SST, SSM, MST) compared with baselines. We evaluate DPR in comparison with Slot Attention combined with Learning without Forgetting (LwF) Li & Hoiem (2017) (SA+LWF) and Elastic Weight Consolidation (EWC) Kirkpatrick et al. (2017) (SA+EWC). Implementation of SA+LWF is discussed at Appendix B.4, and SA+EWC is demonstrated in Appendix B.4.

Figure 23 shows comparisons with prior regularization based method. Compared to DPR, SA+LwF and SA+EWC are less effective when applied to Slot Attention, as they continue to suffer performance degradation after training on new tasks. Interestingly, combining SA+EWC or SA+LwF with DPR yields slightly improved performance on MSE metric over using DPR alone. Specially, simultaneously using both DRP and SA+EWC showed the best performance among them, suggesting that DPR is robust to integration with regularization-based continual learning methods.

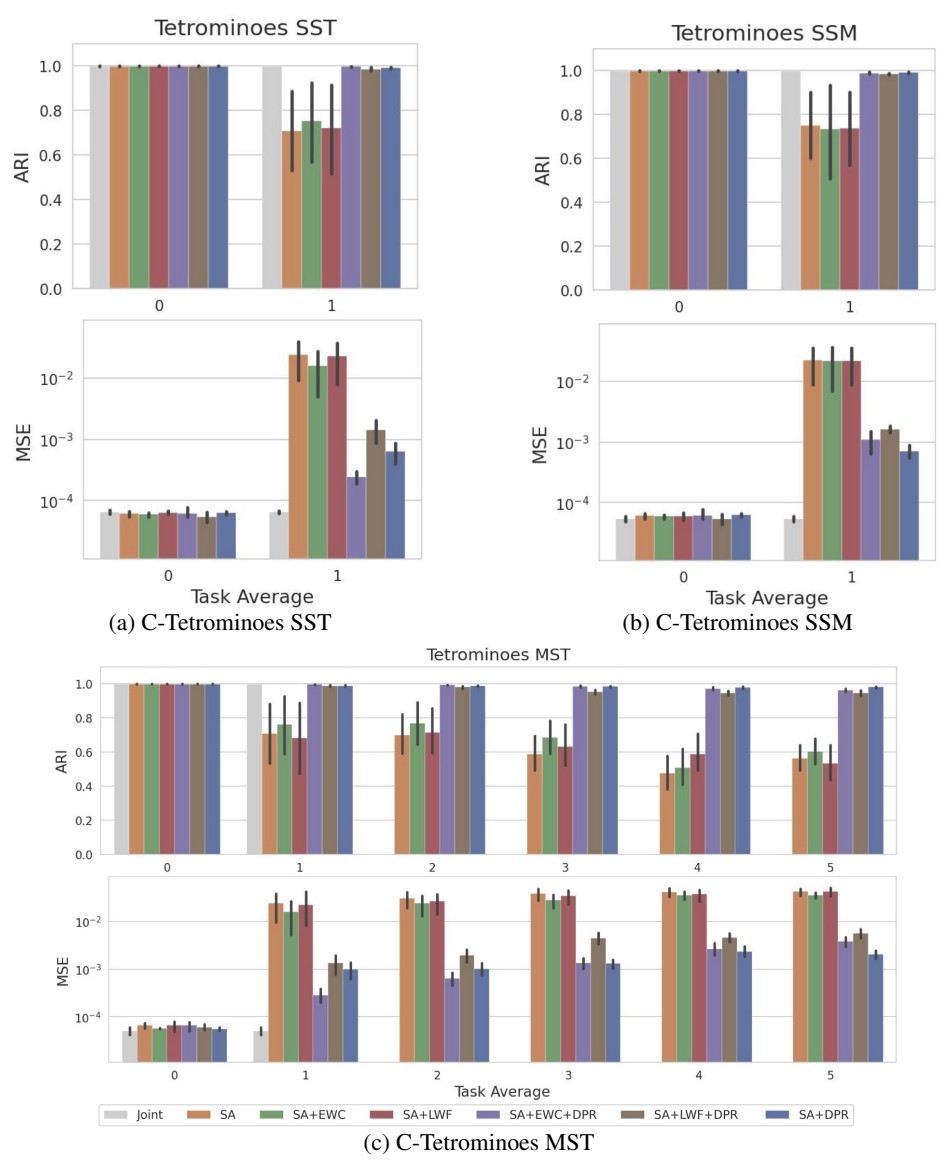

(a) C-Tetrominoes SST           (b) C-Tetrominoes SSM

(c) C-Tetrominoes MST

Figure 23: Comparison with prior regularizer-based methods.

## A.12 PERFORMANCE ON MORE CHALLENGING SETTINGS

In this section, we aim to evaluate whether the proposed *factor-wise homogeneity* and DPR extends to challenging real-world environments. We first describe the dataset configurations used for continual learning benchmarks (COCO and PASCAL), then present the baseline setting with DINOSAUR as a representative slot-based model for complex images. Finally, we report and analyze the experimental results to demonstrate that our inductive bias remains effective beyond synthetic domains.

**Datasets and evaluation settings**  We extend our evaluation of *factor-wise homogeneity* to more challenging environments to assess whether this property persists beyond simplified synthetic benchmarks. To systematically increase task complexity, we consider two complementary axes: (i) real-world datasets, which introduce diverse and uncontrolled visual variations, and (ii) complex synthetic datasets, which exhibit high variability in texture and background.

For the real-world setting, we adopt two widely used real-world benchmarks in OCL Seitzer et al.; Didolkar et al. (2025); Kakogeorgiou et al. (2024): MS COCO 2017, which contains multiple objects with diverse labels, and PASCAL VOC 2012, which primarily consists of single labeled objects per image. Since COCO and PASCAL are originally defined as single-task datasets, we follow prior works Shmelkov et al. (2017); Michieli & Zanuttigh (2019); Cermelli et al. (2020) and split each dataset into $|\mathcal{T}|$ disjoint tasks according to object categories. We consider two continual learning scenarios: (i) Single-Step addition of Multiple classes (SSM), and (ii) Multi-Step addition of Two classes (MST). We set $|\mathcal{T}| = 2$ for SSM and $|\mathcal{T}| = 4$ for MST, ensuring that each task includes the same number of object categories, divided alphabetically. Following the disjoint settings of prior works, we retain only images whose object labels do not overlap across tasks. Table 12 summarizes the dataset configurations used in our evaluation.

| Dataset | Scenario | $|\mathcal{T}|$ | Objects per Task | Total Objects |
|---------|----------|-----|-----------------|---------------|
| COCO    | SSM      | 2   | 40              | 80            |
|         | MST      | 4   | 20              | 80            |
| PASCAL  | SSM      | 2   | 10              | 20            |
|         | MST      | 4   | 5               | 20            |

Table 12: Dataset configuration for COCO and PASCAL on continual setting

For the complex synthetic setting, we construct a hybrid scenario that combines CLEVRTex Karazija et al., which introduces diverse textures and realistic backgrounds, with Continual-CLEVR. Specifically, we train the model on the initial task $\mathcal{T}_0$ of Continual-CLEVR and subsequently introduce CLEVRTex as the second task. To ensure a fair comparison, we sample the same number of images from CLEVRTex as in $\mathcal{T}_0$, restricting to scenes with at most six objects. This setup allows us to evaluate whether *factor-wise homogeneity* remains effective when transferring from a relatively simple synthetic environment to one with significantly richer visual complexity.

**Baseline Setting**  To evaluate on challenging real-world images, we adopt COCO Lin et al. (2014) and PASCAL VOC Everingham et al. (2010), two widely used benchmarks in OCL. Since Slot Attention does not scale well to complex synthetic or real-world data Seitzer et al., we employ DINOSAUR Seitzer et al. as our baseline — a slot-based model integrated with pre-trained DINO Caron et al. (2021) that surpasses vanilla Slot Attention and its variants on real-world images. DINOSAUR follows an autoencoder framework similar to vanilla Slot Attention, but instead of reconstructing the input image, it reconstructs latent patch features $\boldsymbol{h}_i \in \mathbf{R}^{N \times D}$ from a target encoder (e.g., pre-trained DINO). This enables the slots to capture more high-level semantics, leading to significant improvements on complex images.

As a baseline configuration, we use ImageNet Deng et al. (2009) pre-trained DINO-Base/16 as both the feature extractor and target encoder, following the setup in Seitzer et al.. We train the network with Adam Kingma & Ba (2014) using a learning rate of $4 \times 10^{-4}$, a warm-up phase, and an exponential decay schedule. We adopt a global batch size of 64 and set 7 slots for COCO and 6 slots for PASCAL. We adopt a global batch size of 64 and set 7 slots for COCO and 6 slots for PASCAL. The reproduced performance of DINOSAUR is reported in Table 13, and we use the

| Dataset | Model | FG-ARI | $\text{mBO}^c$ | $\text{mBO}^i$ |
|---|---|---|---|---|
| COCO | DINOSAUR | $32.37_{\pm 0.8}$ | $42.94_{\pm 0.7}$ | $31.25_{\pm 0.7}$ |
| PASCAL | DINOSAUR | $23.63_{\pm 0.7}$ | $52.96._{\pm 0.8}$ | $46.29_{\pm 0.7}$ |

Table 13: Reproduced performance of DINOSAUR on real-world COCO and PASCAL VOC dataset (5 runs, $\text{mean}_{\pm \text{std}}$).

same configurations for all following experiments using DINOSAUR. Results are evaluated using Foreground Adjusted Rand Index and mean Best Overlap Pont-Tuset et al. (2016), averaged over 5 independent runs.

Since DINOSAUR reconstructs latent patch features from the target encoder rather than pixel-level inputs as in vanilla Slot Attention, it does not directly output image reconstructions. To enable decoder fine-tuning in the DPR phase, we store a subset of randomly sampled input images as a replay buffer. We use the same buffer size as in the synthetic benchmarks.

For CLEVRTex, we follow the same configuration as in CLEVR (Table 22) and adopt the hyper-parameters in Table 20, ensuring consistency across synthetic experiments. We report results of ARI and MSE averaged over 5 independent runs.

**Factor-wise Homogeneity in real-world Image**  Table 1 and Table 14 present results on continual learning with real-world datasets, consisting of four and two sequential tasks, respectively. Our method achieves consistent performance improvements across both datasets, indicating that our findings are not limited to simplified synthetic benchmarks but also hold in real-world scenarios. This further supports our main claim that *factor-wise homogeneity* serves as an effective inductive bias for continual object-centric learning.

| Dataset | Model | Evaluation on $\mathcal{T}_0$ / Trained on $\mathcal{T}_0$ | | Evaluation on $\mathcal{T}_0$ / Trained on $\mathcal{T}_1$ | |
|---|---|---|---|---|---|
| | | FG-ARI | $\text{mBO}^c/\text{mBO}^i$ | ARI | $\text{mBO}^c/\text{mBO}^i$ |
| COCO | DINOSAUR DPR$^+$ | $22.84_{\pm 0.8}$ | $42.93_{\pm 0.6}$ / $31.25_{\pm 0.7}$ | $21.02_{\pm 0.1}$ $\mathbf{22.51}_{\pm 0.5}$ | $40.63_{\pm 0.4}$ / $29.04_{\pm 0.4}$ $\mathbf{41.74}_{\pm 0.2}$ / $\mathbf{30.57}_{\pm 0.5}$ |
| PASCAL | DINOSAUR DPR$^\dagger$ | $16.61_{\pm 0.6}$ | $53.47_{\pm 0.5}$ / $47.74_{\pm 0.6}$ | $15.45_{\pm 1.2}$ $\mathbf{16.58}_{\pm 0.9}$ | $51.79_{\pm 1.6}$ / $46.61_{\pm 1.2}$ $\mathbf{52.70}_{\pm 0.7}$ / $\mathbf{47.37}_{\pm 0.5}$ |

Table 14: Evaluation on $|\mathcal{T}| = 4$ tasks of real-world images (5 runs, $\text{mean}_{\pm \text{std}}$). We evaluate on $\mathcal{T}_0$ after training on $\mathcal{T}_0$ and continuously trained to $\mathcal{T}_1$. DPR$^\dagger$ denotes DPR with DINOSAUR.

In DINOSAUR, both the feature extractor and target decoder leverage DINO pre-trained on large-scale external resources (ImageNet), which substantially enhance baseline continual learning performance by implicitly providing essential information for new tasks. While this strong prior can obscure the benefits of inductive biases such as factor-wise homogeneity and DPR, we emphasize that DPR yields significant and consistent gains in settings without such external information, and even with DINOSAUR the improvements remain consistent. Moreover, as the number of continual learning steps increases, DINOSAUR's performance on earlier tasks degrades, widening the gap across tasks and revealing its limitations in retaining prior representations (compared to Table 1). In these cases, our method provides increasingly larger gains compared to the results in Table 14. These findings confirm that *factor-wise homogeneity* and DPR act as effective inductive biases for continual object-centric learning, even under strong pre-trained representations such as DINO.

We also visualize the slot distributions of DINOSAUR trained on continual COCO ($T = 2$), using t-SNE as described in Section 4.2. Representations are collected across different tasks $\mathcal{T}_t$. We label them as *task pairs* (E$i$/T$j$), representing evaluation on task $\mathcal{T}_i$ using a model continuously trained from the initial task up to task $\mathcal{T}_j$. Figure 24 shows slots from different tasks in real-world images preserve *factor-wise homogeneity* and remain well separated across tasks, consistent with result using C-Tetrominoes (Figure 1a). We observe that slots remain distinct between (i) upcoming tasks with unseen objects (e.g., red vs. green or purple) and (ii) previous tasks without overlaps in

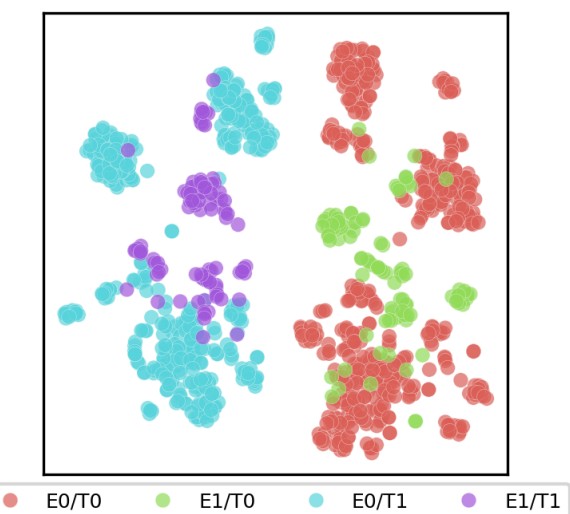

Figure 24: Inter-task separation visualization of Slots from DINOSAUR via t-SNE. We use COCO dataset split into $|\mathcal{T}| = 2$ tasks. Each dot denotes a slot representation from different environments, where evaluation tasks (E$i$) are obtained after continuous training up to task (T$j$).

continual training (e.g., blue vs. purple). Moreover, slots corresponding to the same evaluation task but from different training phases also remain separated.

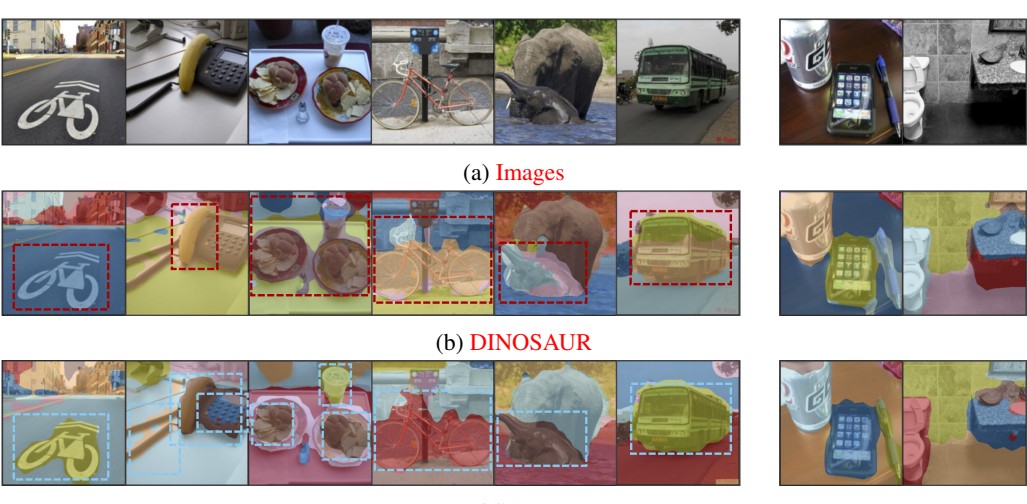

Figure 25: Example results on COCO dataset with $|\mathcal{T}| = 4$ continual tasks. First six images belong to previous tasks ($\mathcal{T}_{0\sim2}$) after training in novel task ($\mathcal{T}_3$), and last two images are from evaluation set of novel task ($\mathcal{T}_3$). Red bounding boxes highlights the failed points of DINOSAUR and blue bounding boxes highlights modifications after DPR. (1) DINOSAUR fails to mask objects that appeared in previous tasks (e.g., a "bicycle", "banana", "sandwich"), while DPR successfully recovers and masks these objects. (2) Also DPR improves to produces a coherent single-slot mask (e.g., a "bicycle", "elephant", "bus") compared to DINOSAUR. (3) Finally, DPR does not degrade mask quality of novel task (e.g., last two images).

Figure 25 presents qualitative mask-prediction results on the COCO dataset. From these examples, we observe three consistent patterns. (1) In the first three (column-wise) images, DINOSAUR fails to mask objects that appeared in previous tasks but are absent in the current task (e.g., "bicycle", "banana", "sandwich"), whereas DINOSAUR with DPR successfully recovers and masks these objects. In particular, DINOSAUR does not separately mask the "bicycle" or "banana", while DPR

clearly separates them from surrounding regions (e.g., separating the "bicycle" from the road and the "banana" from the telephone and telephone line). We further observe cases where DINOSAUR produces integrated mask of "sandwich" while DPR masked both "sandwich" and "cup" of previous task. (2) In the next three images, DPR yields cleaner object-wise segmentation. For instance, DINOSAUR over-segments the baby "elephant" into multiple slots, whereas DPR produces a coherent single-slot mask. Similar improvements are also observed for the "bicycle" and "bus". (3) In the final two images, which belong to the novel task, DPR does not degrade mask quality; both DINOSAUR and DINOSAUR with DPR produce effective masks for unseen objects.

These results further confirm that *factor-wise homogeneity* helps preserve slot representations against task interference, not only in synthetic images but also in complex real-world data.

**Factor-wise Homogeneity in Complex Synthetic Images**   Table 15 presents the results for $\mathcal{T}_0$ (CLEVR) after initial training on $\mathcal{T}_0$ followed by continual training on $\mathcal{T}_1$ (CLEVRTex). Without DPR, Slot Attention exhibits degradation on the evaluation of the previous task ($\mathcal{T}_0$) when trained continuously on the more complex dataset. In contrast, incorporating DPR substantially improves performance, indicating that *factor-wise homogeneity* together with DPR provides a stable and robust inductive bias in complex synthetic environments, extending beyond simple synthetic settings.

| Model | Evaluation on $\mathcal{T}_0$ / Trained on $\mathcal{T}_0$ | | Evaluation on $\mathcal{T}_0$ / Trained on $\mathcal{T}_1$ | |
| | FG-ARI | MSE | FG-ARI | MSE |
|---|---|---|---|---|
| Slot Attention | $0.97_{\pm 0.01}$ | $0.0004_{\pm 0.0}$ | $0.68_{\pm 0.1}$ | $0.004_{\pm 0.01}$ |
| DPR$^\dagger$ | | | $\mathbf{0.81}_{\pm 0.01}$ | $\mathbf{0.001}_{\pm 0.0}$ |

Table 15: Evaluation on $|\mathcal{T}| = 4$ tasks of complex synthetic images (5 runs, mean$_{\pm \text{std}}$). We evaluate on $\mathcal{T}_0$ after training on $\mathcal{T}_0$ (using CLEVR) and continuously trained to $\mathcal{T}_1$ (using CLEVRTex). DPR$^\dagger$ denotes DPR with Slot Attention.

### A.13 ANALYSIS ON ADVANCED SLOT ATTENTION VARIANTS

In this section, we analyze the factor-wise homogeneity and inter-task separation phenomenon in advanced variants of Slot Attention. We examine BO-QSA Jia et al., which employs learnable queries for object-specific bindings (used for slot initialization) and a bi-level optimization scheme for stable training.

**Experiment settings**    We use C-Tetrominoes to analyze the factor-wise homogeneity and inter-task separation phenomenon in advanced variants. BO-QSA Jia et al. uses two different types of decoders: (1) mixture-based decoders from Locatello et al. (2020), (2) transformer-based decoders from Singh et al. (a). Experiments from Jia et al. demonstrates that mixture-based decoders obtains better object discovery (FG-ARI) performance on synthetic dataset compared to transformer-based decoders. Building on this observation, we implement mixture-based decoders from Locatello et al. (2020) as BO-QSA's decoder for reproduce on C-Tetrominoes dataset. We use same encoder and decoder architecture as Slot Attention demonstrated at Table 21.

**T-SNE visualization of latent representations**    We first analyze the distribution of each models latent representations (slots) via t-SNE as in Section 4. Figure 26 and Figure 27 demonstrates that, BO-QSA also exhibit factor-wise homogeneity property and inter-task separation in slot representation space. We observe that BO-QSA shows more tighter regions within identical semantic factors compared to Slot Attention.

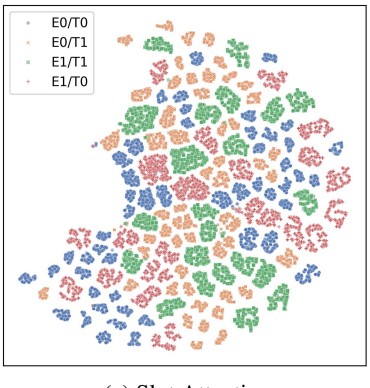 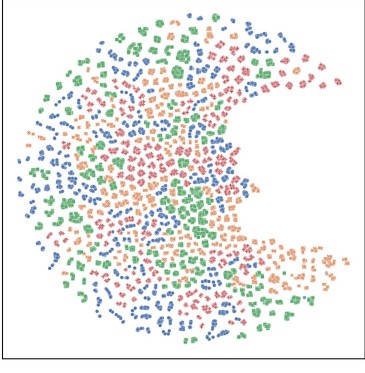

(a) Slot Attention          (b) BO-QSA

Figure 26: Inter-task separation visualization of (a) Slot Attention, (b) SlotMLP, (c) MONet via t-SNE.

**Quantitative analysis of Inter-Task Separation**    We also evaluate BO-QSA's inter-task separation following the quantitative analysis in Section 4.2 (i.e., inter-task class similarity and nearest neighbor class purity). Figure 28 demonstrates that BO-QSA maintains high inter-task separation, showing similar results compared to Slot Attention. While Slot Attention shows higher nearest neighbor class purity and the, BO-QSA shows significantly higher scores compared to other baselines (e.g. MONet in Figure 14).

**Object discovery performance of BO-QSA on continual setting**    We further evaluate object discovery performance (FG-ARI) of BO-QSA on C-Terominoes SST dataset (Table 16). We also evaluate performance of applying DPR on BO-QSA. Note that we report FG-ARI results of 5 trials

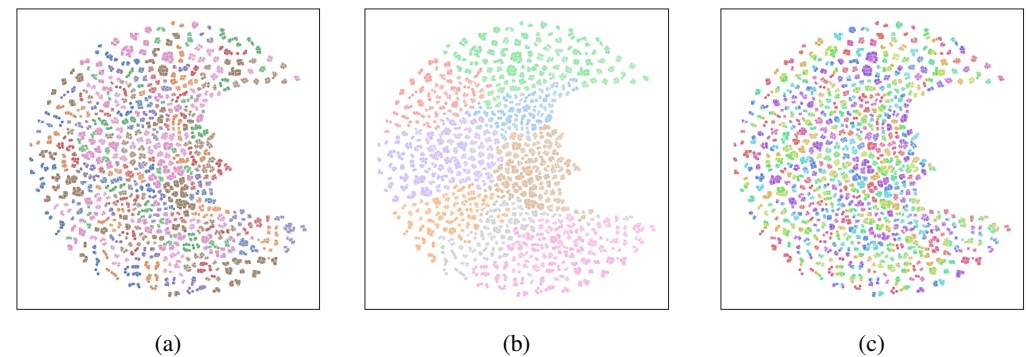

(a)           (b)           (c)

Figure 27: t-SNE visualization results of slots from BO-QSA. Each color of the dots represent (a) *shape*, (b) *position*, (c) *color*.

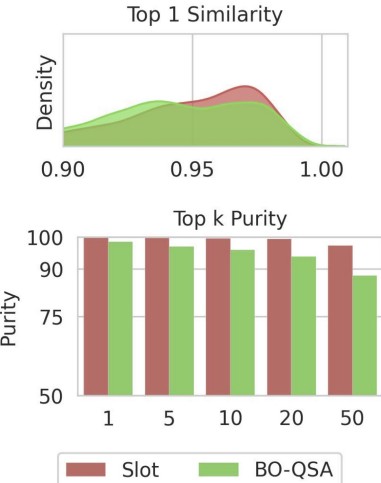

Figure 28: Quantitative evaluation of inter-task separation of BO-QSA using (top) Top-$k$ Inter-Task Class Similarity and (bottom) Top-$k$ Nearest Neighbor Class Purity. "Slot" denotes Slot Attention, which exhibits well-separated features across tasks.

for Slot Attention and 3 trials for BO-QSA (FG-ARI$_{\pm std.}$). BO-QSA shows better performance on C-Tetrominoes throughout continual tasks compared to Slot Attention. While, BO-QSA also exhibit degradation of performance on previous task after training on novel task (E0/T1), applying DPR on BO-QSA significantly improvements performances on previous tasks while maintaining performance on novel task.

| Model | E0/T0 | E0/T1 | E1/T1 |
|---|---|---|---|
| Slot Attention | 0.9986 $_{\pm 0.00}$ | 0.4171 $_{\pm 0.03}$ | 0.9979 $_{\pm 0.00}$ |
| Slot Attention + DPR | 0.9988 $_{\pm 0.00}$ | 0.9881 $_{\pm 0.00}$ | 0.9970 $_{\pm 0.00}$ |
| BO-QSA | 0.9998 $_{\pm 0.00}$ | 0.5046 $_{\pm 0.08}$ | 0.9983 $_{\pm 0.00}$ |
| BO-QSA + DPR | 0.9988 $_{\pm 0.00}$ | 0.9654 $_{\pm 0.02}$ | 0.9953 $_{\pm 0.00}$ |

Table 16: FG-ARI$_{(\pm std.)}$ performance on C-Tetrominoes SST evaluated across previous tasks $\mathcal{T}_i$ (E$i$) after training on novel task $\mathcal{T}_j$ (T$j$). SA$^\dagger$ denotes Slot Attention.

Both qualitative and quantitative analyses of BO-QSA's slot representations indicate that this advanced Slot Attention–based variant also exhibits factor-wise homogeneity and inter-task separation.

Furthermore, Table 16 shows that this property provides an effective inductive bias for continual tasks, and that applying DPR yields substantial performance improvements.

A.14 ROBUSTNESS ACROSS DIFFERENT SEMANTIC FACTORS

In this paper, we primarily focus on introducing novel *shape* classes during sequential training, as *shape* provides a broader range of variation compared to *position* or *color*, which are limited to bounded continuous ranges (SectionSection 3). However, as shown in Section 4, the *factor-wise homogeneity* property of Slot Attention shows that slot representations small and well-separated regions preserve identical semantic factors not only for *shape*, but also for *position* and *color* factors (Section4). This observation is supported by the t-SNE visualizations in Figure 2. To further explore the effectiveness of factor-wise homogeneity property, we evaluate the robustness of DPR under variations in other factors by (i )introducing novel *color* classes and (ii) combination with *shape* and *color* classes during sequential tasks.

**Introducing Novel Color Factor**  To this end, we introduce *C-Tetrominoes Single Step addition of Two classes of Color (SST-Color)*. Unlike C-Tetrominoes SST, which introduces novel *shape* classes, C-Tetrominoes SST-Color introduces novel *color* classes while preserving identical *shape* classes across tasks. Table 17 demonstrates the configurations of C-Tetrominoes SST-Color, and full semantic label definitions are provided in Appendix C.2.

| Task | Shape | Color | Position | Object | Background | Train | Eval |
|---|---|---|---|---|---|---|---|
| $\mathcal{T}_0$ | 1, 2, 3, 4, 5 | 1, 2, 3, 4, 9 | x, y | 3 | 1 | 25,000 | 5,000 |
| $\mathcal{T}_1$ | 1, 2, 3, 4, 5 | 5, 7 | x, y | 3 | 1 | 10,000 | 5,000 |

Table 17: Configuration for C-Tetrominoes SST-Color.

We provide both quantitative and qualitative evaluations of DPR on the C-Tetrominoes SST-Color benchmark. Figure 29 shows evaluation results on each tasks. Similar to results on C-Tetrominoes SST. Figure 29 reports per-task performance. Consistent with results from C-Tetrominoes SST, we observe that Slot Attention suffers performance degradation on previously learned tasks after training on new ones In contrast, applying DPR significantly improves performance across various evaluation metrics. While the MSE result for task pair *(1-1)* shows slightly lower performance with DPR, the average performance across $t = 1$ is improved, as shown in Figure 29b. Qualitative results in Figure 30 further support this observation: whereas vanilla Slot Attention generates objects with inaccurate semantics, Slot Attention with DPR successfully reconstructs objects that retain the correct semantics of previously seen tasks. These results highlight the robustness of both the *factor-wise homogeneity* property and DPR across different semantic factors beyond *shape*, such as *color*.

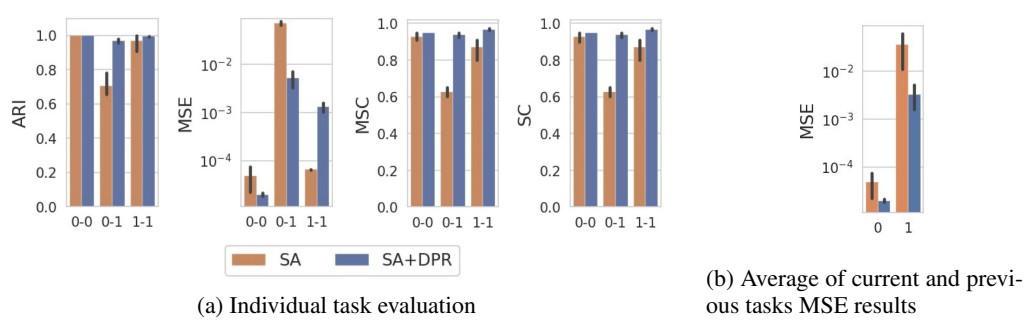

(a) Individual task evaluation

(b) Average of current and previous tasks MSE results

Figure 29: Quantitative results of Slot Attention with and without DPR on C-Tetronimoes SST-Color, where novel *color* factor is introduced.

**Introducing Novel Combination of Shape and Color Factor**  Moreover, to examine the generality of our approach across different types of factor variation, we also consider a setting where both shape and color factors are incrementally introduced. Specifically, we design *C-Tetrominoes Single-Step addition of Two classes of Combination of shape and color (SST-Comb)*, which introduces novel combinations of *shape* and *color* classes. Table 17 summarizes the task configurations for C-Tetrominoes SST-Comb, while the full semantic label definitions are provided in Appendix C.2.

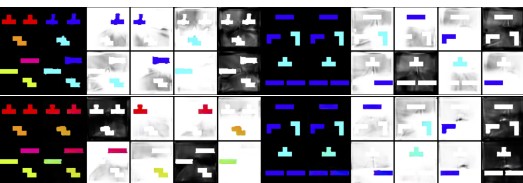

Figure 30: Qualitative results of Slot Attention with (top) without (bottom) DPR on C-Tetronimoes SST-Color.

| Task | Shape | Color | Position | Object | Background | Train | Eval |
|------|-------|-------|----------|--------|------------|-------|------|
| $\mathcal{T}_0$ | 1, 2, 3, 4, 5 | 1, 2, 3, 4, 9 | x, y | 3 | 1 | 25,000 | 5,000 |
| $\mathcal{T}_1$ | 5, 6 | 5, 7 | x, y | 3 | 1 | 10,000 | 5,000 |

Table 18: Configuration for C-Tetrominoes SST-Comb.

| Model | E0/T0 (*After Initial Task*) | | E0/T1 (*After Last Task*) | |
|-------|------|------|------|------|
| | FG-ARI | MSE | FG-ARI | MSE |
| Slot Attention | | | $0.48_{\pm0.00}$ | $0.093_{\pm0.01}$ |
| DPR$^\dagger$ | $0.97_{\pm0.01}$ | $0.0004_{\pm0.01}$ | $\mathbf{0.93}_{\pm0.02}$ | $\mathbf{0.008}_{\pm0.01}$ |

Table 19: Qualitative results of Slot Attention with without DPR on C-Tetronimoes SST-Comb, where combination of novel *shape* and *color* factor is introduced. DPR$^\dagger$ denotes DPR with Slot Attention.

As shown in Table 19, our method significantly improves the retention of previous task knowledge while maintaining strong performance on the new task. These results corroborate our claim that *factor-wise homogeneity* facilitates well-separated slot representations across tasks, thereby mitigating interference in continual learning. Building on this structure, DPR further strengthens the ability to preserve object-specific factors, yielding consistent benefits across scenarios with multiple types of variation.

Taken together, these findings indicate that our discovery and proposed method are not confined to a specific type of factor, but instead provide a general and robust effect that consistently contributes to performance improvements across diverse settings.

## B  APPENDIX: IMPLEMENTATIONS AND TRAINING CONFIGURATIONS

### B.1  ALGORITHMS

We provide pseudo-code implementations for three replay strategies used in our experiments: (1) Decoder-only Post Replay, (2) Post Replay, and (3) Decoder-only Replay without Task Training. Each method differs in which components are updated and when replay is applied, as summarized in Algorithms 1, 2, and 3. We highlight the differences between three replay strategies.

---

**Algorithm 1** Decoder only Post Replay for Slot Attention

1: **Input:** Encoder $\mathcal{F}_\theta$, Slot Attention $\mathcal{SA}_\theta$, Decoder $\mathcal{G}_\theta$, loss $\mathcal{L}$, learning rate $\eta$, replay buffer $R$
2: **for** $t := 1$ to $T$ **do**
3:     Train $\mathcal{F}_\theta, \mathcal{SA}_\theta, \mathcal{G}_\theta$ on $\mathcal{T}_t$                         ▷ Main task training phase
4:     Sample subset of current task $\mathcal{T}_t^* \subseteq \mathcal{T}_t$
5:     $R_t^* \leftarrow R \cup \mathcal{T}_t^*$                         ▷ Replay + current task subset
6:     Freeze $\mathcal{F}_\theta, \mathcal{SA}_\theta$
7:     **for** ep $:= 1$ to $N_{replay}$ **do**                         ▷ Post replay phase
8:         Sample mini-batch $B$ from $R_t^*$
9:         $\mathcal{G}_\theta \leftarrow \mathcal{G}_\theta - \eta \cdot \nabla_{\mathcal{G}_\theta} \mathcal{L}(B; \mathcal{F}_\theta, \mathcal{SA}_\theta, \mathcal{G}_\theta)$
10:     **end for**
11:     Store subset of generated samples in buffer $R_t \subseteq \hat{X}_t$
12:     $R \leftarrow R \cup R_t$
13: **end for**

---

**Algorithm 2** Post Replay for Slot Attention

1: **Input:** Encoder $\mathcal{F}_\theta$, Slot Attention $\mathcal{SA}_\theta$, Decoder $\mathcal{G}_\theta$, loss $\mathcal{L}$, learning rate $\eta$, replay buffer $R$
2: **for** $t := 1$ to $T$ **do**
3:     Train $\mathcal{F}_\theta, \mathcal{SA}_\theta, \mathcal{G}_\theta$ on $\mathcal{T}_t$                         ▷ Main task training phase
4:     Sample subset of current task $\mathcal{T}_t^* \subseteq \mathcal{T}_t$
5:     $R_t^* \leftarrow R \cup \mathcal{T}_t^*$                         ▷ Replay + current task subset
6:     **for** ep $:= 1$ to $N_{replay}$ **do**                         ▷ Post replay phase
7:         Sample mini-batch $B$ from $R_t^*$
8:         $\{\mathcal{F}_\theta, \mathcal{SA}_\theta, \mathcal{G}_\theta\} \leftarrow \{\mathcal{F}_\theta, \mathcal{SA}_\theta, \mathcal{G}_\theta\} - \eta \cdot \nabla \mathcal{L}(B; \mathcal{F}_\theta, \mathcal{SA}_\theta, \mathcal{G}_\theta)$
9:     **end for**
10:     Store subset of generated samples in buffer $R_t \subseteq \hat{X}_t$
11:     $R \leftarrow R \cup R_t$
12: **end for**

---

**Algorithm 3** Decoder only Replay without Task training for Slot Attention

1: **Input:** Encoder $\mathcal{F}_\theta$, Slot Attention $\mathcal{SA}_\theta$, Decoder $\mathcal{G}_\theta$, loss $\mathcal{L}$, learning rate $\eta$, replay buffer $R$
2: **for** $t := 1$ to $T$ **do**
3:     Sample subset of current task $\mathcal{T}_t^* \subseteq \mathcal{T}_t$
4:     $R_t^* \leftarrow R \cup \mathcal{T}_t^*$                         ▷ Replay + current task subset
5:     Freeze $\mathcal{F}_\theta, \mathcal{SA}_\theta$
6:     **for** ep $:= 1$ to $N_{replay}$ **do**                         ▷ Replay phase
7:         Sample mini-batch $B$ from $R_t^*$
8:         $\theta \leftarrow \theta - \eta \cdot \nabla_\theta \mathcal{L}(B; \mathcal{F}_\theta, \mathcal{SA}_\theta, \mathcal{G}_\theta)$
9:     **end for**
10:     Store subset of generated samples in buffer $R_t \subseteq \hat{X}_t$
11:     $R \leftarrow R \cup R_t$
12: **end for**

---

### B.2  TRAINING IMPLEMENTATIONS

**Hyper-parameters**  We provide detailed training configurations used in our experiments. Following prior works Locatello et al. (2020); Dittadi et al. (2021), we adopt their implementation settings

with modifications to accommodate our C-OCL benchmarks. Table 20 summarizes the default hyperparameters used across all experiments. Note that for C-CLEVR, which uses higher-resolution images ($128 \times 128$), we set the batch size to 32.

| Hyperparameter | Default ($\mathcal{T}_0$) | ($\mathcal{T}_{t>0}$) | Post Replay at $t$ | Joint Replay at $t$ |
|---|---|---|---|---|
| Optimizer | Adam Kingma & Ba (2014) | - | - | - |
| Learning rate | $4 \times 10^{-4}$ | - | - | - |
| Warm-up steps | 2% of total steps | - | - | - |
| Batch size | 64 | - | - | 32+32 |
| Resolution | ($64 \times 64$) | - | - | - |
| Training epochs | 200 | 100 | 50 | 50 |
| Train | 25000 | 10000 | $2000 \times t$ | $\mathcal{T}_t + 2000 \times (t-1)$ |
| Evaluation | 5000 | 5000 | None | None |
| Buffer size | None. | - | 2000 per $t$ | 2000 per $t$ |

Table 20: Default hyper-parameters.

**Grid search on hyper-parameters for DPR**  We provide a comparison of implementation variants of Decoder-Only Post Replay, focusing on two key elements: (1) replay buffer size and (2) number of replay epochs. Figure 31 presents the results for each variation.

In terms of ARI, a buffer size of 2,000 provides strong performance, while increasing the size beyond 2,000 does not yield further significant improvement. In contrast, reducing the buffer size to 100 leads to noticeable performance degradation. For MSE, the best result is achieved with a buffer size of 5,000, though 2,000 achieves comparable performance. For replay epoch ablation, we compare against the baseline of 50 replay epochs, which matches the number of epochs used in task training. Variants with fewer epochs show slight degradation in both ARI and MSE. Based on these findings, we use buffer size of 2,000 and replay epochs of 50 for our DPR.

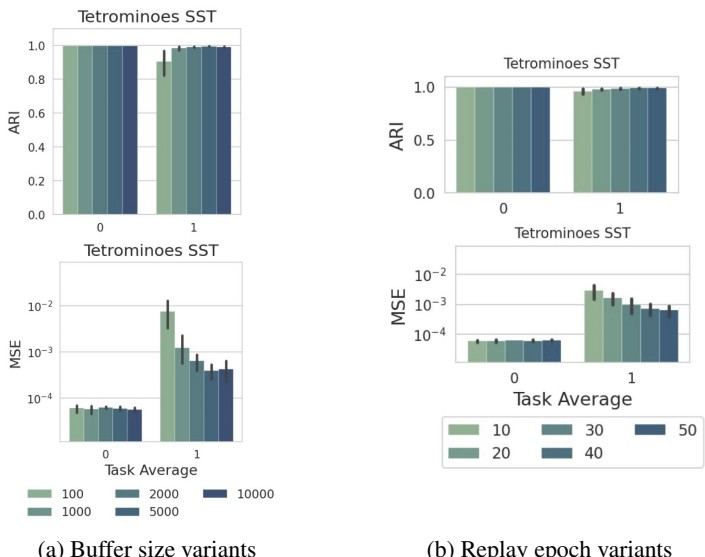

(a) Buffer size variants      (b) Replay epoch variants

Figure 31: Grid search on (1) replay buffer size, (2) replay epoch.

## B.3 MODEL IMPLEMENTATIONS

**Slot Attention** We follow the Slot Attention implementation from Locatello et al. (2020); Dittadi et al. (2021). For the C-Tetrominoes dataset, which has a resolution of $64 \times 64$, we use the architecture described in Table 21. For the C-CLEVR dataset (Table 22, we apply several modifications to handle the larger image resolution of $128 \times 128$. Specifically, we use convolutional channels of size 64 and broadcast the slot representations to a spatial resolution of $8 \times 8$ instead of matching the full input resolution, when feeding them into the broadcast decoder.

| Module | Type / Operation | Details |
|---|---|---|
| Encoder | Conv $5 \times 5$ | 32 channels, stride 1, padding 2, ReLU |
| | Conv $5 \times 5$ | 32 channels, stride 1, padding 2, ReLU |
| | Conv $5 \times 5$ | 32 channels, stride 1, padding 2, ReLU |
| | Conv $5 \times 5$ | 32 channels, stride 1, padding 2, ReLU |
| | Positional Embedding | Added to conv output |
| | GroupNorm + Conv $1 \times 1$ | 32 channels, ReLU, Flatten |
| Slot Attention | Slot Initialization | Learnable $\mu, \sigma$ |
| | Slot | $D_{slots} = 64$ |
| | GRU Update | GRUCell(dim=64) per slot |
| | MLP | 2-layer FFN (hidden dim=128), ReLU |
| | Iteration | 3 |
| Decoder | Spatial Broadcast | Repeat slot $\rightarrow (width \times height)$ |
| | Positional Embedding | Added to broadcasted slots |
| | Conv $5 \times 5$ | 32 channels, stride 1, padding 2, ReLU |
| | Conv $5 \times 5$ | 32 channels, stride 1, padding 2, ReLU |
| | Conv $5 \times 5$ | 32 channels, stride 1, padding 2, ReLU |
| | Conv $3 \times 3$ | 4 channels (RGB + alpha), stride 1, padding 1 |

Table 21: Slot Attention architecture for Tetrominoes ($64 \times 64$) resolution.

| Module | Type / Operation | Details |
|---|---|---|
| Encoder | Conv $5 \times 5$ | 64 channels, stride 1, padding 2, ReLU |
| | Conv $5 \times 5$ | 64 channels, stride 1, padding 2, ReLU |
| | Conv $5 \times 5$ | 64 channels, stride 1, padding 2, ReLU |
| | Conv $5 \times 5$ | 64 channels, stride 1, padding 2, ReLU |
| | Positional Embedding | Added after conv block |
| | GroupNorm + Conv $1 \times 1$ | 64 channels, ReLU, Flatten |
| Slot Attention | Slot Initialization | Learnable $\mu, \sigma$ |
| | Slot | $D_{slots} = 64$ |
| | GRU Update | GRUCell (dim=64) per slot |
| | MLP Block | 2-layer FFN with ReLU, hidden dim = 128 |
| | Iteration | 3 |
| Decoder | Spatial Broadcast | Repeat slot $\rightarrow (8 \times 8)$ via repeat |
| | Positional Embedding | Added to broadcasted slot maps |
| | Transposed Conv $5 \times 5$ | 64 ch., stride 2, padding 2, output pad 1, ReLU |
| | Transposed Conv $5 \times 5$ | 64 ch., stride 2, padding 2, output pad 1, ReLU |
| | Transposed Conv $5 \times 5$ | 64 ch., stride 2, padding 2, output pad 1, ReLU |
| | Transposed Conv $5 \times 5$ | 64 ch., stride 2, padding 2, output pad 1, ReLU |
| | Conv $5 \times 5$ | 64 ch., stride 1, padding 2, ReLU |
| | Conv $3 \times 3$ | 4 ch. (RGB+mask), stride 1, padding 1 |

Table 22: Slot Attention architecture for CLEVR ($128 \times 128$) resolution.

### B.4 BASELINE IMPLEMENTATIONS

**Variational Auto Encoder** Implementation of Variational Auto Encoder (VAE) Kingma et al. (2013) is demonstrated at Table 23. Following Dittadi et al. (2021), we set the dimension of the latent vector to $D_{latent} = D_{slot} \times K$, where $K$ refers to the number of slots (objects including the background) introduced in the single image.

| Module | Type / Operation | Details |
|---|---|---|
| Encoder | Conv 1 | Conv2D(3 → 64,kernel=5,stride=2,pad=2),LeakyReLU |
| | ResBlock 1.1 | Conv2D(64 → 64), BN, LeakyReLU |
| | ResBlock 1.2 | Conv2D(64 → 64), BN, LeakyReLU |
| | Downsample 1 | Conv2D(64 → 128, stride=2) |
| | ResBlock 2.1 | Conv2D(128 → 128), BN, LeakyReLU |
| | ResBlock 2.2 | Conv2D(128 → 128), BN, LeakyReLU |
| | Downsample 2 | Conv2D(128 → 128, stride=2) |
| | ResBlock 3.1 | Conv2D(128 → 256), BN, LeakyReLU |
| | ResBlock 3.2 | Conv2D(256 → 256), BN, LeakyReLU |
| | Flatten | $256 \times H' \times W'$ to 1D |
| | MLP | Linear($\cdot$ → 512) + LayerNorm + LeakyReLU |
| Decoder | Input Layer | Conv2D(256 → 256,kernel=5,pad=2),LeakyReLU |
| | ResBlock 1.1 | Conv2D(256 → 256), BN, LeakyReLU |
| | ResBlock 1.2 | Conv2D(256 → 256), BN, LeakyReLU |
| | Upsample 1 | ConvTranspose2D(256 → 128, stride=2) |
| | ResBlock 2.1 | Conv2D(128 → 128), BN, LeakyReLU |
| | ResBlock 2.2 | Conv2D(128 → 128), BN, LeakyReLU |
| | Upsample 2 | ConvTranspose2D(128 → 64, stride=2) |
| | ResBlock 3.1 | Conv2D(64 → 64), BN, LeakyReLU |
| | ResBlock 3.2 | Conv2D(64 → 64), BN, LeakyReLU |
| | Final Conv | Conv2D(64 → 3, kernel=5, pad=2), Sigmoid |

Table 23: Architecture specification of the Baseline VAE.

**Learning without forgetting** Learning without forgetting (LwF) Li & Hoiem (2017) is a regularization-based continual learning method that preserves knowledge (knowledge distillation) of previous tasks by maintaining the output predictions (logits) of the old model through distillation, without storing any past data. It enables the model to learn new tasks while minimizing performance degradation on old tasks using a combination of cross-entropy and knowledge distillation losses. However, implementing LwF to Slot Attention (SA+LwF) requires few modifications. Since Slot Attention trained by mean squared error (MSE), it does not have a classifier. So, instead of maintaining output predictions (logits), SA+LwF performs knowledge distillation by maintaining the output of decoder $\mathcal{G}$.

$$\mathcal{L} = \lambda \cdot \mathcal{L}_{new}(X_n, \hat{X}_n) + \mathcal{L}_{old}(X_o, \hat{X}_o) \tag{5}$$
$$\text{where } \hat{X}_n = \mathcal{G}_o(\mathcal{SA}_s(\mathcal{F}_s(X_n)))$$
$$\text{and } \hat{X}_n = \mathcal{G}_n(\mathcal{SA}_s(\mathcal{F}_s(X_n)))$$

We utilize parameters of decoder as task specific parameters and parameters of encoder $\mathcal{F}$ and Slot Attention $\mathcal{SA}$ as shared parameters. We train SA+LwF by Equation 6 with additional weight decay (5e-04). We follow details from Li & Hoiem (2017), and performed grid search to find the best $\lambda$ for SA+LwF 32 and use the value of $\lambda$=1, softmax temperature 2.0.

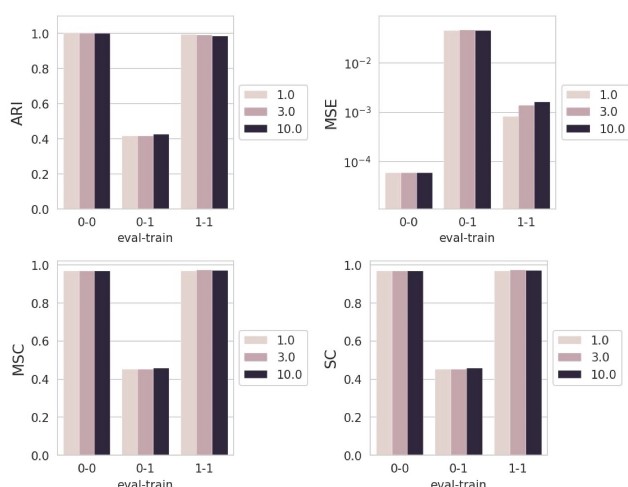

Figure 32: SA+LwF performance for different balancing parameter $\lambda$.

**Elastic Weight Consolidation** Elastic Weight Consolidation (EWC) Kirkpatrick et al. (2017) is a regularization-based continual learning method that prevents forgetting by penalizing changes to important weights for previous tasks. To constrain important parameters to stay close to their old values, EWC implements this constraints as a quadratic penalty. It uses the Fisher Information Matrix to estimate the importance of each parameter. This helps the model retain old knowledge while learning new tasks.

$$\log p(\theta \mid \mathcal{D}) = \log p(\mathcal{D}_B \mid \theta) + \log p(\theta \mid \mathcal{D}_A) - \log p(\mathcal{D}_B) \tag{6}$$

In order to justify this choice of constraint and to define which weights are most important for a task, EWC consider neural network training from a probabilistic perspective. Equation 6 shows that the approximation of the posterior from task A as a prior for task B, helping the model preserve knowledge from previous tasks while learning new ones. However, since true posterior probability is intractable, EWC approximate the posterior as a Gaussian distribution with mean given by the parameters $\theta_A$ and a diagonal precision given by the diagonal of the Fisher information matrix $F$.

$$\mathcal{L} = \mathcal{L}_B(\theta) + \sum_i \frac{\lambda}{2} F_i (\theta_i - \theta_{A,i}^*)^2 \tag{7}$$

Equation 7 is objective of EWC. $\lambda$ is an hyper-parameter, which balances between old task and new one based on the importance of each task. We follow the implementation of EWC Kirkpatrick et al. (2017) and train (SA+EWC). For $\lambda$, EWC use $\{0.7, 10, 25, 90\}$ for sequential MNIST, CIFAR10, and mini-ImageNet dataset. However, For Slot Attention which use MSE loss, we found that previous settings are not effective. We performed a grid search (Figure 33), and use $\lambda$=1e+04 for SA+EWC.

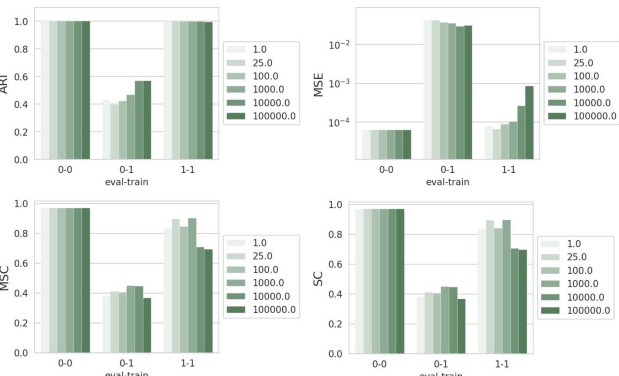

Figure 33: SA+EWC performance for different balancing parameter $\lambda$.

**Experience Replay**   We apply standard Experience Replay (ER) to Slot Attention (SA+ER). Following prior work Lopez-Paz & Ranzato (2017a); Rebuffi et al. (2017); Chaudhry et al. (2019); Buzzega et al. (2020a), we perform *joint training* by concatenating replay samples with the current task's training data. SA+ER maintains a replay buffer that stores reconstruction results, similar to DPR (Post Replay), and uses the same buffer size. However, unlike DPR, which updates only the decoder after training, SA+ER performs joint training by sampling previous task data from the buffer and mixing it with current task data at the mini-batch level to mitigate catastrophic forgetting. Following Lopez-Paz & Ranzato (2017a), we concatenate replay and current samples in equal proportion. Details of training implementation of joint training are in Table 20.

**Generative Replay**   We also evaluate a generative variant (SA+GR) inspired by Generative Replay (GR) methods Shin et al. (2017); Wu et al. (2018); Ayub & Wagner; Zhai et al. (2019); Gao & Liu (2023), which utilize auxiliary generative models (e.g., GANs Shin et al. (2017), VAEs Shin et al. (2017), or diffusion models Gao & Liu (2023)) to synthesize replay samples. In our implementation, we adopt a Variational Autoencoder (VAE) Kingma et al. (2013) as the generative model, following Shin et al. (2017), and use the architecture described in Table 23. Replay samples are generated from a Gaussian prior $\mathcal{N}(0, 1)$ and matched in quantity to those used in DPR. As in ER, we perform joint training by concatenating generated samples with the current task data in each mini-batch. Details of training implementation of joint training are in Table 20.

B.5   EVALUATION METRICS

**Adjusted Rand Index (ARI)**   ARI Rand (1971) shows the segmentation quality of each slot via interpreting each segmentation mask as clustering. It quantifies the similarity between predicted segmentation of slots and ground-truth segmentation using clustering evaluation metrics. It corrects the Rand Index (RI) by accounting for chance agreement. The ARI ($\uparrow$) ranges from $-1$ to $1$, where $1$ indicates perfect clustering, $0$ indicates random labeling, and negative values indicate worse than random.

Let $a$ be the number of pairs of elements that are in the same cluster in both the predicted and ground-truth partitions, and $b$ be the number of pairs in different clusters in both. Then the ARI is defined as:

$$\mathrm{ARI} = \frac{\sum_{ij} \binom{n_{ij}}{2} - \left[ \sum_i \binom{a_i}{2} \sum_j \binom{b_j}{2} \Big/ \binom{n}{2} \right]}{\frac{1}{2} \left[ \sum_i \binom{a_i}{2} + \sum_j \binom{b_j}{2} \right] - \left[ \sum_i \binom{a_i}{2} \sum_j \binom{b_j}{2} \Big/ \binom{n}{2} \right]} \tag{8}$$

where:

- $n_{ij}$ is the number of elements in both predicted cluster $i$ and ground-truth cluster $j$,
- $a_i = \sum_j n_{ij}$, the number of elements in predicted cluster $i$,
- $b_j = \sum_i n_{ij}$, the number of elements in ground-truth cluster $j$,
- $n$ is the total number of samples.

Following details from Locatello et al. (2020) and only consider masks of foreground objects for computing ARI.

**Mean Squared Error (MSE)**   Mean Squared Error (MSE) quantifies the average squared difference between an original input image and its reconstructed version. We use the MSE score as an measure to evaluate the semantic accuracy captured by the model through its reconstruction quality

Given the input image $x \in \mathbb{R}^{H \times W \times C}$ and the reconstructed image $\hat{x} \in \mathbb{R}^{H \times W \times C}$, the MSE is defined as:

$$\mathrm{MSE} = \frac{1}{HWC} \sum_{i=1}^{H} \sum_{j=1}^{W} \sum_{k=1}^{C} (x_{ijk} - \hat{x}_{ijk})^2 \tag{9}$$

where $H$, $W$, and $C$ denote the height, width, and number of channels of the image, respectively.

**Segmentation Covering (SC) and mean Segmentation Covering (mSC)**    Segmentation Covering (SC) Arbelaez et al. (2010) evaluates how well each ground-truth segment is covered by the best-matching predicted segment. Higher SC($\uparrow$)

Given ground-truth segments $G = G_1, G_2, \ldots, G_m$ and predicted segments $P = P_1, P_2, \ldots, P_n$, the SC score is computed as:

$$\text{SC}(G, P) = \frac{1}{\sum_i |G_i|} \sum_i \max_j \frac{|G_i \cap P_j|}{|G_i \cup P_j|} \tag{10}$$

where:

- $|G_i|$ is the number of pixels in ground-truth segment $G_i$,
- $|G_i \cap P_j|$ is the intersection (overlap) between $G_i$ and predicted segment $P_j$,
- $|G_i \cup P_j|$ is the union of the two segments.

Mean Segmentation Covering (mSC) Engelcke et al. is an extended version of SC. mSC computes the average covering score across all images in a dataset. It quantifies how well predicted segments align with ground-truth segments over the entire evaluation set.

Given a dataset of $N$ images, where $G^{(i)}$ and $P^{(i)}$ denote the ground-truth and predicted segment sets for image $i$, the mSC is defined as:

$$\text{mSC} = \frac{1}{N} \sum_{i=1}^{N} \text{SC}(G^{(i)}, P^{(i)}), \quad \text{where } \text{SC}(G, P) = \frac{1}{\sum_k |G_k|} \sum_k \max_j \frac{|G_k \cap P_j|}{|G_k \cup P_j|} \tag{11}$$

where:

- $|G_k|$ is the number of pixels in the $k$-th ground-truth segment,
- $|G_k \cap P_j|$ is the overlap between ground-truth segment $G_k$ and predicted segment $P_j$,
- $|G_k \cup P_j|$ is their union.

A high SC ($\uparrow$) score indicates good alignment and accurate localization of objects in the predicted segmentation. mSC ($\uparrow$) values indicate better and robust segmentation quality and object alignment across images.

**Mean Best Overlap (mBO)**    The mean Best Overlap (mBO) Pont-Tuset et al. (2016) measures the quality of object discovery by evaluating the overlap between predicted slot masks and ground-truth object masks. For each ground-truth object, we compute the Intersection-over-Union (IoU) with all predicted masks and take the maximum IoU as the best match. The mBO ($\uparrow$) is then defined as the average of these best-match IoUs across all ground-truth objects in the dataset. A higher mBO indicates that predicted slots more accurately align with true objects, regardless of label permutation. Formally, let $\mathcal{G} = g_1, \ldots, g_{|\mathcal{G}|}$ denote the set of ground-truth masks and $\mathcal{P} = p_1, \ldots, p_{|\mathcal{P}|}$ the set of predicted masks. The best overlap score for a ground-truth object $g_i$ is defined as:

$$\text{BO}(g_i) = \max_{p_j \in \mathcal{P}} \frac{|g_i \cap p_j|}{|g_i \cup p_j|}, \tag{12}$$

where $|\cdot|$ denotes the number of pixels. The mean Best Overlap is then given by:

$$\text{mBO} = \frac{1}{|\mathcal{G}|} \sum_{i=1}^{|\mathcal{G}|} \text{BO}(g_i). \tag{13}$$

## C  APPENDIX: CONTINUAL-OBJECT CENTRIC LEARNING BENCHMARKS

In this section, we demonstrate our Continual-Object Centric benchmark. The goal is to evaluate the ability of object-centric methods on the task of unsupervised object discovery. we introduce two benchmarks: (1) Continual-Tetrominoes, and (2) Continual-CLEVR. These datasets build upon the original Tetrominoes and CLEVR Johnson et al. (2017). Details on these datasets are discussed in the following sections.

Each dataset consists stream of tasks $\mathcal{T} = \{\mathcal{T}_\sqcup\}_{t=1}^{N}$, where $N$ is the the total number of tasks. Each $\mathcal{T}_t$ comprises multi-object images, with additional semantic labels for each object. During training unsupervised object discovery, the model is not given semantic labels. In C-OCL, novel object classes ($\mathcal{C}_t$) are incrementally introduced across tasks, where object classes of each task $\mathcal{T}_t$ are mutually exclusive across tasks (*disjoint*), ensuring that objects presented in task $\mathcal{T}_t$ never appear in any previous or future task. In this work, we focus on introducing novel *shape* classes, as shape provides a broader range of variation compared to *position* or *color*, which are limited to bounded continuous ranges.

### C.1  TRAINING SCENARIOS

We adopt three training and evaluation scenarios inspired by prior work Shmelkov et al. (2017); Michieli & Zanuttigh (2019); Cermelli et al. (2020), with modifications tailored for object-centric learning. In all scenarios, we ensure that at the initial task is consisted with five classes and at least two distinct object classes are introduced whenever novel classes are presented. The scenarios are defined as follows: (1) *Single Step addition of Two classes* (SST), (2) *Single Step addition of Multiple classes* (SSM), and (3) *Multi Step addition of Two classes* (MST) per step. Table 24 demonstrates details of each training scenarios.

| Setting | Task 0 | | | Task 1 | | | Task 2–5 | | |
|---|---|---|---|---|---|---|---|---|---|
| | $\mathcal{C}_0$ | Train | Eval. | $\mathcal{C}_1$ | Train | Eval. | $\mathcal{C}_{2\sim5}$ | Train | Eval. |
| SST | 5 | 25000 | 5000 | 2 | 10000 | 5000 | – | – | – |
| SSM | 5 | 25000 | 5000 | 5 | 10000 | 5000 | – | – | – |
| MST | 5 | 25000 | 5000 | 2 | 10000 | 5000 | 2 | 10000 | 5000 |

Table 24: Task configuration for C-OCL under three scenarios.

### C.2  CONTINUAL-TETROMINOES

Our C-Tetrominoes dataset builds upon the original Tetrominoes Kabra et al. (2019), which consists of images containing 3 Tetris pieces placed on a black background. The original dataset includes 19 different Tetris shapes and 6 distinct colors. Some of these shapes are symmetric under rotation or reflection.

In our work, we follow the 6 canonical Tetris shapes used in Tetrominoes Kabra et al. (2019), excluding those with symmetric equivalence. To introduce additional object classes, we incorporate shapes from the Pentomino dataset Montero et al., which originally contains twelve pentomino shapes. From these, we select 9 distinct shapes to ensure visual and semantic diversity.

In total, our C-Tetrominoes dataset consists of 15 object shapes, each rendered in one of 10 different colors, placed on a uniform black background. Object placement is limited to 8 fixed positions per image to control for spatial variability.

We adopt the terminology from Dittadi et al. (2021) to define semantic attributes in C-Tetrominoes. Specifically, the semantic labels consist of:

- *Shape*: 15 foreground object classes + 1 background class
  - 0 background class
  - 1 'Horizontal I'

- 2 'Horizontal Z'
- 3 'L pointing downward'
- 4 'T pointing upward'
- 5 'L pointing right'
- 6 'O'
- 7 'F'
- 8 'T'
- 9 'U'
- 10 'Z'
- 11 'X'
- 12 'L'
- 13 'N'
- 14 'P'
- 15 'V'

- *Color*: 10 foreground color classes + 1 background class
    - 0 $(0, 0, 0)$
    - 1 $(227, 74, 51)$
    - 2 $(227, 170, 55)$
    - 3 $(190, 222, 67)$
    - 4 $(101, 222, 101)$
    - 5 $(77, 213, 202)$
    - 6 $(81, 133, 244)$
    - 7 $(138, 91, 254)$
    - 8 $(203, 91, 232)$
    - 9 $(226, 70, 127)$
    - 10 $(228, 60, 105)$

- *Position*: 8 foreground spatial bins + 1 background class

Our C-OCL benchmarks have three different evaluation scenarios. We introduce details of each scenarios in the following paragraphs. For each dataset, we provide semantic labels of each object, with additional *mask* for each object.

**C-Tetrominoes Single Step addition of Two classes** C-Tetrominoes Single Step addition of Two classes (SST) consist of two sequential tasks, introducing two new object shapes. Other factors except *shape* are identically shared across tasks.

| Task | Shape | Color | Position | Object | Background | Train | Eval |
|------|-------|-------|----------|--------|------------|-------|------|
| $\mathcal{T}_0$ | 1, 2, 3, 4, 5 | 1–10 | x, y | 3 | 1 | 25,000 | 5,000 |
| $\mathcal{T}_1$ | 6, 7 | 1–10 | x, y | 3 | 1 | 10,000 | 5,000 |

Table 25: Configuration for C-Tetrominoes SST.

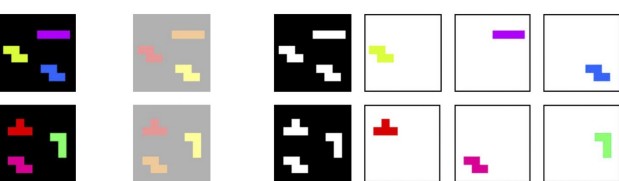

Figure 34: Examples of images of C-Tetrominoes SST $\mathcal{T}_0$. Starting from *left*, original image, concatenated mask, individual objects.

**C-Tetrominoes Single Step addition of Multiple classes** C-Tetrominoes Single Step addition of Multiple classes (SSM) consist of two sequential tasks, introducing five new object shapes. Other factors except *shape* are identically shared across tasks.

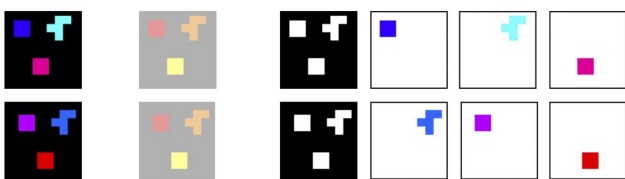

Figure 35: Examples of images of C-Tetrominoes SST $\mathcal{T}_1$. Starting from *left*, original image, concatenated mask, individual objects.

| Task | Shape | Color | Position | Object | Background | Train | Eval |
|---|---|---|---|---|---|---|---|
| $\mathcal{T}_0$ | 1, 2, 3, 4, 5 | 1–10 | x, y | 3 | 1 | 25,000 | 5,000 |
| $\mathcal{T}_1$ | 6, 7, 8, 9, 10 | 1–10 | x, y | 3 | 1 | 10,000 | 5,000 |

Table 26: Configuration for C-Tetrominoes SSM.

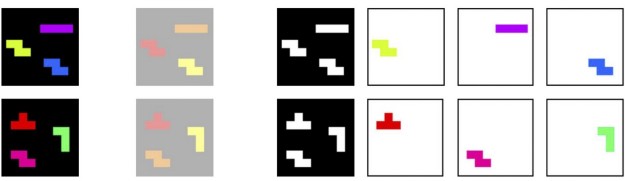

Figure 36: Examples of images of C-Tetrominoes SSM $\mathcal{T}_0$. Starting from *left*, original image, concatenated mask, individual objects.

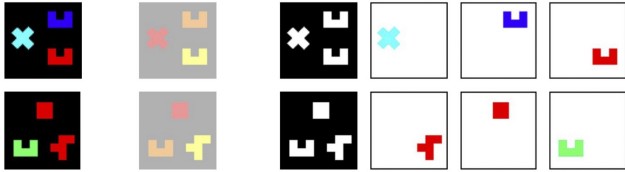

Figure 37: Examples of images of C-Tetrominoes SSM $\mathcal{T}_1$. Starting from *left*, original image, concatenated mask, individual objects.

**C-Tetrominoes Multi Step addition of Two classes** C-Tetrominoes Multi Step addition of Two classes (MST) consist of multiple sequential tasks, introducing two new object shapes. Other factors except *shape* are identically shared across tasks.

| Task | Shape | Color | Position | Object | Background | Train | Eval |
|---|---|---|---|---|---|---|---|
| $\mathcal{T}_0$ | 1, 2, 3, 4, 5 | 1–10 | x, y | 3 | 1 | 25,000 | 5,000 |
| $\mathcal{T}_1$ | 6, 7 | 1–10 | x, y | 3 | 1 | 10,000 | 5,000 |
| $\mathcal{T}_2$ | 8, 9 | 1–10 | x, y | 3 | 1 | 10,000 | 5,000 |
| $\mathcal{T}_3$ | 10, 11 | 1–10 | x, y | 3 | 1 | 10,000 | 5,000 |
| $\mathcal{T}_4$ | 12, 13 | 1–10 | x, y | 3 | 1 | 10,000 | 5,000 |
| $\mathcal{T}_5$ | 14, 15 | 1–10 | x, y | 3 | 1 | 10,000 | 5,000 |

Table 27: Configuration for C-Tetrominoes MST.

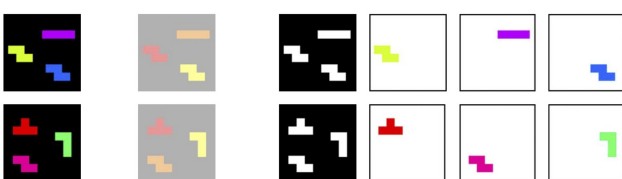

Figure 38: Examples of images of C-Tetrominoes MST $\mathcal{T}_0$. Starting from *left*, original image, concatenated mask, individual objects.

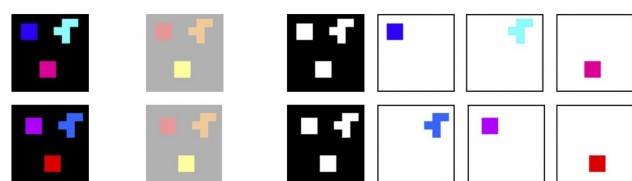

Figure 39: Examples of images of C-Tetrominoes MST $\mathcal{T}_1$. Starting from *left*, original image, concatenated mask, individual objects.

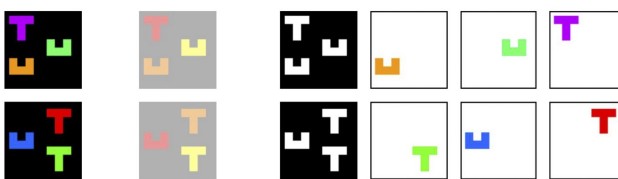

Figure 40: Examples of images of C-Tetrominoes MST $\mathcal{T}_2$. Starting from *left*, original image, concatenated mask, individual objects.

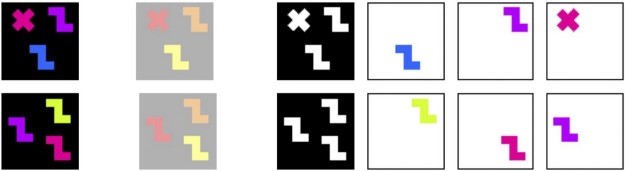

Figure 41: Examples of images of C-Tetrominoes MST $\mathcal{T}_3$. Starting from *left*, original image, concatenated mask, individual objects.

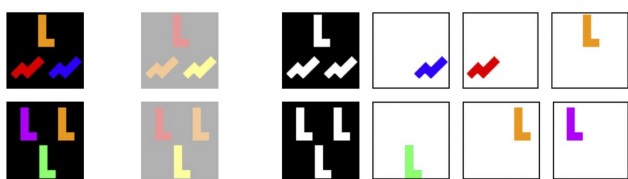

Figure 42: Examples of images of C-Tetrominoes MST $\mathcal{T}_4$. Starting from *left*, original image, concatenated mask, individual objects.

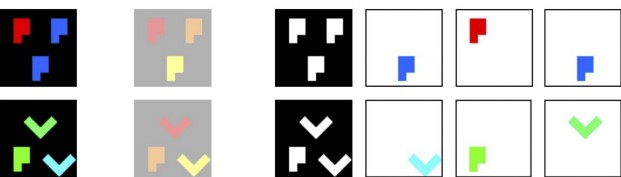

Figure 43: Examples of images of C-Tetrominoes MST $\mathcal{T}_5$. Starting from *left*, original image, concatenated mask, individual objects.

## C.3 CONTINUAL-CLEVR

Our C-CLEVR dataset builds upon the CLEVR dataset Johnson et al. (2017), which features synthetic 3D scenes composed of up to 10 objects placed on a uniform gray background. Unlike Tetrominoes, CLEVR incorporates object occlusions, providing a more complex visual setting. The original CLEVR dataset includes 3 object shapes, 6 colors, continuous $(x, y)$ positions, 2 sizes, 2 materials, and object rotations. The CATER dataset Girdhar & Ramanan (2020), an extension of CLEVR designed for evaluating spatiotemporal reasoning, introduces two additional object shapes.

In constructing our C-CLEVR dataset, we incorporate 5 object shapes drawn from CLEVR and CATER, while preserving the original semantic attributes. To simulate continual learning, we introduce additional object classes that are incrementally added across tasks.

In total, C-CLEVR contains 15 object shapes, each rendered in one of 6 colors, 3 sizes, and 2 material types, with placement on a continuous $(x, y)$ space over a gray background. Similar to CLEVR6, a subset of CLEVR, our C-CLEVR always consist 6 objects in the synthetic 3D scene.

- *Shape*: 15 foreground object classes + 1 background class
    - 0 background class
    - 1 "cube"
    - 2 "sphere"
    - 3 "cylinder"
    - 4 "cone"
    - 5 "spl"
    - 6 "hourglass"
    - 7 "pudding"
    - 8 "tetrahedron"
    - 9 "octahedron"
    - 10 "dodecahedron"
    - 11 "icosahedron"
    - 12 "cross"
    - 13 "stellateddodecahedron"
    - 14 "torus"
    - 15 "spring"
- *Color*: 6 foreground color classes + 1 background class
    - 0 "gray" $(87, 87, 87)$
    - 1 "red" $(173, 35, 35)$
    - 2 "blue" $(42, 75, 215)$
    - 3 "green" $(29, 105, 20)$
    - 4 "brown" $(129, 74, 25)$
    - 5 "purple" $(129, 38, 192)$
    - 6 "cyan" $(41, 208, 208)$
    - 7 "yellow" $(255, 238, 51)$
- *Position*
    - x
    - y
- *Size*
    - "large"
    - "medium"
    - "small"
- *Materials*
    - "metal"
    - "rubber"

Our C-OCL benchmarks have three different evaluation scenarios. We introduce details of each scenarios in the following paragraphs. For each dataset, we provide semantic labels of each object, with additional *mask* for each object.

**C-CLEVR Tetrominoes Single Step addition of Two classes**   C-CLEVR Single Step addition of Multiple classes (SSM) consist of two sequential tasks, introducing two new object shapes. Other factors except *shape* are identically shared across tasks.

| Task | Shape | Color | Position | Object | Background | Train | Eval |
|------|-------|-------|----------|--------|------------|-------|------|
| $\mathcal{T}_0$ | 1, 2, 3, 4, 5 | 1–7 | x, y | 6 | 1 | 25,000 | 5,000 |
| $\mathcal{T}_1$ | 6, 7 | 1–7 | x, y | 6 | 1 | 10,000 | 5,000 |

Table 28: Configuration for C-CLEVR SSM.

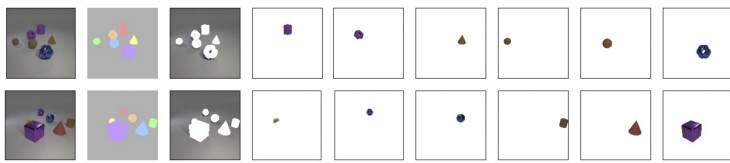

Figure 44: Examples of images of C-CLEVR SST $\mathcal{T}_0$. Starting from *left*, original image, concatenated mask, individual objects.

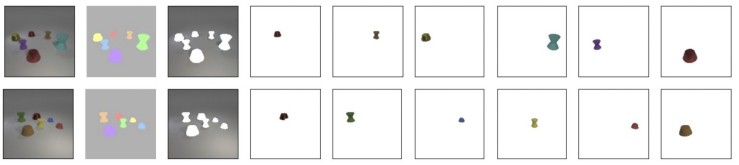

Figure 45: Examples of images of C-CLEVR SST $\mathcal{T}_1$. Starting from *left*, original image, concatenated mask, individual objects.

**C-CLEVR Single Step addition of Multiple classes**   C-CLEVR Single Step addition of Multiple classes (SSM) consist of two sequential tasks, introducing five new object shapes. Other factors except *shape* are identically shared across tasks.

| Task | Shape | Color | Position | Object | Background | Train | Eval |
|------|-------|-------|----------|--------|------------|-------|------|
| $\mathcal{T}_0$ | 1, 2, 3, 4, 5 | 1–10 | x, y | 6 | 1 | 25,000 | 5,000 |
| $\mathcal{T}_1$ | 6, 7, 8, 9, 10 | 1–11 | x, y | 6 | 1 | 10,000 | 5,000 |

Table 29: Configuration for C-CLEVR SSM.

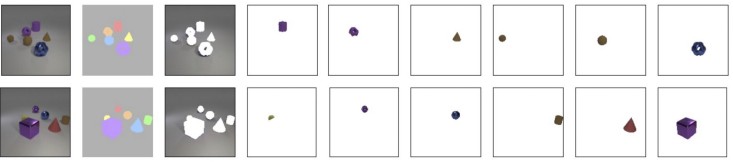

Figure 46: Examples of images of C-CLEVR SSM $\mathcal{T}_0$. Starting from *left*, original image, concatenated mask, individual objects.

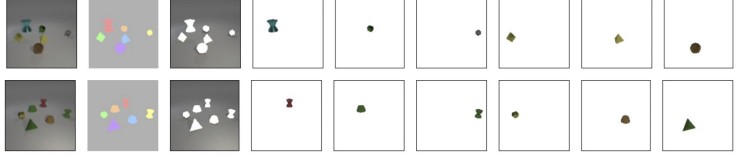

Figure 47: Examples of images of C-CLEVR SSM $\mathcal{T}_1$. Starting from *left*, original image, concatenated mask, individual objects.

**C-CLEVR Multi Step addition of Two classes**   C-CLEVR Multi Step addition of Two classes (MST) consist of multiple sequential tasks, introducing two new object shapes. Other factors except *shape* are identically shared across tasks.

| Task | Shape | Color | Position | Object | Background | Train | Eval |
|------|-------|-------|----------|--------|------------|-------|------|
| $\mathcal{T}_0$ | 1, 2, 3, 4, 5 | 1–10 | x, y | 6 | 1 | 25,000 | 5,000 |
| $\mathcal{T}_1$ | 6, 7 | 1–10 | x, y | 6 | 1 | 10,000 | 5,000 |
| $\mathcal{T}_2$ | 8, 9 | 1–10 | x, y | 6 | 1 | 10,000 | 5,000 |
| $\mathcal{T}_3$ | 10, 11 | 1–10 | x, y | 6 | 1 | 10,000 | 5,000 |
| $\mathcal{T}_4$ | 12, 13 | 1–10 | x, y | 6 | 1 | 10,000 | 5,000 |
| $\mathcal{T}_5$ | 14, 15 | 1–10 | x, y | 6 | 1 | 10,000 | 5,000 |

Table 30: Configuration for C-CLEVR MST.

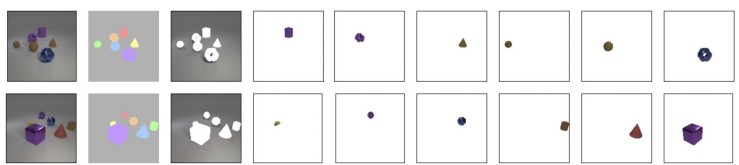

Figure 48: Examples of images of C-CLEVR MST $\mathcal{T}_0$. Starting from *left*, original image, concatenated mask, individual objects.

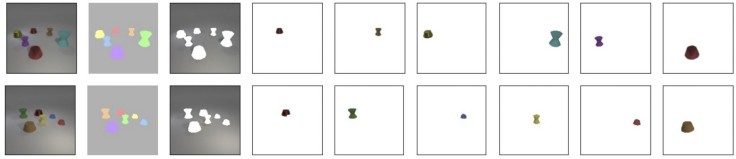

Figure 49: Examples of images of C-CLEVR MST $\mathcal{T}_1$. Starting from *left*, original image, concatenated mask, individual objects.

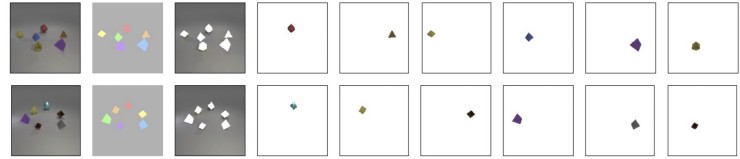

Figure 50: Examples of images of C-CLEVR MST $\mathcal{T}_2$. Starting from *left*, original image, concatenated mask, individual objects.

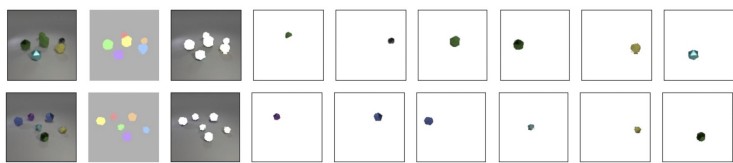

Figure 51: Examples of images of C-CLEVR MST $\mathcal{T}_3$. Starting from *left*, original image, concatenated mask, individual objects.

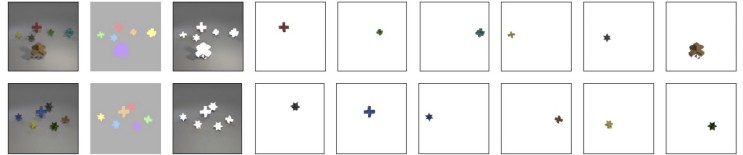

Figure 52: Examples of images of C-CLEVR MST $\mathcal{T}_4$. Starting from *left*, original image, concatenated mask, individual objects.

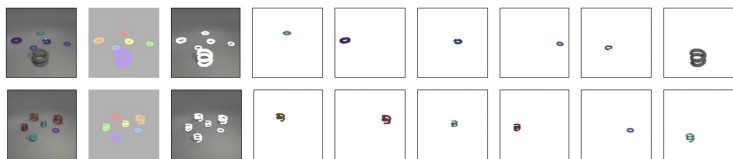

Figure 53: Examples of images of C-CLEVR MST $\mathcal{T}_5$. Starting from *left*, original image, concatenated mask, individual objects.

