# OpenReview forum: "Factor-Wise Homogeneity of Slot-Attention for Continual Object-Centric Learning"
_ICLR.cc/2026/Conference — Submitted to ICLR 2026_

### Official Review · Reviewer_o9jF · 2025-10-24

**Soundness:** 2
**Presentation:** 2
**Contribution:** 1
**Rating:** 2
**Confidence:** 5

**Summary:**

The paper studies continual object-centric learning and finds that Slot Attention naturally organizes slot features into small, factor-consistent, well-separated neighborhoods across tasks, even for unseen objects and tasks. Following this, they propose Decoder-Only Post Replay (DPR): after each task, freeze encoder + slots and fine-tune only the decoder on a mix of buffered past reconstructions and some current data to combat forgetting. Across Tetrominoes/CLEVR/COCO and multiple task schedules, SA+DPR improves FG-ARI and MSE vs. vanilla SA.

**Strengths:**

This paper studies the problem continual learning in object-centric learning. There is not lot of work on this topic therefore it seems like an interesting topic to study.

**Weaknesses:**

My main concern is that the the authors say that the encoder + slot attention can transfer to new tasks and only the decoder needs to be trained for adaptation. I have two critiques regarding this -

1. I think the main goal of object centric representations is to learn good representations for downstream tasks. This paper only evaluates segmentation via ARI. SAM and other segmentation methods can already do that well and still generalize, so in my view evaluating these methods on ARI is not well motivated.
2. If the authors observe that only the decoder needs to be updated for transfer, why not just use the masks from slot attention for computing ari? why do you need the decoder at all? Many works do this - https://arxiv.org/abs/2110.11405, https://arxiv.org/abs/2209.14860

Secondly, the observation that the encoder + slot representations transfer well across domains was already shown in https://openreview.net/forum?id=bSq0XGS3kW, so I believe that the observation is nothing new. In fact the mentioned paper shows that object-centric models can transfer to new domains and objects very well without seeing much samples during training so I don't see why we need a continual learning setup at all in object-centric learning?

**Questions:**

See weaknesses

---

> ### Author Response · Authors · 2025-11-21
>
> We would like to thank the reviewer (**o9jF**) for valuable insights. We address the reviewer's concerns as follows:
>
> ### W1-1. OCL vs. Segmentation and justification for the ARI metric
>
> 1. Clarification of different research fields
>
> First, we clarify the fundamental difference in objectives between Object-Centric Learning (OCL) models and specialized segmentation methods like SAM. SAM (Segment Anything Model) is a segmentation model, which is typically supervised or prompt-driven, primarily focused on generating high-quality masks. In contrast, the main goal of Object-Centric Learning is to learn individual, disentangled representations for every object present in a scene, without supervised segmentation masks.
>
> 2. Motivations for using ARI metric
>
> Since OCL aims for unsupervised object discovery, Adjusted Rand Index (ARI) is the standard and most direct metric used to verify the quality of the learned individual slot representations and their ability to segment the scene. This practice is consistent with the original Slot Attention[1] paper and the broader OCL community[2,3] for evaluating the success of the foundational object discovery task.
>
> ### W1-2. Necessity of decoder for mask prediction
>
> We acknowledge the reviewer’s suggestion of reducing the reliance of the decoder for prediction. However, we note that while using raw attention for evaluation masks is technically feasible, the majority of advanced Slot Attention-based models rely on the decoder's output for the final mask used in evaluation.
>
> To address this concern, we have added a clarification in **Section 5.5 (Discussion & Limitation)**, including a discussion of insightful suggestions for future work and the limitations of current evaluation frameworks.
>
> 1. Types of decoders and prediction method in Slot Attention and its variations
> - Slot Attention [1]: Broadcasting CNN Decoder (Alpha-mask)
> - SPOT [2]: Transformer decoder (Decoder attention mask)
> - DINOSAUR [3]: Transformer decoder (Decoder attention mask for COCO and Pascal VOC) or Broadcasting MLP Decoder (Alpha-mask for MOVi and KITTY)
>
>     > Quote from source paper: "Masks for Evaluation As mentioned in the main text, we use the attention mask from the decoder as the slot mask used for evaluation."
>     >
> - SLATE [4]: Transformer decoder (with dVAE decoder). While its original paper focuses on generation measures (FID, MSE,  training curves of CNN discriminator) and lacks ARI, implementations from DINOSAUR [3] utilize decoder attention masks for ARI.
>
> Majority of advanced Slot Attention-based models rely on the decoder's output for the final mask used in evaluation, reinforcing the full generative objective of OCL.
>
> 2. Clarification of our main focus and updates in the manuscript
>
> We explicitly note that our focus is on analyzing the intrinsic continual-learning behavior of Slot Attention rather than introducing new decoding strategies, and we leave systematic improvements to decoding or prediction modules for future work.
>
> We acknowledge reviewer’s suggestion regarding reducing reliance on the decoder to avoid generalization bottlenecks. In the revised manuscript, we have added a clarification in **Section 5.5 (Discussion & Limitation).** We explain that, although reducing decoder dependence may seem intuitive, majority of object-centric frameworks yet require decoder outputs to satisfy reconstruction-based objectives.

---

> > ### Author Response · Authors · 2025-11-21
> > **Follow up**
> >
> > ### W2 Separation of Roles: Transfer vs. Continual Learning and the structural significance of Factor-wise Homogeneity
> >
> > The cited work demonstrated that OCL models exhibit strong zero-shot generalization to new objects and domains, providing observations of transfer performance. We clarify that transfer and continual learning address fundamentally different challenges under different constraints. In this context, our main finding highlights the effectiveness of factor-wise homogeneity in enabling inter-task separation during sequential adaptation—specifically, by making Slot Attention robust against latent interference. We believe this addresses a challenge that is distinct from, and not resolved by, zero-shot transfer learning.
> >
> > 1. Distinct challenges in transfer and continual learning
> > - Transfer (Zero-Shot) focuses on generalizing knowledge from one task $\mathcal{T}_A$ to an unseen task $\mathcal{T}_B$ without further training on $\mathcal{T}_B$. It does not account for the accumulation of knowledge over many tasks or the need to preserve $\mathcal{T}_A$ performance while adapting to $\mathcal{T}_B$.
> > - Continual Learning focuses on accumulating knowledge over a sequence of tasks, utilizing a structure that is robust against latent interference during sequential adaptation and aiming to mitigate catastrophic forgetting of previous tasks while retaining plasticity for new ones.
> >
> > Zero-shot transfer provides a beneficial advantage but does not inherently solve the core CL challenge of knowledge preservation during incremental learning.
> >
> > 1. Clarification or our main findings and main challenges addressed
> >
> > The reviewer emphasizes transferability—that object-centric models can generalize to various datas without further training. However, **our contribution lies on identifying and analyzing the preservation of factor-wise property and inter-task separation during sequential adaptation**, which we think that is different phenomenon. Our analysis shows that Slot Attention uniquely maintains this separation under continual updates, enabling robust multi-task retention—something not addressed or evaluated in prior transfer-only studies.
> >
> > - References
> >
> > [1] Locatello et al., “Object-Centric Learning with Slot Attention,” NeurIPS 2020.
> >
> > [2] Zablotskaia et al., “SPOT: Slot Attention with Transformers for Object-Centric Learning,” CVPR 2023.
> >
> > [3] Seitzer et al., “DINOSAUR: Bridging the Gap to Real-World Object-Centric Learning,” ICCV 2023.
> >
> > [4] Singh et al., “SLATE: Illiterate DALL-E Learns to Compose,” NeurIPS 2022.

---

> > > ### Comment · Reviewer_o9jF · 2025-11-26
> > >
> > > I thank the authors for their comments.
> > >
> > > I understand that majority of works in OCL used decoder based masks but my point was mainly that you could still use masks from an attention and would not have the update the decoder at all. I am curious if the authors could atleast try this for their setup and report the results? I think the decoder being used to obtain masks for other works is not a good reason why that should be the case in this work. Is there any reason to believe that masks from slot attention are worse than the ones from the decoder? Because if the masks from slot attention are as good as those from the decoder then there is no need for your method right?
> > >
> > > I am still not convinced about why this setting needs to be studied if you keep the encoder + slot attention fixed across different tasks, hence I would like to keep my score.

---

> ### Author Response · Authors · 2025-12-03
>
> ### Q1. ARI results using Attention mask prediction
>
> To reduce the reviewer’s concern, we added evaluation using attention mask for mask prediction. The results show that attention masks from Slot Attention exhibit stronger robustness under continual updates, which aligns with our central finding of factor-wise homogeneity and inter-task separation. However, attention masks still underperform decoder-based masks in more complex environments, consistent with observations reported in prior OCL work.
>
> Table below compares FG-ARI results for three settings that differ in how masks are predicted:
>
> (1) Slot Attention using decoder outputs,
>
> (2) Slot Attention using attention masks from the slot attention module, and
>
> (3) Slot Attention + DPR using decoder outputs.
>
> | Dataset | Model | Mask prediction | E0/T0 | E1/T0 (Zero-shot) | E0/T1 (Previous) | E1/T1 |
> | --- | --- | --- | --- | --- | --- | --- |
> | C-Tetrominoes | SA* | Decoder output | 0.9986$_{\pm 0.00}$ | 0.3997$_{\pm 0.06}$ | **0.4171$_{\pm 0.03}$** | 0.9978$_{\pm 0.00}$ |
> | C-Tetrominoes | SA* | Attention mask | 0.9890$_{\pm 0.00}$ | 0.9864$_{\pm 0.00}$ | **0.9862$_{\pm 0.00}$** | 0.9888$_{\pm 0.00}$ |
> | C-Tetrominoes | SA + DPR | Decoder output | 0.9987$_{\pm 0.00}$ | - | **0.9880$_{\pm 0.00}$** | 0.9970$_{\pm 0.00}$ |
> |  |  |  |  |  |  |  |
> | C-CLEVR | SA* | Decoder output | 0.9600$_{\pm 0.00}$ | 0.6044$_{\pm 0.05}$ | **0.6383$_{\pm 0.03}$** | 0.962$_{\pm 0.00}$ |
> | C-CLEVR | SA* | Attention mask | 0.9254$_{\pm 0.05}$ | 0.8261$_{\pm 0.06}$ | **0.8519$_{\pm 0.14}$** | 0.8567$_{\pm 0.18}$ |
> | C-CLEVR | SA + DPR | Decoder output | 0.9987$_{\pm 0.00}$ | - | **0.9880$_{\pm 0.00}$** | 0.9970$_{\pm 0.01}$ |
>
> *: SA (Slot Attention)
>
> -: do not apply DPR for unseen task (E1) after training on initial task (T0)
>
> First, we observe that attention masks continue to reliably identify objects from the previous task after training on the novel task (E0/T1), whereas decoder masks degrade due to limited generalization capability of the decoder. **Attention masks exhibit stronger robustness under continual updates**, which is consistent with our central finding that Slot Attention’s factor-wise homogeneity and the resulting inter-task separation provide effective inductive bias against inter-task interference in continual learning.
>
> Second, further evaluation on unseen-task samples (E1/T0) shows that attention masks maintain their effectiveness, while decoder-based masks fail to generalize. This behavior follows the same observation in [1], indicating that **our findings remain valid in zero-shot settings** and offering an interpretable explanation for why Slot Attention maintains object-level consistency even for unseen objects.
>
> Finally, we observe that attention masks perform strongly on relatively simple datasets (such as Tetrominoes) but yield lower performance than decoder-based masks on more complex scenes (CLEVR). This suggests that, while attention masks can provide meaningful and robust object masks in continual settings, their **performance remains limited compared to decoder outputs**.
>
> ### Q2. Is there any reason to believe that masks from slot attention are worse than the ones from the decoder?
>
> Building on the findings in the previous table, we also reference prior work showing that decoder-based mask prediction generally provides superior segmentation performance compared to attention masks.
>
> > The Transformer decoder does not produce an alpha mask. Instead, we have two options: the attention masks of Slot Attention (used by SLATE), or the decoder’s attention mask over the slots. **We found that the latter performed better**, and we use it throughout. [2]
> >
>
> > Empirically, we **observe that masks produced during decoding demonstrate superior object decomposition**, i.e., better object segmentation…
> Building on this insight, we propose self-training scheme that distills slot-based attention masks from the decoder to the encoder, thereby enhancing the object segmentation information captured by the slots. [3]
> >
>
> [1] Didolkar et al., “On the Transfer of Object-Centric Representation Learning”, ICLR 2025.
>
> [2] Seitzer et al., “DINOSAUR: Bridging the Gap to Real-World Object-Centric Learning,” ICLR 2023.
>
> [3] Kakogeorgiou et al., “SPOT: Slot Attention with Transformers for Object-Centric Learning,” CVPR 2024.

---

### Official Review · Reviewer_cgDZ · 2025-10-31

**Soundness:** 3
**Presentation:** 3
**Contribution:** 3
**Rating:** 6
**Confidence:** 3

**Summary:**

This paper investigates the factor-wise homogeneity property of Slot Attention and its implications for Continual Object-Centric Learning (C-OCL). The authors introduce new benchmarks—Continual-Tetrominoes and Continual-CLEVR—to evaluate continual unsupervised object discovery, and propose a simple method termed Decoder-only Post Replay (DPR). DPR freezes the encoder and slot attention modules while fine-tuning only the decoder, thereby leveraging the observed factor-wise homogeneity to mitigate catastrophic forgetting. Extensive experiments, both quantitative and qualitative, demonstrate the stability of this property and the effectiveness of DPR across synthetic and real-world datasets.

**Strengths:**

- The identification and empirical characterisation of factor-wise homogeneity in Slot Attention is an original and conceptually interesting contribution. It provides a new perspective on how object-centric representations are organised in latent space.
- The introduction of the Continual-Tetrominoes and Continual-CLEVR benchmarks represents a meaningful step toward systematic evaluation of continual object-centric learning. These resources could be of lasting value to the community.
- The proposed Decoder-only Post Replay (DPR) is simple, elegant, and easy to reproduce. Its minimalistic design strengthens the argument that the observed behaviour of Slot Attention itself underpins the improvements.
- The experiments are extensive and cover synthetic, complex, and real-world datasets. The authors further conduct ablations, comparisons with diverse baselines, and analysis of compatibility with regularisation-based methods.
- The manuscript is well structured, with clear motivation, detailed explanations of methodology, and coherent visualisations that support the claims.

**Weaknesses:**

- While the empirical findings are compelling, the paper lacks a rigorous theoretical analysis explaining why factor-wise homogeneity emerges in Slot Attention. The argument remains largely observational.
-Although effective, the DPR method could be viewed as a minor algorithmic variation (a post-hoc replay scheme). The conceptual novelty mainly lies in the analysis, not in the proposed learning strategy.
- The continual learning scenarios considered (mainly new shape classes) are somewhat narrow. Broader evaluations involving more complex semantic shifts (e.g., textures, dynamics, or object relationships) would better support general claims.
-The improvements, while consistent, are moderate in some settings. It would strengthen the paper to include statistical significance tests or further qualitative explanations of the observed gains.
-The paper could engage more deeply with recent object-centric continual learning frameworks or compositional representation studies (e.g., those integrating diffusion-based or transformer-based architectures).
- Although the authors mention code availability in the supplement, hyperparameter and architectural details are partly deferred to the appendix, which may limit immediate reproducibility from the main text.
-Some sections (especially Section 4) are lengthy and could benefit from tighter exposition. The key insights are occasionally obscured by repetition and detail overload.

**Questions:**

plz see my detailed comments above

---

> ### Author Response · Authors · 2025-11-21
>
> We appreciate the reviewer’s (**cgDZ**) detailed comments and helpful suggestions. We address the reviewer's concerns as follows:
>
> ### W1. R**igorous theoretical analysis explaining of our core findings**
>
> We acknowledge that a fully formal theory for the emergence of factor-wise homogeneity is limited in the current paper and not yet established.
>
> Motivated from reviewer **jYDk**’s suggestion, we conducted an in-depth analysis of GRU dynamics within Slot Attention. Our analysis demonstrate that GRU dynamics serve as the key mechanism supporting both factor-wise consistency and continual task separation, providing the structural foundation for effective continual object-centric learning.
>
> We kindly ask the reviewers to refer to:
>
> - Our motivation and conclusions: This post
> - For summary: **Official Response (Official Response: Analysis on GRU Dynamics and Factor-Wise Homogeneity)**
> - Full description: **Appendix A.3 (red highlights)**
> - For explanation: Response to jYDk (Q1-2)
> 1. Motivations of our analysis
>
> The main question we address is whether the recurrent update dynamics of the GRU are responsible for the stability of factor-wise homogeneity under continual adaptation. To empirically analyze this phenomenon, we leverage recent theoretical studies on time-scale dynamics in gated RNNs, which demonstrate that gating mechanisms effectively create "slow modes"—hidden-state dimensions that evolve slowly and stably, serving as reliable channels for long-term information preservation.
>
> 2. Analysis conclusions
>
> Our analysis shows that the GRU’s marginally stable slow modes effectively perform an identical-mapping operation conditioned on each object’s intrinsic variability. This mechanism stabilizes slots that share the same semantic factors, providing the structural basis for the robustness of factor-wise homogeneity. Furthermore, the preservation of these factor-specific slow-mode dimensions under continual learning naturally explains the task-level separation observed in Slot Attention. Taken together, these findings demonstrate that GRU dynamics serve as the key mechanism supporting both factor-wise consistency and continual task separation, providing the structural foundation for effective continual object-centric learning.
>
> ### W2 Narrow evaluation scopes
>
> We acknowledge the importance of covering broader range of evaluation scopes for strengthen our paper. First, we would like to clarity the scope of our experiments. We acknowledge the importance of engaging more deeply with various OCL methods. To address this concern, we have added **Section 5.5 (Discussion & Limitation)** to the revised manuscript, where we discuss how different architectures may influence the behavior and applicability of our approach. We also discuss previous presented experiment results related to this concern, demonstrating that our findings are not tied to any particular architecture.
>
> 1. Clarification of our experiment scenarios
>
> While our main continual learning scenario introduces novel *shape* classes, we wish to highlight that our experiments covers a broader range of semantic shifts.
>
> - As noted in Section 5.2, we additionally evaluate settings with novel \textit{color} classes (Fig. 25 in Appendix), and combinations of novel \textit{shape} and \textit{color} (Table 9 in Appendix). Across all cases, DPR consistently preserves previous-task performance while maintaining strong performance on new tasks.
> - We also validate our findings on complex real-world datasets such as COCO and Pascal VOC using DINOSAUR, a advanced Slot-Attention variant with a pretrained Vision Transformer (DINO) backbone and transformer decoders. As shown in Table 1 (Sec. 5.3), DPR yields consistent improvements.
> - Furthermore, Table 7 (in Appendix) demonstrates robustness on CLEVRTex, which includes challenging texture variations. These results collectively show that our observations and improvements generalize not only beyond simple shape variations but also beyond simple CNN-based architectures.
> 2. Response to concerns of limited OCL method application
>
> We acknowledge the importance of engaging more deeply with various OCL methods. To address this concern, we have added **Section 5.5 (Discussion & Limitation)** to the revised manuscript, where we discuss how different decoder architectures may influence the behavior and applicability of our approach.
>
> Our focus, however, remains on analyzing the intrinsic continual-learning behavior of Slot Attention itself. We conducted our analysis on both standard Slot Attention and an advanced variant (DINOSAUR), and the results indicate that our method relies on the underlying mechanisms of Slot Attention rather than a specific architecture. We agree that extending experiments to a broader range of OCL methods would further strengthen the paper, but we believe our current results demonstrate that our findings are not tied to any particular architecture, leaving broader exploration for future work.

---

> > ### Author Response · Authors · 2025-11-21
> > **Follow up**
> >
> > ### W3 Limited training details in the main paper and unclear descriptions
> >
> > We agree that certain experimental and architectural details were deferred to the appendix. In the updated version (**red highlight**), we added the essential hyper-parameters and implementation settings into the main paper to improve immediate reproducibility. We also acknowledge that parts of Section 4 can be streamlined, and we  revised (red highlight) the exposition to present the core insights more clearly and concisely.

---

> > > ### Author Response · Authors · 2025-11-25
> > > **Follow up (additional modifications)**
> > >
> > > ### W2-2 Statistical significance and qualitative explanations
> > >
> > > We acknowledge the reviewer’s suggestion to include statistical significance and further qualitative explanations. In response, we made two additional improvements to strengthen the validity of our findings:
> > >
> > > 1. Numerical Evaluation Results
> > >
> > > Because the main performance trends in Figure 3 were primarily presented in graphical form, we additionally provide numerical FG-ARI scores to support statistical interpretation. **Appendix A.7 (red highlight)** includes Tables 6,7,8 summarizing results on C-Tetrominoes and Tables 9,10,11 summarizing results on C-CLEVR. These tables report mean and standard deviation across five runs, offering direct evidence of performance stability. We kindly ask the reviewer to refer to Appendix A.7 for detailed numerical results.
> > >
> > > 2. Qualitative Explanations
> > >
> > > We further include qualitative evidence in more complex real-world environments. As noted by the reviewer, improvements can be moderate in some settings; therefore, we added additional qualitative comparisons in **Figure 25 (in Appendix A.12, red highlight).** We kindly ask the reviewer to refer to Figure 25 for further details.
> > >
> > > Figure 25 illustrates mask-prediction results on the COCO dataset using DINOSAUR with and without DPR. We observe that DINOSAUR in certain cases fails to generate mask for objects from previous tasks after training on the novel task and sometimes segments them into multiple slots. In contrast, DPR successfully recovers these objects with coherent single-slot masks, while also maintaining mask quality on novel-task objects. Figure 25 illustrates mask-prediction results on the COCO dataset using DINOSAUR with and without DPR. We observe that DINOSAUR in certain cases fails to generate mask for objects from previous tasks after training on the novel task and sometimes segments them into multiple slots. In contrast, DPR successfully recovers these objects with coherent single-slot masks, while also maintaining mask quality on novel-task objects. These qualitative observations reinforce our central finding that factor-wise homogeneity provides a robust inductive bias that extends beyond synthetic settings to real-world continual learning scenarios.

---

> > > > ### Author Response · Authors · 2025-12-03
> > > >
> > > > ### W2-3 Recent Slot Attention model
> > > >
> > > > To more clearly resolve the reviewer’s concern, we add additional analysis of a variant of the mentioned advanced Slot Attention, BO-QSA, which employs learnable object-binding queries for slot initialization and a bi-level optimization for more stable training. Our results show that BO-QSA also exhibits factor-wise homogeneity and clear inter-task separation in its slot representation space.
> > > >
> > > > We kindly ask the reviewers to refer to **Appendix A.13 (red highlight)** in the updated manuscript.
> > > >
> > > > 1. **Analysis of advanced Slot Attention variant (BO-QSA)**
> > > >
> > > > Qualitative t-SNE visualizations **(Figure 26 and Figure 27)** show that BO-QSA also exhibits factor-wise homogeneity and clear inter-task separation in its slot representation space. The quantitative evaluation of inter-task separation **(Figure 28)** further supports this observation: BO-QSA produces results comparable to Slot Attention while maintaining a substantial gap over baselines that do not implement the Slot Attention mechanism.
> > > >
> > > > 2. **Performance of with and without applying DPR to BO-QSA**
> > > >
> > > > We provide object discovery performances (FG-ARI) of BO-QSA trained on continual settings (C-Tetrominoes). We also evaluate performance of applying DPR on BO-QSA. While, BO-QSA also exhibit critical degradation of performance on previous task after training on novel task (E0/T1), applying DPR on BO-QSA significantly improvements performances on previous tasks while maintaining performance on novel task. We ask the reviewers to refer to **Table 16**  in the updated manuscript.
> > > >
> > > > |  | E0/T0 | E0/T1 | E1/T1 |
> > > > | --- | --- | --- | --- |
> > > > | SA | 0.9986$_{\pm 0.00}$ | 0.4171$_{\pm 0.03}$ | 0.9979$_{\pm 0.00}$ |
> > > > | SA + DPR | 0.9988$_{\pm 0.00}$ | 0.9981$_{\pm 0.00}$ | 0.9970$_{\pm 0.00}$ |
> > > > | BO-QSA | 0.9998$_{\pm 0.00}$ | **0.3396**$_{\pm 0.08}$ | 0.9998$_{\pm 0.00}$ |
> > > > | BO-QSA + DPR | 0.9988$_{\pm 0.00}$ | **0.9654**$_{\pm 0.02}$ | 0.9953$_{\pm 0.00}$ |
> > > >
> > > > .

---

### Official Review · Reviewer_jYDk · 2025-11-01

**Soundness:** 2
**Presentation:** 3
**Contribution:** 2
**Rating:** 2
**Confidence:** 4

**Summary:**

This paper aims to tackle an important and underexplored problem: how to enable object-centric learning (OCL) models to perform continual learning (CL), i.e., to learn new object categories without catastrophically forgetting old ones. In summary, this work would make for a good workshop paper. It successfully defines a problem, establishes a benchmark, and provides a simple yet effective baseline. However, it lacks the "eye-opening" insight and technical depth, and is unlikely to have a profound methodological impact on the field. Therefore, I recommend rejecting this paper.

**Strengths:**

1. The authors observed a phenomenon they name "Factor-wise Homogeneity." Specifically, after training, the latent representations (slots) of the classic OCL model, Slot Attention, spontaneously organize into compact and separated clusters, where each cluster corresponds to the same semantic factor (e.g., shape). More importantly, this separation property also holds for unseen object categories.

2. Based on this discovery, the authors propose an extremely simple method, "Decoder-only Post Replay" (DPR). After learning a new task, this method freezes the encoder and the Slot Attention module (treating them as a stable generator of factor-wise homogeneous representations) and then fine-tunes only the decoder using a replay buffer containing both old and new samples.

3. The authors propose the first benchmark for Continual Object-Centric Learning (C-OCL), including Continual-Tetrominoes and Continual-CLEVR, providing an evaluation platform for future research.

**Weaknesses:**

1. The novelty of the core discovery (Factor-wise Homogeneity) is limited. This is the cornerstone of the paper, but its novelty is questionable. A well-trained representation learning model's fundamental goal is to map inputs of different semantics to separable regions in the latent space. The "factor-wise homogeneity" observed by the authors can largely be seen as an expected property that any successful representation learning model should possess, rather than a surprising, entirely new discovery. The authors' work feels more like naming and empirically verifying that Slot Attention has this desirable property, rather than unveiling a previously unknown mechanism. Therefore, packaging it as a core "discovery" seems like an overstatement.

2. The technical solution (DPR) severely lacks novelty. This is the paper's most critical weakness. The DPR method can be seen as a simple combination of existing techniques: freezing the feature extractor, experience replay, and two-stage training are all standard procedures. Essentially, DPR just assembles these simple building blocks. While effective, it feels more like a clever engineering shortcut or a "trick" than a new algorithm with profound insights. It introduces no new theory, model architecture, or optimization objective.

3. Although the authors' logic is that "the simple DPR works precisely because of factor-wise homogeneity," this feels more like a post-hoc explanation for the effectiveness of a simple method, rather than the discovery itself inspiring a novel and ingenious solution.

4. It is questionable whether the core idea proposed in this paper can lead the field. A truly inspiring work should excite other researchers and make them willing to explore new models and theories based on its core ideas. For instance, if the authors had proposed a new regularization term or model architecture to actively enhance this "homogeneity" instead of merely exploiting it, the paper's inspirational value would be much greater. The current DPR method feels more like an endpoint than a starting point.

**Questions:**

1. You present "factor-wise homogeneity" as a core discovery. A critical question is: is this property unique to Slot Attention, or is it a general characteristic found in other mainstream object-centric learning (OCL) models (e.g., MONet, IODINE, SAVi)? If it is a common property, the novelty of this discovery is diminished. If it is unique to Slot Attention, can you provide a mechanistic explanation as to why its iterative attention and GRU updates specifically give rise to this phenomenon? The current ablation study shows the importance of these components but falls short of offering a fundamental explanation.

2. The central claim of the paper is that DPR is effective precisely because of the existence of "factor-wise homogeneity." This is a causal assertion that requires more direct evidence. Could you design an experiment to demonstrate this link more explicitly? For instance, what happens if you apply the DPR method to a model that does not exhibit this property (like the SlotMLP baseline in your experiments)? Does its performance collapse catastrophically? Conversely, if you were to disrupt the homogeneity in Slot Attention through some means, would DPR's effectiveness also fail? This would provide strong support for your central argument.

---

> ### Author Response · Authors · 2025-11-21
>
> We sincerely thank the reviewer (**jYDk**) for time and effort in evaluating our work. We address the reviewer's concerns as follows:
>
> ### W1. Limited novelty of the core discovery (Factor-wise Homogeneity)
>
> We agree to the reviewer’s comments that semantic separation is a common goal in representation learning. However, we would like to clarify that our contribution is not to claim this idea as new, but to fill an analytical gap by showing that no prior work has analyzed how Slot Attention organizes its latent space.
>
> Our analysis reveals that its factor-wise homogeneity naturally induces task-separated structure in continual learning, acting as a strong inductive bias—a phenomenon not previously reported in OCL or continual learning literature.
>
> The significance of our observation lies in its utility: Slot Attention was designed for decomposed object representation learning. Introducing this specific structure of Slot Attention and the fact that it naturally yields non-overlapping representations across continual tasks is a new and meaningful observation both in OCL and continual learning domain.
>
> ### W2 & W3. Lack of contribution of DPR
>
> We acknowledge the technical simplicity of DPR but respectfully clarify that the core contribution of this work lies in analytical insight rather than architectural complexity.
>
> 1. Clarification of Our Main Contribution
>
> Our main contribution is to more clearly identify the factor-wise homogeneity present in Slot Attention representations, a behavior that has been noted only implicitly or observationally in a few prior works.
>
> 2. Motivations of our proposed method (DPR)
>
> Our goal was to analyze Slot Attention's behavior in Continual Learning and understand its strengths and limitations as an object-centric backbone.
>
> - Core Analytical Insight: We first identify the factor-wise homogeneity property of Slot Attention and empirically show that it naturally induces strong task-wise separation in CL settings, serving as a beneficial inductive bias.
> - DPR as a Minimal Solution: Based directly on this structural insight, we propose DPR (Decoder-only Post Replay). DPR is a minimal implementation derived from our empirical finding that the Slot Attention inherently maintains strongly separated representations between previous and novel tasks without external continual learning mechanisms. However, our empirical experiments suggests that poor decoder's generalization capability (as shown in Figure 5 in the main paper).
>
> Therefore, we proposed a method that revise the biased decoder after all initial task training is completed (Post Replay) to avoid interfering with the critical slot learning phase. While DPR is not technically complex, its simplicity is a feature. It serves as a simple proof-of-concept demonstrating that factor-wise homogeneity can be effectively exploited in practice.
>
> ### W4. Insights for future works
>
> We agree with the reviewer that proposing a new regularization term or model architecture to actively enhance this property (as suggested) is the next logical step. However, by identifying and understanding this unique phenomenon of Slot Attention dynamics, we believe our work raises an insightful research question that can be addressed through future developments, such as new regularization terms for enhancing this property or solving existing problems of OCL.
>
> 1. Clarification
>
> We would like to clarify that DPR (Decoder-only Post Replay) is not intended to be the main conceptual contribution of our work; it is simply a minimal, effective demonstration showing that the discovered property can be leveraged in practice.
>
> 2. Insights of our findings
>
> The identification and empirical analysis of factor-wise homogeneity, is a previously unreported phenomenon of Slot Attention that importantly provides a strong inductive bias for continual learning. This phenomenon is not commonly found in continual learning. Also, this has not been directly reported within OCL regarding how Slot Attention structures its representation space or how it behaves for unseen and previous samples, thus filling a significant gap in the field.

---

> ### Author Response · Authors · 2025-11-21
> **Follow up**
>
> ### Q1-1. Is factor-wise homogeneity property unique to Slot Attention?
>
> We acknowledge the reviewer’s comments of providing evidence of showing that our findings are unique to Slot Attention. To fill this gap, we reproduced another representative OCL baseline (MONet) to examine whether factor-wise homogeneity is unique to Slot Attention.  Experimental results show that factor-wise homogeneity is not a universal feature of Object-Centric Learning models.
>
> We kindly ask the reviewers to refer to **Appendix A.4 (updated version, red highlight)** for full descriptions.
>
> 1. Experimental results
> - As shown in Figure 12 (in Appendix A.4), MONet does not exhibit the latent separation observed in Slot Attention. When representations are labeled by \textit{task pairs}, a large portion of representations from different tasks overlap, making task-wise separation difficult.As shown in Figure 12 (in Appendix A.4), MONet does not exhibit the latent separation observed in Slot Attention. When representations are labeled by \textit{task pairs}, a large portion of representations from different tasks overlap, making task-wise separation difficult.
> - A similar pattern appears when labeled with \textit{semantic pairs} (Figure 13 in Appendix A.4): MONet shows little to no generalization for the \textit{shape} factor and only mild alignment for \textit{color} and \textit{position}, which remains insufficient to consider the representations well separated compared to Slot Attention.
> - We also evaluated MONet’s inter-task separation following the analysis in Section 4.2 (i.e., inter-task class similarity and nearest neighbor class purity). The results in Figure 14 (in Appendix A.4) further confirm that MONet lacks factor-wise homogeneity in its latent space.
> 2. Conclusion
>
> In summary, the results from both the qualitative t-SNE visualizations and the quantitative inter-task separation metrics demonstrates that factor-wise homogeneity is not a universal feature of Object-Centric Learning models. Instead, this desirable property—which is crucial for enabling effective continual learning—is uniquely attributable to the Slot Attention architecture.
>
> ### Q1-2. Fundamental explanation of this property
>
> We acknowledge the reviewer’s comment of importance of providing explanation of the emergence of factor-wise homogeneity and its stability in continual learning. With the response to reviewer’s suggestion, we conducted an in-depth analysis of GRU dynamics within Slot Attention. Our analysis demonstrate that GRU dynamics serve as the key mechanism supporting both factor-wise consistency and continual task separation, providing the structural foundation for effective continual object-centric learning.
>
> We have posted a **Official Response (2. Addition of Deeper Analysis for Understanding the Cause of the Phenomena (via GRU dynamics in Slot Attention))** and updated **Appendix A.3 (red highlights)** in the manuscript.
>
> In this post, we provide details of our analysis experiment results. We kindly ask the reviewers to refer to:
>
> - Details of our motivation and results: This post
> - For summary: Official Responses to Reviewers concerns > 2. Addition of Deeper Analysis for Understanding the Cause of the Phenomena (via GRU dynamics in Slot Attention)
> - Full description: Appendix A.3
> 1. Motivations of our analysis
>
> Our analysis is grounded in prior studies on time-scale dynamics in gated RNNs, where different hidden-state directions evolve at different rates—some changing rapidly (fast modes) while others vary slowly (**"slow mode"**)—as a consequence of gating mechanisms [1, 2, 3]. Specifically, [1] shows that gating effectively learns local time-steps, giving rise to slowly-varying modes that serve as stable channels for long-term information. [2] demonstrates that gated RNNs generate marginally stable slow modes that persist over extended horizons. [3] further shows that gates induce eigen-directions whose eigenvalues cluster near one, enabling long-timescale stability and facilitating continuous manifolds of attractors. In addition, [4] shows that trained RNNs can be locally approximated by low-dimensional linear dynamics dominated by a few slowly evolving directions, and [5] demonstrates that recurrent updates operate near stable, nearly-identity mappings. Taken together, these findings establish that gated RNNs reliably form persistent, slowly evolving subspaces—a phenomenon directly aligned with our use of GRU update-gate behavior to characterize slow modes.
>
> In this view, the theory establishes that a slow mode ****in a GRU corresponds to a hidden-state dimension whose update gate ($z_t$) remains consistently small, where the final update in each step of GRU denoted as:
>
> $h_t = (1 - z_{t}) \odot h_{t-1} + z_{t} \odot \tilde{h}_{t}$.
>
> The significance of this phenomenon is that these dimensions change minimally across iterations, acting as stable structural channels for preserving key semantic information over long horizons.

---

> ### Author Response · Authors · 2025-11-21
> **Follow up**
>
> We hypothesize that this stability is the theoretical basis for the emergence of stable, factor-specific representations. We leverage this stability property by using the GRU update gate $z_t$ as an empirical measure of dimensional stability (a 'slow mode rank'). We then apply this stability measure to analyze factor-wise homogeneity by correlating these 'slow mode ranks' across slots to confirm that slots sharing identical semantic factors rely on the same highly stable (slow mode) dimensions.
>
> 2. GRU Slow Mode Consistency Across Semantic Factors
>
> We first seek to **validate if the stability of the GRU's slow mode dimensions directly corresponds to semantic factors and remains highly consistent across slots sharing identical factors**. This test confirms the meaningfulness of our theoretical approach in explaining the observed structural robustness.
>
> To assess consistency, we analyze the slot representation of the i-th object, denoted as $o^i_{s,c,p}$ as in Figure 1 (where $s$ is shape, $c$ is color, and $p$ is position). We collect GRU update-gate activations $z^i_{t,d}$ for each slot dimension $d \in \mathbb{R}^D$ and compute a slow mode rank vector $r^i \in \mathbb{R}^D$.
>
> We then compute the Spearman’s rank correlation to analyze whether the slow modes of each dimension show consistency of the ordering (rank) of these stable dimensions across slots with respect to factors:
>
> $\rho = \mathrm{Spearman}\left( r^i,\, r^j \right)$
>
> for (1) object pairs with identical factors and (2) object pairs with different factors.
>
> - Table 1: Spearman’s rank correlation of GRU slow mode ranks for objects within the same task.
>
> | Evaluation Setting Ei / Tj | Objects with identical factors | Objects with different factors |
> | --- | --- | --- |
> | E0 / T0 | 0.9921 | 0.7931 |
> | E1 / T1 | 0.9920 | 0.7958 |
>
> We observe a clear gap between the two conditions. Objects sharing identical factors exhibit a nearly perfect correlation ($\rho \approx 0.99), showing that slow-mode GRU dimensions are highly consistent across slots with the same semantics. Conversely, although correlations for different-factor pairs are numerically substantial, the consistent and significant gap indicates the presence of distinctive factor-specific slow-mode dimensions. These specialized slow modes allow for the selective preservation of unique semantic information.
>
> These findings directly align with the geometric observation in Figure 2, where slots form distinct homogeneous regions for identical factors within intra-task settings (e.g. $E0/T0$ or $E1/T1$). Consequently, the **GRU effectively performs an identical-mapping operation conditioned on each object’s intrinsic variability, stabilizing slots that share the same semantics along these consistent slow-mode dimensions.**
>
> 3. Preservation of Slow Mode Structure Under Continual Learning
>
> We then examine whether the observed structural robustness persists under continual settings. Specifically, we compute correlations for (1) samples from the previous task (intra-task) and (2) inter-samples across previous and novel tasks (inter-task). **The primary purpose of this analysis is to test whether the GRU slow modes remain consistent during sequential adaptation and thus maintain the task-level separation necessary for continual learning.** For validation, we compare these results with the earlier correlations, treating them as the baseline under standard (non-continual) conditions. We further assess whether the factor-wise separation observed above is preserved across objects from the novel task.
>
> For validation, we explicitly compare results against the Table 1 correlations, treating them as the baseline under standard (non-continual) conditions.
>
> - Table 2: Spearman’s rank correlation of GRU slow mode ranks under the continual learning setting
>
> | Evaluation Setting (Ei/Tj) | Objects with identical factors | Objects with different factors |
> | --- | --- | --- |
> | E0/T1 | 0.9935 | 0.8054 |
> | E0/T1 VS E1/T1 | - | 0.7929 |
>
> Results in Table 2 strongly support the preservation of factor-wise homogeneity through the GRU dynamics:
>
> - Intra-Task Retention: The intra-task correlation for the previous task (E0/T1) remains high and consistent with the E0/T0 or E1/T1 baseline (Table 1). This indicates that the GRU performs an identical-mapping operation even for previously seen samples that do not appear in the current training set.
> - Inter-Task Separation: Crucially, the inter-task correlation (E0/T1 VS E1/T1) closely matches the correlation observed for objects with different semantic factors under the standard (non-continual) condition (Table 1). This demonstrates that both tasks retain their distinctive factor-specific slow-mode dimensions, confirming that structural factor-wise separation is robustly maintained under continual settings.

---

> ### Author Response · Authors · 2025-11-21
> **Follow up**
>
> These **consistency proves that the GRU dynamics provide a robust mechanism for task-level separation by stabilizing factor-specific slow modes**, which is the necessary structural foundation for effective continual object-centric learning.
>
> 4. Summary
>
> We summarize the results this analysis as follows:
>
> Motivated by the degradation observed in our ablation study, we analyzed the emergence of factor-wise homogeneity and task-level separation in continual learning through the lens of **GRU slow modes**. Our analysis shows that the GRU’s marginally stable slow modes **effectively perform an identical-mapping operation** conditioned on each object’s intrinsic variability, **stabilizing slots that share the same semantic factors**. This slow-mode mechanism successfully explains both the robustness of factor-wise homogeneity and the natural task separation observed in Slot Attention under continual learning.
>
> ### W3 & Q2. Causal Link between Factor-Wise Homogeneity and DPR Effectiveness
>
> We acknowledge the reviewer’s comment of showing that whether factor-wise homogeneity property is a direct prerequisite for the effectiveness of our DPR strategy. In response to the reviewer’s suggestion, we conducted additional experiments during the rebuttal period to verify the causal link between factor-wise homogeneity and the effectiveness of DPR. Experimental results empirically supports without pre-existing structural alignment such as factor-wise homogeneity, the DPR strategy is unable to effectively improve degradation of previous tasks.
>
> We kindly ask the reviewers to refer to **Appendix A.5 (updated version, red highlight)** for detailed clarification.
>
> 1. Experiments
>
> We conducted comparative experiments: applying DPR (1) where latent representations (slots) do not exhibit factor-wise homogeneity property, and (2) when the representations of slots are disrupted to diminish this property.
>
> 2. Experiment Results
> - Table 1. With and without DPR on C-Tetrominoes evaluated previous task (E0) trained on novel task (T1). We report average of 5 runs.
>
> |  | FG-ARI$ _{\pm std.}$ (E0 / T1) | FG-ARI$ _{\pm std.}$ (E0 / T1) |
> | --- | --- | --- |
> | Model | Without DPR | With DPR |
> | SlotMLP | 0.3285 $_{\pm 0.04}$ | 0.4575 $_{\pm 0.1}$ |
> | MONet | 0.4351 $_{\pm 0.06}$ | 0.4441 $_{\pm 0.04}$ |
> | Slot w/o GRU | 0.3583 $_{\pm 0.04}$ | 0.5754 $_{\pm 0.02}$ |
> | Slot Attention | 0.4139 $_{\pm 0.03}$ | **0.9683** $_{\pm 0.01}$ |
>
> As shown in the Table 1:
>
> - Baselines without Homogeneity: Methods lacking this property—such as SlotMLP and MONet—showed little to no performance gains on the previous task (E0/T1) when trained with DPR, compared to the significant gains observed with Slot Attention.
> - Disrupted Homogeneity: Crucially, when we disrupted Slot Attention’s homogeneity by ablating the GRU (removing the iterative gating dynamics), the benefits of DPR largely diminished.
> - Non-OCL Model with reconstruction objective: Figure 5 (in the main paper) further demonstrates that a standard Variational Autoencoder (VAE) also fails to benefit from DPR.
> 3. Conclusion
>
> These results empirically supports that DPR is effective precisely because ****Slot Attention provides a well-separated and consistently aligned latent representation space (factor-wise homogeneity). Without this pre-existing structural alignment, the DPR strategy cannot effectively improve degradation of previous tasks.
>
> - References
>
> [1] Tallec et al., "Can recurrent neural networks warp time?", ICLR 2018.
>
> [2] Krishnamurthy et al., "Theory of Gating in Recurrent Neural Networks", Phys. Rev. X 2022.
>
> [3] Can et al., "Gating creates slow modes and controls phase-space complexity in GRUs and LSTMs", MSML 2020.
>
> [4] Linderman et al., “Reverse Engineering Recurrent Neural Networks with Jacobian Switching Linear Dynamical Systems,” NeurIPS 2021.
> [5] Miller & Hardt, “Stable Recurrent Models,” ICLR 2019.[1] Tallec et al., "Can recurrent neural networks warp time?", ICLR 2018.

---

> > ### Comment · Reviewer_jYDk · 2025-11-25
> >
> > Thank you for the detailed response.
> >
> > The analysis presented in the paper, along with your replies, is quite comprehensive and extensive. However, these characteristics can still be viewed as certain inherent features of slots and are applicable in many downstream tasks.
> >
> > The necessity for these traits to independently constitute an acceptable paper requires further clarification from you. (Especially considering that reviewer ZFMM pointed out several flaws regarding the writing of the paper.)

---

> ### Author Response · Authors · 2025-12-03
>
> ### Clarification of our contribution
>
> We emphasize that our primary contribution is to uncover very strong property of Slot Attention which exhibits factor-wise and inter-task feature separation, that is not yet sufficiently examined in the literature and also its significant role in continual object-centric learning.
> Building on this, our work then identifies how this structure emerges within Slot Attention and provides the explanation for why it remains stable, ultimately enabling a simple and practical continual OCL strategy (DPR).
>
> We agree that factor-wise homogeneity is an inherent feature of Slot Attention. Our work is motivated by understanding how these inherent features behave specifically in continual learning settings where models typically suffer from interference and forgetting. We show that Slot Attention’s inherent factor-wise homogeneity is not only present but also preserved under sequential adaptation, and that this preservation directly leads to stable inter-task separation—a phenomenon that has not been examined in prior OCL or continual learning research offering an insightful contribution to the community.
>
> We consider this finding to be a novel and insightful contribution to the community for the following reasons:
>
> 1. **Absence of prior analysis of our findings**
>
> Although factor-wise homogeneity is an inherent feature of Slot Attention, no prior work has directly reported or analyzed this property, nor its preservation under sequential adaptation. Existing research has also not examined how this structure emerges both within standard OCL (single-task) settings and under continual learning scenarios. Such an investigation contributes more than merely confirming an expected observation; it provides a deeper understanding of the model’s underlying behavior.
>
> In the related work section, we reviewed prior studies analyzing slot representations. Compared to our work, papers such as [1,2,3] introduce explicit architectural or representational modifications to obtain disentangled slots, rather than examining the inherent geometry of the unmodified Slot Attention. Similarly, papers [4,5] assume that object-centric models learn factor-wise representations: [4] notes that object-centric factorization aligns with causal variables because causal generative mechanisms typically act over independent objects and attributes, while [5] shows that object-level independence supports compositional generalization by enabling recombination of independently encoded object factors. However, none of these works investigates how Slot Attention itself organizes its latent space or whether it inherently forms factor-wise aligned regions.
>
> We also reviewed OCL works that go beyond a single task, including [6,7,8]. While these studies evaluate generalization, robustness, or transfer of object-centric models, or propose methods to improve the reusability of OCL representations for segmentation, the implicit behavior of Slot Attention under the continuous introduction of new information has not yet been systematically investigated. As a result, the emergence and preservation of factor-wise homogeneity in continual learning settings remain unexplored in the existing literature.
>
> Therefore, our work fills this gap by demonstrating a previously unexamined property. We provide the first analysis of factor-wise homogeneity in Slot Attention, its preservation in continual settings, and the mechanism by which it emerges and persists.
>
> 2. **A Non-trivial phenomenon and unique to Slot Attention from OCL Perspectives**
>
> Through our experiments and analysis, we show that this effect is neither trivial nor generic but a unique structural behavior of Slot Attention. This demonstrates that our work identifies and analyzes a meaningful and previously unexamined property, constituting a non-trivial contribution rather than an obvious or superficial one.
>
> Our experiments (Appendix A.4) show that the behavior does not appear in other object-centric models. Thus, although factor-wise homogeneity is an inherent feature of Slot Attention, it is not a trivial or generic consequence of OCL architectures. Neither MONet nor SlotMLP exhibits the same factor-level alignment or cross-task consistency, and the phenomenon disappears when the GRU is removed from Slot Attention—while it remains present in advanced slot-based models such as DINOSAUR and BO-QSA (Appendix A.13). This indicates that the phenomenon is mechanism-dependent and specific to Slot Attention’s iterative attention and gating dynamics, rather than a universal property of slot-based models.
>
> These results reinforce the validity of our finding and analysis regarding Slot Attention’s factor-wise homogeneity and its preservation in continual settings.

---

> ### Author Response · Authors · 2025-12-03
> **Follow up**
>
> 3. **Non-trivial Inter-task separation phenomenon from Continual Learning perspectives**
>
> The reviewer questions whether our analysis holds independent research value, suggesting that identifying Slot Attention’s structural behavior may be too trivial to serve as a meaningful contribution. We believe that revealing this structure is not only meaningful within OCL but also is directly relevant to a core challenge in continual learning: task-wise interference. Under this perspective, understanding Slot Attention’s factor-wise homogeneity—and specifically its preservation under sequential updates—provides insight that extends beyond object discovery and explains why Slot Attention behaves unusually well in continual settings.
>
> The rationale is grounded in continual learning literature, where works such as [9,10,11,12] consistently show that the representations of old tasks are susceptible to changes induced by new-task learning—manifested as logit or representation drift [9,11], prototype shift and boundary distortion [10], and cross-class feature collision that leads to decision-boundary collapse [12]. These studies collectively demonstrate that achieving stable task-level separation is far from trivial, even for supervised networks.
>
> > Intuitively, catastrophic forgetting is caused by overlapping or confusion between the representations of new and old classes in the feature space. When learning new classes, the decision boundary for previous classes can be dramatically changed, and the unified classifier is severely biased. [10]
> >
>
> Therefore, our analysis does not merely document an internal OCL property; it identifies a structural mechanism of Slot Attention—factor-wise homogeneity and its preservation—that directly addresses the representational instability highlighted in continual learning research. This shows that our contribution is neither trivial nor restricted to OCL, but provides insight into how the structural behavior of Slot Attention connects directly to the interference-related challenges emphasized in both continual object-centric learning and broader continual learning.
>
> [1] Chang et al., "Hierarchical Abstraction for Combinatorial Generalization in Object Rearrangement”, NeurIPS workshop 2022.
>
> [2] Singh et al., "Neural Systematic Binder”, ICLR 2023.
>
> [3] Wu et al., "Neural Language of Thought Models”, ICLR 2024.
>
> [4] Mansouri et al., "Object-centric Architectures Enable Efficient Causal Representation Learning”, ICLR 2024.
>
> [5] Wiedemer et al., "Provable Compositional Generalization for Object-Centric Learning”, ICLR 2024.
>
> [6] Dittadi et al., "Generalization and Robustness Implications in Object-Centric Learning”, ICML 2022.
>
> [7] Didolkar et al., "On the Transfer of Object-Centric Representation Learning", ICLR 2025.
>
> [8] Pan et al., "Object Pursuit: Building a Space of Objects via Discriminative Weight Generation”, ICLR 2022.
>
> [9] Buzzega et al., "Dark Experience for General Continual Learning: a Strong, Simple Baseline”, NeurIPS 2020.
>
> [10] Zhu et al., "Prototype Augmentation and Self-Supervision for Incremental Learning”, CVPR 2021.
>
> [11] Venkatesh et al., "Anatomy of Catastrophic Forgetting: Hidden Representations and Task Semantics”, ICLR 2021.
>
> [12] Kim et al., "Cross-Class Feature Augmentation for Class Incremental Learning”, AAAI 2024.

---

### Official Review · Reviewer_ZFMM · 2025-11-02

**Soundness:** 3
**Presentation:** 2
**Contribution:** 3
**Rating:** 4
**Confidence:** 3

**Summary:**

The paper studies an interesting and less explored topic about combining object-centric learning and continual learning. The idea of using Slot Attention’s factor-wise homogeneity for continual learning is quite novel. The proposed method is simple and seems effective. The experiments are a bit limited, but with more comprehensive studies, the work could be much stronger.

**Strengths:**

- originality: 4/5
- quality: 4/5
- clarity: 3/5
- significance: 3/5

**Weaknesses:**

W1
---
Section 2 Related Work: Should review OCL literatures.
Especially various Slot Attention variants, e.g., BO-QSA, ISA and MetaSlot.

It is necessary to experiment or at least discuss the potential effects of different OCL decoders, as the spatial broadcast CNN decoder was proposed 5 years ago, while there are many advanced OCL decoders have been proposed:
- auto-regressive-based: SLATE, using conditional Transformer as the decoder;
- spatial broadcast-based: DINOSAUR, using MLP as the decoder (similar to CNN as the decoder, used in this paper, according to Line 138-139);
- de-noising-based: SlotDiffusion, using conditional Diffusion model as the decoder.
The authors can either conduct the suggested experiments on these three OCL methods separately, or on a unified OCl method, VVO at once.

Including such experiments or discussions could make this work more complete and more impactful to broader audiences.


W2
---
Typo:
In Line 144, `where and M is the` should remove "and".


W3
---
Line 191, the authors included SlotMLP, which is an earlier work of Slot Attention (both fairly old). However, there are many later works with great improvements, e.g., BO-QSA, ISA and MetaSlot, which should be included in the analysis.


W4
---
Line 202,
> Since the model has not yet encountered images from (E1) during training on (T0), this separation suggests that the behavior is primarily determined by the encoder and slot attention modules rather than influenced by the decoder.

In Section 4.3, DPR is described, i.e., freezing the encoder and slot attention. So DPR is not applied here, right?

Besides, based on the well separation of E1/T0, how was that only the encoder and slot attention matters concluded? The logic is unclear.


W5
---
Line 212.
> inter-task representations For each slot,

There should be a "." before "For".


W6
---
Line 215,
> one highlights inter-task similarity, the other emphasizes within-task consistency

According to your contexts, "within-task" should be "intra-task".


W7
---
Line 287
> preserve factor-wise homogeneous
Should be "homogeneity".


W8
---
Line 296-301,
> DPR is based on two core components: (1) we freeze the encoder and slot attention module and only fine-tune the decoder, in order to maintain factor wise separated representations observed in Slot Attention space S; and (2) we introduce a Post Replay (PR) strategy, wherein the model is fine-tuned after the task training phase, i.e., training on the current task Tt without any continual learning methods, thus completely excluding sources of interference during initial slot representation learning.

Unclear and repetitive writing. Please reoganize and polish it.


That said, I am willing to change my rating if my conerns are addressed.


Reference
---
- BO-QSA: Improving Unsupervised Object-centric Learning with Query Optimization
- ISA: Invariant Slot Attention: Object Discovery with Slot-Centric Reference Frames
- MetaSlot: Break Through the Fixed Number of Slots in Object-Centric Learning
- SLATE: Illiterate DALL-E Learns to Compose
- DINOSAUR: Bridging the Gap to Real-World Object-Centric Learning
- SlotDiffusion: Object-Centric Generative Modeling with Diffusion Models
- VVO: Vector-Quantized Vision Foundation Models for Object-Centric Learning

**Questions:**

Please refer to the former section.

---

> ### Author Response · Authors · 2025-11-21
>
> We thank the reviewer (**ZFMM**) for their thoughtful and constructive feedback. We address the reviewer's concerns as follows:
>
> ### W1-1. Limited review of OCL literatures
>
> We agree that reviewing the broader landscape of OCL methods is important. We would like to clarify that, although a few works were initially missing (e.g., MetaSlot), we did discuss prior studies focused on understanding and improving slot representations—such as BO-QSA and ISA—in Section 2, paragraph 2 (Representation Analysis of Slot Attention).
> To further address the concern regarding broader coverage, we have strengthened the Related Work section (**red highlight**) to provide a more comprehensive overview of the OCL community, explicitly incorporating advanced Slot Attention variants, including MetaSlot.
>
> ### W1-2. Discussion of the potential effects of different OCL decoders
>
> We acknowledge the importance of discussing the potential effects of different OCL decoders. To address this issue, we added **Section 5.5 (Discussion & Limitation, red highlight)** to the revised manuscript, where we explicitly discuss
>
> - (1) how various decoder architectures may influence the behavior and applicability of our approach,
> - and (2) which decoder variations are covered in this work and which are not.
>
> We agree with the potential of advanced decoders; utilizing a decoder with strong robustness or high generalization capabilities could indeed improve reconstruction and generation quality even without DPR. While our experiments cover Slot Attention with a spatial broadcast decoder and DINOSAUR with an autoregressive transformer decoder, we do not explore a broader spectrum decoders, such as diffusion-based object-centric decoder. This constitutes a limitation of the present work and an interesting direction for future research
>
> However, our core contribution lies in demonstrating that Slot Attention itself possesses inherent robustness against inter-task overlap in continual learning settings. We believe that addressing this fundamental issue of latent interference solely through the decoder has inherent limitations, as a identifiable latent input is required for the decoder to operate effectively in a sequential learning environment.
>
> Also we wish to highlight that we have verified our core finding with advanced architectures:
>
> - We provide additional experiments using DINOSAUR, which employs a pre-trained Vision Transformer (DINO) encoder paired with an auto-regressive Transformer decoder for complex real-world datasets like COCO and Pascal VOC (as noted in Section 5.3).
> - The results (Table 1 and Figure 24 in the Appendix) show consistent performance improvements with DPR even on these advanced architectures. This demonstrates that factor-wise homogeneity provides a robust inductive bias that extends beyond synthetic settings and simple CNN-based implementations.
>
> ### W3. Analysis with recent baselines
>
> We acknowledge the reviewer’s concern regarding the need to analyze more advanced OCL methods beyond the relatively old SlotMLP. Here, we would like to clarify the motivation behind our comparison.
>
> SlotMLP replaces the Slot Attention mechanism with a simple MLP and therefore serves as a baseline for isolating the Slot Attention mechanism to analyze its intrinsic properties. Introducing advanced methods into this specific analysis would blur the core behavior we aim to study and limit our ability to examine the fundamental properties of the original Slot Attention mechanism.
> While we agree that verifying generalizability is important, we note that several of our experiments demonstrate applicability to broader Slot Attention variants.
>
> 1. Clarifications
>
> Our primary focus was to analyze the intrinsic properties of the original Slot Attention mechanism itself. To rigorously isolate this effect, we employed SlotMLP as a control baseline because it shares the same overall architecture but replaces the attention mechanism with a simple MLP. This comparison allows us to attribute the observed property directly to the SA mechanism itself, rather than other architectural components.
>
> While methods like BO-QSA, ISA, and MetaSlot are significant advancements, they introduce additional inductive biases (e.g., quantization) on top of Slot attention. We were concerned that including them in the primary analysis might make it difficult to disentangle the core behavior of Slot Attention from these added components.
>
> 2. Highlights of our previous experiments
>
> However, we fully agree on the importance of verifying generalizability. We wish to highlight that experiments in Section 5.3 (Table 1) have demonstrated with the DINOSAUR, which employs a pre-trained Vision Transformer (DINO) encoder paired with an auto-regressive Transformer decoder, confirming that the factor-wise homogeneity persists even in advanced, transformer-based architectures, suggesting that our insights are applicable to these broader Slot Attention variants as well.

---

> ### Author Response · Authors · 2025-11-21
> **Follow up**
>
> ### W2 & W5 & W6 & W7
>
> We sincerely thank the reviewer for the detailed feedback. We will correct the typo in the final version of the manuscript (**red highlight**).
>
> ### W4. Unclear logic of isolating decoder’s effects
>
> We acknowledge the reviewer’s concerns of unclear logic of our claims. We revised the manuscript to clarify this reasoning in the final version (**red highlight**).
>
> 1. Clarifications
>
> First, we clarify that the results in Figures 1 and 2 represent an analysis of the standard Slot Attention model itself, without the application of DPR.
>
> Regarding the conclusion that the behavior is determined by the Encoder and Slot Attention:
>
> - Evidence from Unseen Distributions (E1/T0): If the decoder's reconstruction objective were the dominant factor in organizing factor-wise homogeneity, we would not expect such well-separated structures to emerge for unseen samples (E1/T0), unless the decoder was perfectly generalizable across tasks.
> - Decoder as the Bottleneck (Figure 4): However, Figure 4 demonstrates that the final reconstructed images are significantly degraded, even though the slot representations themselves remain well-sorted and separated. This discrepancy indicates that the decoder lacks the generalization capability to handle these features, effectively acting as a bottleneck.
> - Since the latent separation persists (as seen in E1/T0 and E0/T1) despite the decoder's inability to reconstruct them correctly, we conclude that the separation is an intrinsic property established by the Slot Attention mechanism's alignment before the decoder processes the features, rather than being induced by the decoder.
>
> ### W8 Unclear and repetitive writing
>
> We acknowledge the reviewer’s concerns of unclear and repetitive writing demonstrating our method. We revised this section (**red highlight**) to remove redundancy and clearly convey these distinct roles.
>
> 1. Clarifications
>
> We revised the manuscripts to succinctly present Decoder-Only Post Replay (DPR) as a minimal implementation driven by two distinct intuitions.
>
> - Decoder-Only: Freezing the encoder and Slot Attention to strictly maintain the factor-wise separated representations established in the Slot Attention.
> - Post Replay (PR): Applying replay after the task training phase to completely exclude interference from continual learning objectives (e.g. replay sample ratio bias[1,2]) during the slot representation learning on novel task.
>
> ### References
>
> [1] Wu et al., "Large Scale Incremental Learning", CVPR 2019.
>
> [2] Rolnick et al., "Experience Replay for Continual Learning", NeurIPS 2019.”

---

> ### Comment · Reviewer_ZFMM · 2025-11-26
> **Update 1**
>
> Thank you for your detailed rebuttal.
>
> Some of my concerns were addressed, but `W1` and `W3` were not responded to explicitly or directly.
>
> In particular, the authors’ response to `W1` is even more confusing:
>
> > However, ... We believe that addressing this fundamental issue of latent interference solely through the decoder has inherent limitations, as an identifiable latent input is required for the decoder to operate effectively in a sequential learning environment.
>
> Regarding `W3`, the response still avoids the core issue: **SlotMLP is an outdated technique and is no longer adopted by mainstream OCL methods. Using such a “dead” baseline is not meaningful.**
>
> By the way, a brief but direct response is sometimes more convincing than lengthy details.
>
> In summary, I will keep my original rating and leave the final decision to the collective judgment of the other reviewers.

---

> ### Author Response · Authors · 2025-12-03
> **Response to Update 1**
>
> ### W1
>
> To reduce the reviewer’s concern, we re-clarified all the pointed decoder approaches as following tables.
> We already cover two approaches in the paper, and added discussion about SlotDiffusion.
>
> | Type | Network | Model | Dataset | Quantitative | Qualitative |
> | --- | --- | --- | --- | --- | --- |
> | Spatial broadcast-based | CNN | Slot Attention | C-Terominoes, C-CLEVR | Figure 3 | Figure 1, Figure 4 |
> | Auto-regressive-based | Transformer | DINOSAUR | COCO, Pascal | Table 1 | Figure 24, Figure 25 |
>
> We do not extend our analysis to denoising-based decoders such as SlotDiffusion[1], which we consider to have a different underlying objective from our work. SlotDiffusion[1] aims to improve the generative capacity of object-centric models which object slots are used as n guide to Latent Diffusion Model decoder learns to denoise image feature maps.
>
> > Goal of this work is to improve the generative capacity of object-centric models on complex data while preserving their segmentation performance. The Latent Diffusion Model decoder learns to denoise image feature maps guided by object slots, providing training signals for object discovery. [1]
> >
>
> We discuss the potential impact of decoder generalization in **Section 5.5** and identify the exploration of broader decoder families as an important direction for future work.
>
> [1] Wu et al., "SlotDiffusion: Object-Centric Generative Modeling with Diffusion Models”, NeurIPS 2023
>
> ### W3. Advanced Slot Attention
>
> The use of SlotMLP is not for showing the SOTA performance, but for showing more clear analysis. However, to more clearly resolve the reviewer’s concern, we add additional analysis of a variant of the mentioned advanced Slot Attention, BO-QSA, which employs learnable object-binding queries for slot initialization and a bi-level optimization for more stable training.
>
> We kindly ask the reviewers to refer to **Appendix A.13 (red highlight)** in the updated manuscript.
>
> 1. **Analysis of advanced Slot Attention variant (BO-QSA)**
>
> Qualitative t-SNE visualizations **(Figure 26 and Figure 27)** show that BO-QSA also exhibits factor-wise homogeneity and clear inter-task separation in its slot representation space. The quantitative evaluation of inter-task separation **(Figure 28)** further supports this observation: BO-QSA produces results comparable to Slot Attention while maintaining a substantial gap over baselines that do not implement the Slot Attention mechanism.
>
> 2. **Performance of with and without applying DPR to BO-QSA**
>
> We provide object discovery performances (FG-ARI) of BO-QSA trained on continual settings (C-Tetrominoes). We also evaluate performance of applying DPR on BO-QSA. While, BO-QSA also exhibit critical degradation of performance on previous task after training on novel task (E0/T1), applying DPR on BO-QSA significantly improvements performances on previous tasks while maintaining performance on novel task. We ask the reviewers to refer to **Table 16**  in the updated manuscript.
>
> |  | E0/T0 | E0/T1 | E1/T1 |
> | --- | --- | --- | --- |
> | SA | 0.9986$_{\pm 0.00}$ | 0.4171$_{\pm 0.03}$ | 0.9979$_{\pm 0.00}$ |
> | SA + DPR | 0.9988$_{\pm 0.00}$ | 0.9981$_{\pm 0.00}$ | 0.9970$_{\pm 0.00}$ |
> | BO-QSA | 0.9998$_{\pm 0.00}$ | **0.3396**$_{\pm 0.08}$ | 0.9998$_{\pm 0.00}$ |
> | BO-QSA + DPR | 0.9988$_{\pm 0.00}$ | **0.9654**$_{\pm 0.02}$ | 0.9953$_{\pm 0.00}$ |
>
> 3. **Clarification of the purpose of our experiments**
>
> To clarify the purpose of our experiments, we include the table below, which summarizes all models analyzed in the paper (and throughout the rebuttal). The table highlights which components of the Slot Attention mechanism each model retains or removes.
>
> |  | Slot$^{\dagger}$ | random init.* | Cross attention* | GRU* | MLP* | Iterative* | Qualitative |
> | --- | --- | --- | --- | --- | --- | --- | --- |
> | Slot Attention | O | O | O | O | O | O | Figure 1 (a) |
> | $\mathbf{Ablations}$ |  |  |  |  |  |  |  |
> | SlotMLP (w/o Slot Attention mechanism) | O | X (MLP) | X | X | X | X | Figure 1 (b) |
> | Slot Attention w/o random init. | X | X (embedding) | O | O | O | O | Figure 11 (a) |
> | Slot Attention w/o GRU | O | O | O | X | O | O | Figure 11 (b) |
> | $\mathbf{Slot \ Attention \ variants}$ |  |  |  |  |  |  |  |
> | DINOSAUR | O | O | O | O | O | O | Figure 24 |
> | BO-QSA | O | X (embedding) | O | O | O | O** | Figure 26 (b) |
> | $\mathbf{OCL \ method}$ |  |  |  |  |  |  |  |
> | MONet | O | - | - | - | - | - | Figure 12 (c) |
> | $\mathbf{Non-OCL \ method}$ |  |  |  |  |  |  |  |
> | VAE | X | X  | X | X | X | X | Figure 15 (b) |
>
> $^{\dagger}$: Object-wise discrete representations
>
> *: Mechanisms that are implemented in Slot Attention
>
> **: Bi-level Optimization (previous iterations are detached)
>
> -: Different implementation compared to Slot Attention

---

### Author Response · Authors · 2025-11-21
**Official Responses to Reviewers concerns**

We sincerely thank the reviewers for their thoughtful and constructive feedback. We would like to share some of our responses to several reviewer’s concerns.

Below, we summarize the key concerns raised by the reviewers, focusing on their main questions regarding the novelty and explanations of our findings.

- Reviewer **jYDk**: Points out that factor-wise homogeneity is not a novel discovery but rather an expected property of well-trained representation models (Weakness 1). Also, request for mechanistic explanation of how Slot Attention’s iterative attention and GRU updates specifically give rise to this phenomenon (Question 1). Points out uncertainty regarding whether factor-wise homogeneity is a unique property of Slot Attention (Question 1), and raises concerns about the unclear causal link between this property and the effectiveness of DPR (Question 2).
- Reviewer **cgDZ**: Points out that the paper lacks a theoretical explanation for why factor-wise homogeneity emerges (Weakness 1).

## **1. Clarification of Our Main Contribution**

Our main contribution is to more clearly identify the factor-wise homogeneity present in Slot Attention representations, a behavior that has been noted only implicitly or observationally in a few prior works.

More importantly, we focused on sharing the finding that **the strong separation of representations across tasks originates from this factor-wise homogeneity in Slot Attention**, and that this **structural property can substantially improve continual object-centric learning** even when combined with simple constraints such as DPR or PR.

## **2. Addition of Deeper Analysis for Understanding the Cause of the Phenomena (via GRU dynamics in Slot Attention)**

We acknowledge the importance of providing explanation of the emergence of factor-wise homogeneity and its stability in continual learning. To address the reviewer (**jYDk** and **cgDZ**)’s concerns, we conducted an in-depth analysis of GRU dynamics within Slot Attention. Our analysis demonstrate that GRU dynamics serve as the key mechanism supporting both factor-wise consistency and continual task separation, providing the structural foundation for effective continual object-centric learning.

A revised and extended version of this analysis has been added to **Appendix A.3 (red highlights)** in the updated manuscript, and we kindly ask the reviewers to refer to this section for the full details.

Grounded in prior GRU studies showing that gating creates slow modes—hidden-state dimensions that remain stable due to consistently small update gates—these slow modes are understood as stable structural channels that preserve key semantic information over long horizons. The central question, is whether these recurrent slow-mode dynamics stabilize factor-wise homogeneity by acting as object-conditioned identical-mapping channels, and whether this same mechanism also accounts for the persistent task-level separation observed during continual adaptation.

To examine this connection, the update gate is used as an empirical indicator of each dimension’s stability, yielding a “slow-mode rank’’ that reflects how strongly each hidden dimension behaves as a slow mode. This ranking enables a direct test of factor-wise homogeneity by evaluating whether slots that share the same semantic factors rely on the same highly stable dimensions. A consistency analysis (Spearman's rank correlation $\rho$) is then performed to examine whether these slow-mode dimensions align across slots (objects) that share the same semantic factors.

---

> ### Author Response · Authors · 2025-11-21
> **Follow-up**
>
> Continued from 2. Addition of Deeper Analysis for Understanding the Cause of the Phenomena (via GRU dynamics in Slot Attention)
>
> ### **Summary of Analysis Results**
>
> (1) GRU Slow Mode Consistency Across Semantic Factors
>
> - Purpose: To test whether the GRU’s slow-mode dimensions directly correspond to semantic factors and remain consistently aligned across slots that share identical factors. This evaluates whether slow modes can serve as the structural basis of factor-wise homogeneity.
> - Result: Objects sharing identical semantic factors exhibit a high correlation in their slow-mode ranks (e.g., $\rho \approx 0.99$), showing that slow-mode GRU dimensions are highly consistent across slots with the same semantics. Conversely, although correlations for different-factor pairs are numerically substantial (e.g., $\rho \approx 0.79$), the consistent and significant gap indicates the presence of distinctive factor-specific slow-mode dimensions. These results align with our quantitative and qualitative observations and demonstrate that the GRU effectively performs an identical-mapping operation conditioned on each object’s intrinsic variability, stabilizing slots that share the same semantics along these consistent slow-mode dimensions.
>
> (2) Preservation of Slow Mode Structure Under Continual Learning
>
> - Purpose: To examine whether the GRU’s slow-mode structure remains stable under continual learning, thereby retaining factor-wise consistency for previously learned tasks while preserving separation from newly introduced tasks.
> - Result: The intra-task correlation for the previous task maintain a high correlation in their slow-mode ranks (e.g., $\rho \approx 0.99$), indicating that the GRU performs an identical-mapping operation even for previously seen samples. The inter-task correlation closely matches the correlation observed for objects with different semantic factors under the standard condition, demonstrating that both tasks retain their distinctive factor-specific slow-mode dimensions. This confirms that structural factor-wise separation is robustly maintained under continual settings and that the GRU dynamics provide a robust mechanism for task-level separation by stabilizing factor-specific slow modes.
>
> ### **Derived Conclusion of the Analysis**
>
> Our analysis shows that the GRU’s marginally stable slow modes effectively perform an identical-mapping operation conditioned on each object’s intrinsic variability. This mechanism stabilizes slots that share the same semantic factors, providing the structural basis for the robustness of factor-wise homogeneity. Furthermore, the preservation of these factor-specific slow-mode dimensions under continual learning naturally explains the task-level separation observed in Slot Attention. Taken together, these findings demonstrate that GRU dynamics serve as the key mechanism supporting both factor-wise consistency and continual task separation, providing the structural foundation for effective continual object-centric learning.
>
> ## **3. Addition of New Experiments**
>
> ### E1. Is factor-wise homogeneity property unique to Slot Attention?
>
> To response to reviewer **jYDk**’s comments of uncertainty regarding whether factor-wise homogeneity is a unique property of Slot Attention, we reproduced another representative OCL baseline (MONet) to examine whether factor-wise homogeneity is unique to Slot Attention. Experimental results show that factor-wise homogeneity is not a universal feature of Object-Centric Learning models.
>
> We kindly ask the reviewers to refer to **Appendix A.4 (updated version, red highlight)** for detailed clarification.
>
> - As shown in Figure 12 (in Appendix A.4), MONet does not exhibit the latent separation observed in Slot Attention. When representations are labeled by \textit{task pairs}, a large portion of representations from different tasks overlap, making task-wise separation difficult.
> - A similar pattern appears when labeled with \textit{semantic pairs} (Figure 13 in Appendix A.4): MONet shows little to no generalization for the \textit{shape} factor and only mild alignment for \textit{color} and \textit{position}, which remains insufficient to consider the representations well separated compared to Slot Attention.
> - We also evaluated MONet’s inter-task separation following the analysis in Section 4.2 (i.e., inter-task class similarity and nearest neighbor class purity). The results in Figure 14 (in Appendix A.4) further confirm that MONet lacks factor-wise homogeneity in its latent space.
>
> In summary, the results from both the qualitative t-SNE visualizations and the quantitative inter-task separation metrics demonstrates that factor-wise homogeneity is not a universal feature of Object-Centric Learning models. Instead, this desirable property—which is crucial for enabling effective continual learning—is uniquely attributable to the Slot Attention architecture.

---

> > ### Author Response · Authors · 2025-11-21
> > **Follow-up**
> >
> > ### E2. Causal Link between Factor-Wise Homogeneity and DPR EffectivenessE2. Causal Link between Factor-Wise Homogeneity and DPR Effectiveness
> >
> > We acknowledge the reviewer **jYDk**’s comment of showing that whether factor-wise homogeneity property is a direct prerequisite for the effectiveness of our DPR strategy. In response to the reviewer’s suggestion, we conducted additional experiments during the rebuttal period to verify the causal link between factor-wise homogeneity and the effectiveness of DPR. Experimental results empirically supports without pre-existing structural alignment such as factor-wise homogeneity, the DPR strategy is unable to effectively improve degradation of previous tasks.
> >
> > We kindly ask the reviewers to refer to **Appendix A.5 (updated version, red highlight)** for detailed clarification.
> >
> > We conducted comparative experiments: applying DPR (1) where latent representations (slots) do not exhibit factor-wise homogeneity property, and (2) when the representations of slots are disrupted to diminish this property.
> >
> > As shown in the newly added Table 5 (in Appendix A.5):
> >
> > - Baselines without Homogeneity: Methods lacking this property—such as SlotMLP and MONet—showed little to no performance gains on the previous task (E0/T1) when trained with DPR, compared to the significant gains observed with Slot Attention.
> > - Disrupted Homogeneity: Crucially, when we disrupted Slot Attention’s homogeneity by ablating the GRU (removing the iterative gating dynamics), the benefits of DPR largely diminished.
> > - Non-OCL Model with reconstruction objective: Figure 5 (in the main paper) further demonstrates that a standard Variational Autoencoder (VAE) also fails to benefit from DPR.
> >
> > These results empirically supports that DPR is effective precisely because ****Slot Attention provides a well-separated and consistently aligned latent representation space. Without this pre-existing structural alignment, the DPR strategy is unable to effectively improve degradation of previous tasks.

---

### Meta-Review · Area_Chair_cNhe · 2025-12-22

**Summary:**

This paper initially received generally negative recommendations (2, 4, 4, and 6). While the rebuttal substantially improved clarity and added several analyses/experiments, the AC believes some critical concerns remain unresolved which are summarized as follows.

**Reviewer Concerns:**

- Limited analysis: While the empirical observation of factor-wise homogeneity is intriguing, it is not characterized rigorously enough to support the paper’s strongest claims. Much of the evidence remains observational (often in controlled synthetic settings), and the rebuttal’s GRU “slow-mode” explanation is still indirect—closer to correlational diagnostics than a compelling mechanism that would predict when/why the phenomenon should arise and persist.
- Insufficient evaluation of OCL performance: The paper mainly evaluates the performance of OCL with localization accuracy and the reconstruction quality. While the authors argued that it is standard practice in the literature, many recent works employ the downstream performance to directly assess the representation quality of OCL. Considering that the attention-based evaluation is potentially misleading in the current setting (see below), directly evaluating the representation quality could be valuable.
- Unclear justification of core contribution: The rebuttal shows that attention masks from Slot Attention can remain highly robust under continual updates, whereas decoder-based masks degrade and DPR mainly restores performance by post-hoc decoder fine-tuning. This creates a conceptual tension: if the slot/attention masks already provide stable object masks, the continual-learning “forgetting” problem appears substantially weaker and the necessity of DPR is less convincing; if the main issue is decoder generalization, then the contribution reads more like a decoder-specific fix than a principled advance in continual object-centric learning. The paper would benefit from a clearer framing and stronger evidence that DPR addresses a fundamental continual OCL challenge rather than an artifact of the chosen mask extraction/evaluation pipeline.

**Reviewer Scores:**

Due to the concerns remain after the rebuttal, the AC believes that reviewer o9jF, cgDZ, and cgDZ would likely not change their recommendations.

---

### Decision · Program_Chairs · 2026-01-26

Reject